# Understanding Factual Recall in Transformers via Associative Memories

**Eshaan Nichani**
Princeton University

**Jason D. Lee**
Princeton University

**Alberto Bietti**
Flatiron Institute

## Abstract

Large language models have demonstrated an impressive ability to perform factual recall. Prior work has found that transformers trained on factual recall tasks can store information at a rate proportional to their parameter count. In our work, we show that shallow transformers can use a combination of associative memories to obtain such near optimal storage capacity. We begin by proving that the storage capacities of both linear and MLP associative memories scale linearly with parameter count. We next introduce a synthetic factual recall task, and prove that a transformer with a single layer of self-attention followed by an MLP can obtain 100% accuracy on the task whenever either the total number of self-attention parameters or MLP parameters scales (up to log factors) linearly with the number of facts. In particular, the transformer can trade off between using the value matrices or the MLP as an associative memory to store the dataset of facts. We complement these expressivity results with an analysis of the gradient flow trajectory of a simplified linear attention model trained on our factual recall task, where we show that the model exhibits sequential learning behavior.

## 1 Introduction

One hallmark capability of transformer-based large language models (LLMs) is factual recall (Petroni et al., 2019; Jiang et al., 2020; Roberts et al., 2020). Given a prompt of the form "In what year was George Washington born?" an LLM will correctly respond with "1732." Language models thus act as databases, storing somewhere in their parameters mappings of the form (George Washington, birth year) $\mapsto$ (1732) which can be easily accessed during inference time.

Prior work (Allen-Zhu & Li, 2024) has observed that transformers trained on factual recall tasks can store information at a rate proportional to their parameter count. Other studies (e.g., Meng et al., 2022; Geva et al., 2023; Nanda et al., 2023; Lv et al., 2024) have sought to understand the specific mechanism by which transformers implement factual recall, probing models to understand specifically which transformer blocks "contain" certain facts. However, these studies do not consider the memorization capacity of such constructions, and it is thus an open question to understand how transformers optimally encode such factual information within their weights.

In this work, we show that shallow transformers can use a combination of *associative memories* to obtain near-optimal storage capacity for factual recall tasks. Associative memories store pairs of input-output embeddings through their outer products, and are thus well-suited for modeling the weight matrices of a transformer. Prior work (Bietti et al., 2023) has shown that this associative memory model is a key primitive towards understanding both the representational capacity and optimization dynamics of transformers on synthetic tasks.

Our specific contributions are as follows: In Section 3 we begin by studying the ability of linear and MLP associative memory models to store associations between discrete vocabularies. We prove that when the embeddings are sampled randomly over the sphere, these models can store a number of associations proportional to their parameter count, significantly improving over the case where the embeddings are orthogonal. In Section 4, we introduce a synthetic next-token prediction task which models factual recall. The data distribution consists of prompts containing a subject token $s$ and relation token $r$ hidden amongst a set of noise tokens, which the learner must map to a ground truth answer $a^*(s, r)$. Our main theorem is that a transformer consisting of a single multi-head

self-attention layer followed by an MLP can obtain 100% accuracy when *either* the number of self-attention parameters or MLP parameters scales (up to logs) proportionally with the dataset size. In Section 5, we study the gradient descent dynamics of a single linear self-attention head trained on the synthetic task. We prove that the model undergoes a sequential learning dynamics, consisting of a "hallucination" stage where the model outputs the conditional distribution for the answer based on only the relation. Finally, in Section 6 we complement our constructions with lower bounds, showing that they are optimal up to logarithmic factors. Overall, our work makes progress towards understanding the mechanism by which transformers learn and store factual information.

## 2    RELATED WORK

**Associative memories.**    Associative memories have a long history in the neural computation literature (Hopfield, 1982; Kohonen, 1972; Willshaw et al., 1969). More recently there has been renewed interest in extensions of such models with larger capacity (Krotov & Hopfield, 2016; Demircigil et al., 2017; Lucibello & Mézard, 2024). These have been linked to the attention blocks in Transformers (Ramsauer et al., 2020; Schlag et al., 2021), with (Le et al., 2020; Hoover et al., 2023) in particular using the connection between self-attention and associative memories to design new variants of the attention module. (Radhakrishnan et al., 2020) show that overparameterized autoencoders can also behave as associative memories. However, these connections differs from our work, where we consider instead the role of both self-attention and MLP weights as associative memories, in a similar vein to (Bietti et al., 2023; Cabannes et al., 2024).

**Memorization and factual recall.**    Large language models are known to store vast amounts of factual knowledge in their weights (Jiang et al., 2020; Roberts et al., 2020; Geva et al., 2021). Several recent works in the mechanistic interpretability literature have attempted to understand how transformers store facts (Meng et al., 2022; Geva et al., 2023; Nanda et al., 2023; Lv et al., 2024). Allen-Zhu & Li (2024) empirically studied the memorization capacity for Transformer language models of different sizes trained on synthetic factual recall tasks, and observed near-linear scaling with the number of parameters. Jiang et al. (2024) demonstrate how shallow transformers can solve a related latent concept association task by viewing the weight matrices as associative memories. At a more basic level, several works have studied the memorization capacity of neural networks, using constructions that differ from our associative memory approach, both in the context of regression (Bubeck et al., 2020; Vardi et al., 2021; Madden & Thrampoulidis, 2024) and (next) token prediction (Mahdavi et al., 2023; Kajitsuka & Sato, 2023; 2024; Madden et al., 2024).

**Gradient dynamics.**    Training dynamics of transformer models on various tasks has been a popular recent line of research (Jelassi et al., 2022; Snell et al., 2021; Li et al., 2023; Bietti et al., 2023; Tian et al., 2023; Nichani et al., 2024). Zhang et al. (2024); Mahankali et al. (2024) studied training dynamics of transformers with linear attention on in-context learning tasks. Ghosal et al. (2024) studied the fine-tuning dynamics on a similar factual recall task, showing how training on lesser-known facts may hurt performance. Our emphasis differs in that we consider non-orthogonal embeddings, and require the model to additionally filter out the relevant subject and relation tokens from the noise tokens, which requires learning of the key and query matrices.

## 3    ASSOCIATIVE MEMORIES

In this section, we show that associative memories have a storage capacity on the order of the number of parameters (up to logarithmic factors), which is near-optimal (as we show in Section 6).

**Setup.**    Our setting follows that of Cabannes et al. (2024). Let $[N]$ be the set of input tokens, and $[M]$ be the set of output tokens. Our goal is to store a set of associations given by the function $f^* : [N] \to [M]$. For each input token $x \in [N]$ we assign a corresponding embedding vector $\boldsymbol{e}_x \in \mathbb{R}^d$, and likewise for each output token $y \in [M]$ we associate an unembedding vector $\boldsymbol{u}_y \in \mathbb{R}^d$. We primarily focus on the setting where the embeddings $\{\boldsymbol{e}_x\}_{x \in [N]}$ and $\{\boldsymbol{u}_y\}_{y \in [M]}$ are drawn i.i.d uniformly from the sphere of radius 1. Let $F : \mathbb{R}^d \to \mathbb{R}^d$ be our model which "stores" the associations $f^*$. Given such an $F$, the prediction $\hat{f}(x)$ for $f^*(x)$ is given by the arg-max decoding $\hat{f}(x) := \arg\max_{y \in [M]} \boldsymbol{u}_y^\top F(\boldsymbol{e}_x)$.

## 3.1 LINEAR ASSOCIATIVE MEMORIES

We first consider the case where $F$ is a linear map $F(\boldsymbol{e}_x) = \boldsymbol{W}\boldsymbol{e}_x$.

**Theorem 1.** *Assume that $f^*$ is injective. If $d^2 \gtrsim N \operatorname{poly} \log N$, then with high probability over the draw of the embeddings, there exists a $\boldsymbol{W}$ such that*

$$\arg \max_{y \in [M]} \boldsymbol{u}_y^\top \boldsymbol{W}\boldsymbol{e}_x = f^*(x) \quad \text{for all } x \in [N]. \tag{1}$$

*This capacity is obtained by the construction $\boldsymbol{W} = \sum_{x \in [N]} \boldsymbol{u}_{f^*(x)} \boldsymbol{e}_x^\top$. Furthermore, if $\boldsymbol{W}$ is restricted to be a rank $m$ matrix, then such a $\boldsymbol{W}$ exists when $md \gtrsim N \operatorname{poly} \log N$; this construction is $\boldsymbol{W} = \sum_{x \in [N]} \boldsymbol{u}_{f^*(x)} \boldsymbol{e}_x^\top \sum_{i=1}^m \boldsymbol{v}_i \boldsymbol{v}_i^\top$, where $\boldsymbol{v}_i \in \mathbb{R}^d$ are drawn i.i.d from the standard Gaussian.*

Since $\boldsymbol{W}$ has $d^2$ parameters, Theorem 1 shows that the number of associations that can be stored scales (up to log factors) linearly in the number of parameters. We note that in this linear case, the injectivity assumption on $f^*$ is important, as otherwise the capacity may be as low as $d$, as in (Cabannes et al., 2024). Additionally, we remark that these constructions are easily obtained by gradient descent; the general $\boldsymbol{W}$ construction corresponds to one-step of GD on the correlation loss $L(\boldsymbol{W}) = -\sum_x \boldsymbol{u}_{f^*(x)}^\top \boldsymbol{W}\boldsymbol{e}_x$, while the low-rank construction corresponds to parameterizing $\boldsymbol{W} = \boldsymbol{U}\boldsymbol{V}^\top$ for $\boldsymbol{U}, \boldsymbol{V} \in \mathbb{R}^{d \times m}$, and taking one step of GD on $\boldsymbol{U}$ while $\boldsymbol{V}$ is fixed to random Gaussian initialization. The proof of Theorem 1 is deferred to Appendix B.

**Remarks.** Our setting bears similarity to associative Hopfield networks (Hopfield, 1982), yet differs in that we decode to a fixed discrete set of output tokens $[M]$ rather than exactly matching the target output. This more closely resembles the language modeling framework, and allows us to improve the memorization capacity from $d$ to $d^2$ (McEliece et al., 1987). Next, we note that non-orthogonality of the embeddings is necessary for Theorem 1, as the optimal storage capacity for one-hot embeddings is only $N = d$. Since our constructions are in the regime $N \gg d$, the associative memory $\boldsymbol{W}$ is a *superposition* (Elhage et al., 2022) of the outer products $\boldsymbol{u}_{f^*(x)} \boldsymbol{e}_x^\top$. Finally, we remark that the random, rather than trainable, embeddings setting was also studied in Cabannes et al. (2024). The embeddings can be viewed as global quantities in a larger network, of which the associative memory is implementing some subtask, and is thus not able to optimize these embeddings in order to solve its specific task.

## 3.2 MLP ASSOCIATIVE MEMORIES

Next, we consider the case where $F$ is a two-layer neural network with hidden width $m$; that is $F(\boldsymbol{e}_x) = \boldsymbol{V}^\top \sigma(\boldsymbol{W}\boldsymbol{e}_x)$ for $\boldsymbol{V}, \boldsymbol{W} \in \mathbb{R}^{m \times d}$.

**Theorem 2** (Informal)**.** *If $md \gtrsim N \operatorname{poly} \log N$, then with high probability over the draw of the embeddings, there exists $\boldsymbol{V}, \boldsymbol{W}$ such that*

$$\arg \max_{y \in [M]} \boldsymbol{u}_y^\top \boldsymbol{V}^\top \sigma(\boldsymbol{W}\boldsymbol{e}_x) = f^*(x) \quad \text{for all } x \in [N]. \tag{2}$$

Since the MLP has $2md$ parameters, Theorem 2 shows that the MLP associative memory scheme has storage capacity which is (nearly) linear in the parameter count.

**Proof Sketch.** The construction for Theorem 2 mimics that of Theorem 1, after an appropriate random feature transformation. First, sample the rows of $\boldsymbol{W}$ from the standard Gaussian. Then, set each $\boldsymbol{v}_i = m^{-1} \sum_x \boldsymbol{u}_{f^*(x)} h_k(\langle \boldsymbol{w}_i, \boldsymbol{e}_x \rangle)$, where $h_k$ is the $k$th Hermite polynomial (see Appendix F.1). We then see that

$$F(\boldsymbol{e}_x) = \frac{1}{m} \sum_{i=1}^m \sum_{x' \in [N]} \boldsymbol{u}_{f^*(x')} h_k(\langle \boldsymbol{w}_i, \boldsymbol{e}_{x'} \rangle) \sigma(\langle \boldsymbol{w}_i, \boldsymbol{e}_x \rangle) \approx \sum_{x' \in [N]} \boldsymbol{u}_{f^*(x')} \langle \boldsymbol{e}_x, \boldsymbol{e}_{x'} \rangle^k. \tag{3}$$

for sufficiently large $m$. Such *polynomial* associative memory is reminiscent of that in Krotov & Hopfield (2016), and can store many more associations for large $k$. By choosing $k \approx \log_d m$ and appropriately dealing with concentration, one can obtain the $\tilde{O}(md)$ storage capacity (for technical reasons, we must also use the Neural Tangent Kernel (Jacot et al., 2018) rather the random feature model). The full proof of Theorem 2 is deferred to Appendix B.

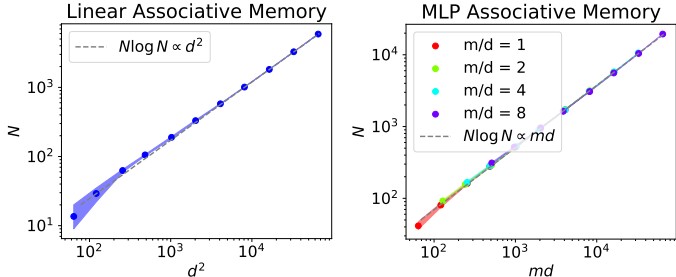

Figure 1: We train linear and MLP associative memories to store $f^*(x) = x$. Left: Linear associative memories require $d^2 \propto N \log N$ parameters to store $N$ associations. Right: MLP associative memories require $md \propto N \log N$ parameters to store $N$ associations, as predicted by Theorem 2.

**On Optimal Storage Capacity.** Prior works (Bubeck et al., 2020; Madden & Thrampoulidis, 2024; Madden et al., 2024) studying the memorization capacity of neural networks focus on the regression setting, and thus do not directly apply to our setup with multiple outputs and a discrete set of output tokens. Other works (Vardi et al., 2021; Kajitsuka & Sato, 2023; 2024) show that one can memorize $N$ arbitrary labels with $\tilde{O}(\sqrt{N})$ parameters, at the expense of using a bit complexity of $\tilde{\Omega}(\sqrt{N})$. Such networks still require $\Omega(N)$ bits, which matches our lower bounds in Section 6. These constructions, however, are unwieldy, and are not learnable if we restrict the precision to be $\operatorname{poly} \log N$. Instead, our constructions are learnable – the linear construction results from one step of GD, while the ReLU construction uses the NTK and can thus be learned via GD on a convex loss. In Corollary 2, we show that a quantized version of the construction from Theorem 1 indeed succeeds with bit precision $\tilde{O}(1)$, and thus more accurately captures realistic training regimes where models do seem to succeed with low precision (Dettmers et al., 2022; Allen-Zhu & Li, 2024).

**Empirical Validation.** In Figure 1, we train both linear and MLP associative memories to store the association $f^*(x) = x$. Given a fixed model size $(d, m)$, we fit datasets with increasing values of $N$ using the cross entropy loss, and plot the largest value of $N$ for which we can obtain at least 99% accuracy. We observe that the linear associative memory can store $\tilde{\Theta}(d^2)$ associations, while the MLP associative memory can store $\tilde{\Theta}(md)$ associations. See Appendix A for additional experiments where the number of output tokens $M$ does not scale with $N$.

## 4    A SYNTHETIC TASK FOR FACTUAL RECALL

In this section we introduce a synthetic factual recall task, and show that one-layer transformers constructed via associative memories can store a number of facts proportional to parameter count.

### 4.1    THE TASK

We first define a global dictionary of facts. Let $\mathcal{S}$ be a set of subject tokens and $\mathcal{R}$ be a set of relation tokens, where $S = |\mathcal{S}|, R = |\mathcal{R}|$. Let $\mathcal{A}$ be the set of answer tokens. We let $a^* : \mathcal{S} \times \mathcal{R} \to \mathcal{A}$ be the ground truth association function, which maps subject-relation tuples $(s, r)$ to their corresponding answer $a^*(s, r)$[1]. A similar such task was considered in Petroni et al. (2019); Ghosal et al. (2024).

Define $\mathcal{A}_r$ to be the set of answers corresponding to a relation $r$, i.e $\mathcal{A}_r := \{a^*(s, r) : s \in \mathcal{S}\}$. Define $\mathcal{A}_s := \{a^*(s, r) : r \in \mathcal{R}\}$ analogously. We assume that each relation corresponds to a disjoint set of answers:

**Assumption 1.** $\mathcal{A}_r \cap \mathcal{A}_{r'} = \emptyset$ for $r, r' \in \mathcal{R}$ with $r \neq r'$. Furthermore, define $D := \max_{r \in \mathcal{R}} |\mathcal{A}_r|$.

For example, $\mathcal{S}$ could be the set of all countries, while $\mathcal{R}$ could be $\{\text{president}, \text{capital}\}$; in this case, the set of all presidents and set of all capitals are disjoint.

---

[1]We focus on one-to-one relations, where each $(s, r)$ pair corresponds to a unique $a^*$. This is in contrast to the one-to-many setting, where each $(s, r)$ maps to many possible answers (for example, $s = $ "France," $r = $ "city," $a^* \in \{$"Paris", "Toulouse", $\cdots\})$

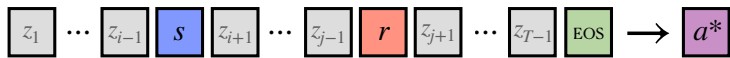

Figure 2: A diagram of the synthetic factual recall task.

We next define our data distribution $\mathcal{D}$ over sequences. Let $T > 0$ be the context length. Let $\mathcal{N}$ be a set of noise tokens, and define the vocabulary to be $\mathcal{V} := \mathcal{S} \cup \mathcal{R} \cup \mathcal{A} \cup \mathcal{N} \cup \{\text{EOS}\}$, where EOS is a special "end-of-sequence" token. The data distribution is over length $T + 1$ sequences $z_{1:T+1} := (z_1, z_2, \ldots, z_T, z_{T+1}) \in \mathcal{V}^{T+1}$, generated via the following procedure:

1. First, sample a subject and relation tuple $(s, r)$ from some distribution $p$ over $\mathcal{S} \times \mathcal{R}$.
2. Next, sample two distinct indices $i, j \in [T-1]$. Set $z_i = s$ and $z_j = r$.
3. For the remainder of tokens $z_k$ where $k \in [T-1] \setminus \{i, j\}$, draw $z_k$ uniformly at random from the noise tokens $\mathcal{N}$.
4. Set $z_T = \text{EOS}$.
5. Finally, set $z_{T+1} = a^*(s, r)$.

The goal of this task is to predict $z_{T+1}$ from $(z_1, \ldots, z_T)$. A model which can successfully do so must (1) be able to isolate the relevant subject and relation tokens from the noise tokens and (2) store all of the associations $(s, r) \mapsto a^*(s, r)$. See Figure 2 for a diagram of the task.

### 4.2 THE MODEL: ONE-LAYER TRANSFORMER

Our learner for the task is a single layer of multi-head self attention followed by an MLP. Define $d$ to be the embedding dimension. The input to the transformer is a sequence of vectors $\boldsymbol{X} := (\boldsymbol{x}_1, \ldots, \boldsymbol{x}_T)^\top \in \mathbb{R}^{T \times d}$. Each self attention head is parameterized by the key, query, and value matrices $\boldsymbol{W}_K, \boldsymbol{W}_Q, \boldsymbol{W}_V \in \mathbb{R}^{d_h \times d}$, where $d_h$ is the *head dimension*. The self attention head is then a map $\text{attn}(\,\cdot\,; \boldsymbol{W}_K, \boldsymbol{W}_Q, \boldsymbol{W}_V) : \mathbb{R}^{T \times d} \to \mathbb{R}^{d_h}$, which operates as

$$\text{attn}(\boldsymbol{X}; \boldsymbol{W}_K, \boldsymbol{W}_Q, \boldsymbol{W}_V) = \boldsymbol{W}_V \boldsymbol{X}^\top \mathcal{S}(\boldsymbol{X} \boldsymbol{W}_K^\top \boldsymbol{W}_Q \boldsymbol{x}_T), \tag{4}$$

where $\mathcal{S}(\boldsymbol{z})_i = \frac{\exp(z_i)}{\sum_j \exp(z_j)}$ is the softmax operator.

A multi-head self-attention layer with $H$ heads is parameterized by $H$ different key, query, and value matrices, along with $H$ output matrices. Let $\boldsymbol{\theta} := \{(\boldsymbol{W}_K^{(h)}, \boldsymbol{W}_Q^{(h)}, \boldsymbol{W}_V^{(h)}, \boldsymbol{W}_O^{(h)})\}_{h \in [H]}$, where $\boldsymbol{W}_K^{(h)}, \boldsymbol{W}_Q^{(h)}, \boldsymbol{W}_V^{(h)}, \boldsymbol{W}_O^{(h)} \in \mathbb{R}^{d_h \times d}$. A multi-head self-attention layer is then a map $F_{\text{MHSA}}(\,\cdot\,; \boldsymbol{\theta}) : \mathbb{R}^{T \times d} \to \mathbb{R}^d$ given by

$$F_{\text{MHSA}}(\boldsymbol{X}; \boldsymbol{\theta}) = \sum_{h \in [H]} \boldsymbol{W}_O^{(h)^\top} \text{attn}(\boldsymbol{X}; \boldsymbol{W}_K^{(h)}, \boldsymbol{W}_Q^{(h)}, \boldsymbol{W}_V^{(h)}). \tag{5}$$

Finally, a single-layer transformer combines a multi-head self-attention layer with an MLP. Let $m$ be the MLP width. Let $\boldsymbol{V}, \boldsymbol{W} \in \mathbb{R}^{m \times d}$ be the MLP parameters, and define $\boldsymbol{\theta}_{\text{TF}} := \boldsymbol{\theta} \cup \{\boldsymbol{V}, \boldsymbol{W}\}$. Then, a single-layer transformer is the map $F_{\text{TF}}(\,\cdot\,; \boldsymbol{\theta}_{\text{TF}}) : \mathbb{R}^{T \times d} \to \mathbb{R}^d$ given by

$$F_{\text{TF}}(\boldsymbol{X}; \boldsymbol{\theta}_{\text{TF}}) = F_{\text{MHSA}}(\boldsymbol{X}; \boldsymbol{\theta}) + \boldsymbol{V}^\top \sigma(\boldsymbol{W} F_{\text{MHSA}}(\boldsymbol{X}; \boldsymbol{\theta})). \tag{6}$$

A single-layer transformer is parameterized by the tuple of hyperparameters $(d, H, d_h, m)$. The model has $4Hdd_h$ self-attention parameters, and $2md$ MLP parameters.

### 4.3 ONE-LAYER TRANSFORMERS HAVE (ALMOST) LINEAR STORAGE CAPACITY

We next characterize how large a single-layer transformer must be in order to obtain 100% accuracy on the synthetic task. For each token $z \in \mathcal{V}$, sample its embedding vectors $\boldsymbol{\varphi}(z) \in \mathbb{R}^d$ i.i.d uniformly over the sphere of radius 1. An input sequence $(z_1, \ldots, z_T)$ gets embedded as $\boldsymbol{X} = (\boldsymbol{\varphi}(z_1), \ldots, \boldsymbol{\varphi}(z_T))^\top$. We use argmax decoding to predict the next token; that is,

$$\hat{f}(z_{1:T}) = \arg\max_{z \in \mathcal{V}} \boldsymbol{\varphi}(z)^\top F_{\text{TF}}(\boldsymbol{X}; \boldsymbol{\theta}_{\text{TF}}). \tag{7}$$

Our first result is that there exists an attention-only single-layer transformer that obtain 100% accuracy on the factual recall task, as long as the total number of self-attention parameters $4Hdd_h$ scales (up to logarithmic factors) linearly with the dataset size $SR$.

**Theorem 3** (Attention-only, informal). *Assume that $d \geq \tilde{\Omega}(\max(R, D))$ and $Hd_h \geq \tilde{\Omega}(S + R)$. With high probability over the embeddings, there exists a single-layer attention-only transformer $F_{\mathrm{TF}}(\,\cdot\,;\boldsymbol{\theta}_{\mathrm{TF}})$ with embedding dimension $d$, number of heads $H$ and head dimension $d_h$ such that*

$$\mathbb{P}_{z_{1:T+1} \sim \mathcal{D}}\left[\arg\max_{z \in \mathcal{V}} \boldsymbol{\varphi}(z)^\top F_{\mathrm{TF}}(\boldsymbol{X};\boldsymbol{\theta}_{\mathrm{TF}}) = z_{T+1}\right] = 1. \tag{8}$$

We next show that a single-layer transformer with an MLP can obtain 100% accuracy on the factual recall task, if the number of MLP parameters $md$ scales linearly with the dataset size:

**Theorem 4** (Attention + MLP, informal). *Assume that $\sigma$ is a polynomial of sufficiently large degree. Define $C(a) = |\{(s, r) : a^*(s, r) = a\}|$. Let $(d, H, d_h, m)$ satisfy*

$$d \geq \tilde{\Omega}(1) \qquad Hd_h \geq \tilde{\Omega}(S + R) \qquad m \geq \tilde{\Omega}(\max_a C(a)) \qquad md \geq \tilde{\Omega}(SR). \tag{9}$$

*Then with high probability over the embeddings there exists a single-layer transformer $F_{\mathrm{TF}}(\,\cdot\,;\boldsymbol{\theta}_{\mathrm{TF}})$ with embedding dimension $d$, number of heads $H$, head dimension $d_h$, and MLP width $m$ such that*

$$\mathbb{P}_{z_{1:T+1} \sim \mathcal{D}}\left[\arg\max_{z \in \mathcal{V}} \boldsymbol{\varphi}(z)^\top F_{\mathrm{TF}}(\boldsymbol{X};\boldsymbol{\theta}_{\mathrm{TF}}) = z_{T+1}\right] = 1. \tag{10}$$

The proofs of Theorem 3 and Theorem 4 are deferred to Appendix C.

**Remarks.** Theorems 3 and 4 each have two main constraints on the size of the architecture needed to obtain 100% accuracy. First, the quantity $Hd_h$ must be larger than $S + R$. This corresponds to self-attention having sufficient capacity to filter out the tokens in $\mathcal{S} \cup \mathcal{R}$ from the noise tokens $\mathcal{N}$. For the attention-only architecture, we additionally require $d = \tilde{\Omega}(\max(R, D))$. When $R \geq D$, the total number of parameters $Hdd_h$ is (up to logs) at least the total number of facts $SR$. For the MLP construction, the second condition is that the number of MLP parameters, $md$, scales nearly linearly with the number of facts $SR$. As such, as long as either the total number of self-attention parameters or the total number of MLP parameters is large enough, 100% accuracy can be obtained. The single-layer transformer can thus trade off MLP and self-attention parameters while still maintaining perfect accuracy. This phenomenon is reflected in the experiments in Section 4.4. We remark that it is straightforward to extend our construction to the case where we only need to store a size $M$ subset of $\mathcal{S} \times \mathcal{R}$, where the constraints now become $Hdd_h, md = \tilde{\Omega}(M)$.

**Proof Sketch.** Theorems 3 and 4 both utilize the associative memory framework of Section 3. First, the key and query matrices of each self-attention head act as a *denoiser*, selecting the relevant subject and relation tokens in $z_{1:T}$ while ignoring the noise tokens. To the $h$th attention head, we associate a subset $\mathcal{S}^{(h)} \subset \mathcal{S} \cup \mathcal{R}$ of subject and relation tokens. Then, setting

$$\boldsymbol{W}_K^{(h)\top} \boldsymbol{W}_Q^{(h)} \approx \beta \sum_{z \in \mathcal{S}^{(h)}} \boldsymbol{\varphi}(z)\boldsymbol{\varphi}(\mathrm{EOS})^\top \tag{11}$$

for a large constant $\beta$, we see that the $h$th head will only attend to the tokens in the subset $\mathcal{S}^{(h)}$. We remark that since the embeddings are $d$-dimensional, at most $d/\mathrm{poly}\log(d)$ embeddings can be in superposition, and thus we must have $\left|\mathcal{S}^{(h)}\right| \leq d/\mathrm{poly}\log(d)$.

For the attention-only construction, the output-value matrix $\boldsymbol{W}_O^{(h)\top} \boldsymbol{W}_V^{(h)}$ acts as a linear associative memory, mapping each $z$ in $\mathcal{S}^{(h)}$ to a superposition of all possible answers associated with the subject/relation $z$. Letting $P_h$ be a projection onto a random $d_h$-dimensional subspace of $\mathbb{R}^d$, we set

$$\boldsymbol{W}_O^{(h)\top} \boldsymbol{W}_V^{(h)} \propto \sum_{z \in \mathcal{S}^{(h)}} \sum_{a \in \mathcal{A}_z} \boldsymbol{\varphi}(a)\boldsymbol{\varphi}(z)^\top P_h. \tag{12}$$

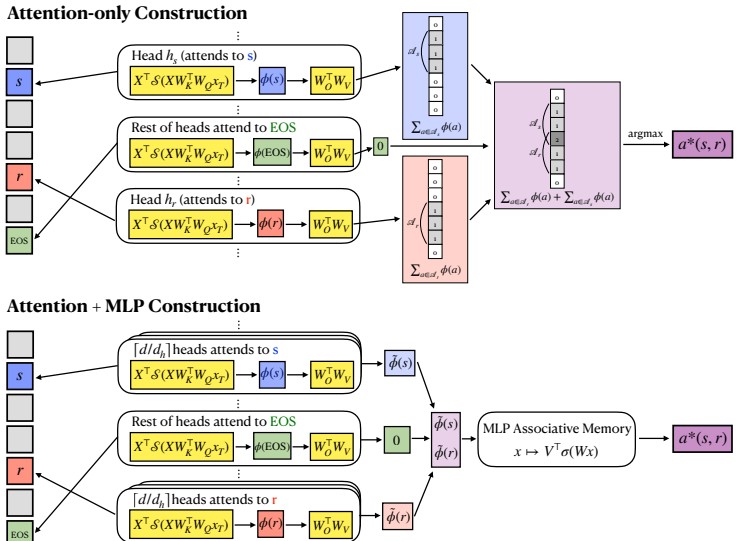

Figure 3: Both the Attention-only and Attention+MLP constructions for the factual recall task.

In Lemma 2, we show that this construction stores at most $d_h$ tokens per head (i.e $\left|\mathcal{S}^{(h)}\right| \lesssim d_h$), and requires the dimension to scale with the number of elements in superposition (i.e $|\mathcal{A}^z| \lesssim d$). Since $|\mathcal{A}^z| \leq R + D$, and the $\mathcal{S}^{(h)}$ partition $\mathcal{S} \cup \mathcal{R}$, it suffices to take $d \gtrsim R + D$ and $Hd_h \gtrsim S + R$.

For the MLP construction, we instead associate the subset $\mathcal{S}^{(h)}$ with $\lceil d/d_h \rceil$ attention heads. This is equivalent to having a single full-rank attention head per subset. We set the aggregate output-value matrix to the identity, so that the output of the self-attention layer is $F_{\text{MHSA}}(\boldsymbol{X}; \boldsymbol{\theta}) = \boldsymbol{\varphi}(s) + \boldsymbol{\varphi}(r)$. Finally, the MLP layer acts as an MLP associative memory, mapping $\boldsymbol{\varphi}(s) + \boldsymbol{\varphi}(r)$ to $\boldsymbol{\varphi}(a^*(s, r))$ for each $(s, r)$ pair. Via a similar computation to Theorem 2, it suffices to make the total number of parameters $md$ be $md = \tilde{\Omega}(SR)$. Since the $\mathcal{S}^{(h)}$ partition $\mathcal{S} \cup \mathcal{R}$, it suffices to take $Hd_h \gtrsim S + R$ as well. See Figure 3 for a diagram describing both constructions.

## 4.4 EMPIRICAL VALIDATION

We next empirically validate the claims of Theorems 3 and 4 that 100% accuracy can be obtained as long as either the total number of self-attention or MLP parameters scales with $SR$. We further observe that 100% accuracy can be achieved as long as the *total* number of parameters scales with $SR$, providing evidence that the model can simultaneously use attention and the MLP to store facts.

In Figure 4, we train a wide range of models of various "shapes" on datasets of varying sizes. A model shape is defined by the tuple $(\alpha, \beta, H)^2$, and corresponds to the family of models satisfying $Hd_h = \alpha d$ and $m = \beta d$. The total number of model parameters is $4Hd_hd + 2md = (4\alpha + 2\beta)d^2$, which can thus be varied by increasing $d$. For a fixed model size $(d, H, d_h, m)$, we binary search on the largest dataset size that can be memorized. Specifically, we fix $D = 8$ and vary $S, R$ jointly as $S = R$. Experiments with different scalings are considered in Appendix A. For each $(S, R, D)$, the fact dataset is generated at random by selecting $|\mathcal{A}| = RD$, $|\mathcal{N}| = S + R$, and for each $s$ sampling $a^*(s, r)$ uniformly at random from $\{(r-1)D + 1, \ldots, rD\}$. We say the dataset was successfully memorized, and as such $SR$ facts were stored, if the model can obtain an accuracy of at least 99%.

On the left panel of Figure 4 we observe that, across different model shapes, the maximum number of facts stored scales linearly with the total number of parameters. On the right panel, we consider a specific dataset with $S = 32, R = 32, D = 8$, and plot the accuracy as the number of parameters vary. We observe that the model can trade off MLP parameters for self-attention parameters, while still maintaining an accuracy of near 1. However, we do still require the total number of attention parameters to be large enough; this corresponds to the $Hd_h = \tilde{\Omega}(S + R)$ constraint.

---

[2]The "standard" transformer scaling takes $\alpha = 1, \beta = 4$.

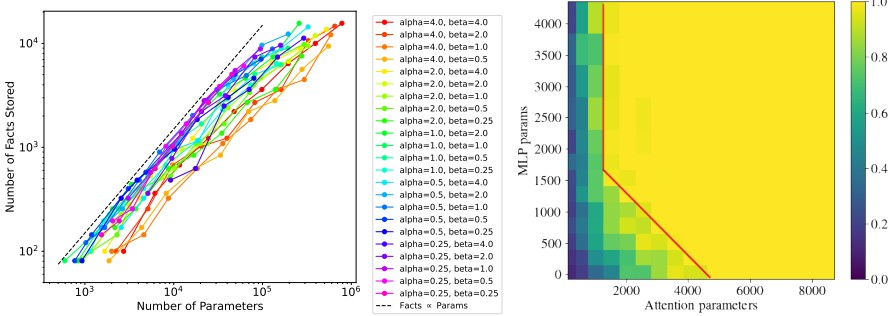

Figure 4: (Left) The number of facts stored scales linearly with the total number of parameters, for a wide range of model sizes. (Right) For a fixed dataset, the model can trade off MLP parameters for attention parameters to obtain 100% accuracy. The heatmap color corresponds to model accuracy.

## 5 OPTIMIZATION DYNAMICS

We next study the optimization dynamics of the factual recall task. To simplify the model, we consider a linear attention transformer (i.e., the softmax is replaced with the identity map) with orthogonal embeddings. We set $d = |\mathcal{V}|$, and let the embedding vectors $\{\varphi(z)\}_{z \in \mathcal{V}}$ satisfy $\langle \varphi(z), \varphi(z') \rangle = \delta_{z=z'}$. Such linear attention or orthogonal embeddings assumptions are common in prior works studying the gradient descent dynamics of transformers (Li et al., 2023; Von Oswald et al., 2023; Ahn et al., 2024; Mahankali et al., 2024; Zhang et al., 2024; Nichani et al., 2024).

The linear attention model is given by

$$F_{\text{lin}}(\boldsymbol{X}; \boldsymbol{\theta}) := \boldsymbol{W}_{OV} \boldsymbol{X}^\top \boldsymbol{X} \boldsymbol{W}_{KQ} \boldsymbol{x}_T, \tag{13}$$

where we set $d_h = d$ and let $\boldsymbol{W}_{OV} := \boldsymbol{W}_O^\top \boldsymbol{W}_V$, $\boldsymbol{W}_{KQ} := \boldsymbol{W}_K^\top \boldsymbol{W}_Q$ denote the non-factorized output-value and key-query matrices. Let $\hat{p}(\cdot \mid z_{1:T}) \in \Delta_{\mathcal{A}}$ be the predicted next token distribution on an input sequence $z_{1:T}$, i.e

$$\hat{p}(a \mid z_{1:T}) := \frac{\exp\left(\langle \varphi(a), F_{\text{lin}}(\boldsymbol{X}; \boldsymbol{\theta}) \rangle\right)}{\sum_{a' \in \mathcal{A}} \exp\left(\langle \varphi(a'), F_{\text{lin}}(\boldsymbol{X}; \boldsymbol{\theta}) \rangle\right)}. \tag{14}$$

One can then rewrite the cross entropy loss as

$$L(\boldsymbol{\theta}) = \mathbb{E}_{z_{1:T+1}}[-\log \hat{p}(z_{T+1} \mid z_{1:T})]. \tag{15}$$

We would like to characterize the output of running gradient flow, (i.e $\dot{\boldsymbol{\theta}} = -\nabla L(\boldsymbol{\theta})$) with respect to the non-factorized parameters $\boldsymbol{\theta} := \{\boldsymbol{W}_{OV}, \boldsymbol{W}_{KQ}\}$ on the cross-entropy loss (15). For notational convenience, we denote $\boldsymbol{W}_{OV}(a, z) := \varphi(a)^\top \boldsymbol{W}_{OV} \varphi(z)$, $\boldsymbol{W}_{KQ}(z) := \varphi(z)^\top \boldsymbol{W}_{KQ} \varphi(\text{EOS})$, and note that by isometry gradient flow on $\boldsymbol{\theta}$ is equivalent to gradient flow on these quantities.

Let us assume that we start from the following "balanced" initialization.

**Assumption 2.** *Given an initialization scale* $\alpha > 0$*, set* $\boldsymbol{W}_{OV}(a, z) = \alpha$ *and* $\boldsymbol{W}_{KQ}(z) = \alpha\sqrt{|\mathcal{A}| + 1}$ *for each* $a \in \mathcal{A}, z \in \mathcal{V}$.

Our first result is that the gradient flow indeed converges to zero loss. As a consequence, the predicted next token probabilities $\hat{p}(z_{T+1} \mid z_{1:T})$ converge to $\mathbf{1}(z_{T+1} = a^*(s, r))$, where $s, r$ are the subject and relation contained in the sequnece $z_{1:T}$.

**Theorem 5** (Global Convergence). *For* $t \geq 0$*, let* $\boldsymbol{\theta}(t)$ *be the output of running gradient flow for* $t$ *time. For any* $\delta > 0$*, there exists a time* $t_\delta$ *such that for* $t \geq t_\delta$*,* $L(\boldsymbol{\theta}(t)) \leq \delta$.

We next show that the model undergoes a sequential learning dynamics. Let us assume that the number of subjects $S$ is much greater than the number of facts $R$. We show that during the first stage of training only the $\boldsymbol{W}_{OV}(a, r)$ and $\boldsymbol{W}_{KQ}(r)$ components grow for relations $r \in \mathcal{R}$, while the remainder of the parameters stay close to zero. As such, the model gets close to outputting the best predictor based on just the relation token $r$. Define $p^*(\cdot \mid r)$ to be the conditional distribution of the answer, given the relation $r$, i.e $p^*(a \mid r) := \sum_{s \in \mathcal{S}} p(s \mid r)\mathbf{1}(a = a^*(s, r))$.

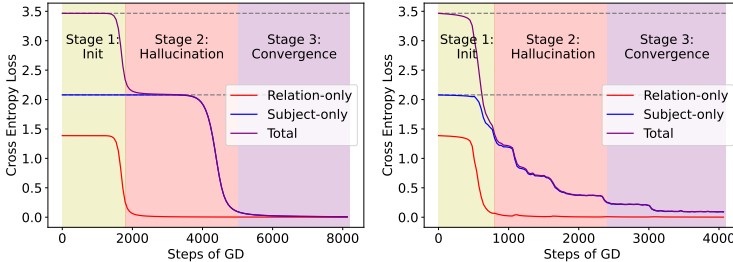

Figure 5: (Left) Loss of the linear attention model with orthogonal embeddings. There is an intermediate *hallucination* stage where the loss plateaus and the model predicts based on only the relation. (Right) Loss of the softmax attention model with random embeddings. We again observe an intermediate hallucination stage, where the relation-only loss is zero but the total loss is still large.

**Theorem 6** (Sequential Learning). *Assume that $S \geq 8R\sqrt{2D}$, and $|\mathcal{N}| \geq 4R\sqrt{2D}T$. Let $p(s, r) = \frac{1}{SR}$. Pick $\epsilon > 0$. There exists runtime $T^*$ and initialization scale $\alpha$ (both depending on $\epsilon$) such that:*

1. *For all $t \leq T^*$ and $z \in \mathcal{S} \cup \mathcal{N}, a \in \mathcal{A}$, we have $|\boldsymbol{W}_{OV}(a, z)|, |\boldsymbol{W}_{KQ}(z)| \leq \alpha^{1/2}$*

2. *There exists $t \leq T^*$ such that, for any input sequence $z_{1:T}$ containing a relation $r$, $\sum_{a \in \mathcal{A}} \left( p^*(a \mid r) - \hat{p}(a \mid z_{1:T}) \right)^2 \leq \epsilon^2.$*

Proofs of Theorems 5 and 6 are deferred to Appendix D.

**Remarks.**  Theorem 6 tells us that at some intermediate time, the prediction of the model $\hat{p}(\cdot \mid z_{1:T})$ is approximately equal to $p^*(\cdot \mid r)$, the conditional distribution of the answer given the relation $r$. At this stage, the model ignores all other tokens in the sequence $z_{1:T}$ – including the useful subject token $s$ – and predicts based only on the relation $r$. For example, if $\mathcal{S}$ is the set of all countries and $r$ is the relation "capital," then on the prompt "What is the capital of France?" the model will output a random countries' capital. We view this as an instance of *hallucination*: the model is outputting a plausible, yet ultimately incorrect, answer to the prompt. We remark that without the assumption that $S \gg R$, it is possible for this intermediate hallucination stage to exhibit different behavior.

**Empirical Validation.**  We next empirically verify Theorems 5 and 6. We first train the linear attention model with orthogonal embeddings (15) with $S = 16, R = 4$ and $D = 8$, and plot the loss over time. In the left pane of Figure 5, we observe three distinct stages. At the start of training, the prediction is close to uniform over all possible answers, and the model obtains a loss of $\log |\mathcal{A}|$. Next, the loss plateaus at $\log D$, and the model outputs the conditional distribution of $a$ given the relation $r$. Finally, as training continues, the model escapes the plateau and converges to zero loss. We include the "relation-only loss" in the plot, defined as $\mathbb{E}_{z_{1:T+1}} \left[ -\log \left( \sum_{a \in \mathcal{A}_r} p(a \mid z_{1:T}) \right) \right]$, where any probability mass assigned to an answer which is valid for the relation $r$ is considered to be correct; the subject-only loss is defined analogously.

In the right pane of Figure 5, we plot the loss of a single softmax attention head with random embeddings trained on the same factual recall task. We observe similar phenomenology as for linear attention, and identify an intermediate "hallucination" stage where the relation-only loss drops to zero, but the subject-only loss is still far from zero.

## 6   LOWER BOUNDS

In this section, we argue via information-theoretic arguments that the results from Sections 3 and 4 are optimal up to logarithmic factors. Proofs are deferred to Appendix E.

**Associative Memories.**  Let $[N]$ and $[M]$ be the input and output vocabularies, respectively. To establish a lower bound, we must consider a *distribution* over association functions $f^*$. For each $x \in [N]$, assume that the output $f^*(x)$ is sampled independently from the uniform distribution over $[M]$. We model the learning protocol as follows. At train time, the learner observes the randomly

sampled ground truth $f^*$, and writes down a $B$ bit model $\boldsymbol{F}$. At test time, the learner generates a set of predictions $\hat{f} \in \mathbb{R}^{N \times M}$ from $\boldsymbol{F}$, where $\hat{f}(x) \in \Delta_M$ is the prediction for $f^*(x)$. Both the mappings $f^* \to \boldsymbol{F}$ and $\boldsymbol{F} \to \hat{f}$ can be randomized. Let $p$ be a probability distribution over the input space $[N]$; assume WLOG that $p(1) \geq p(2) \geq \cdots \geq p(N)$. The goal of the learner is to minimize the cross entropy loss $L(\hat{f}) = -\sum_{x \in [N]} p(x) \log \hat{f}(x)_{f^*(x)}$.

**Theorem 7.** *The expected loss of the learner can be lower bounded by*

$$\mathbb{E}_{f^*, \hat{f}}\left[L(\hat{f})\right] \geq \log M \cdot \sum_{x \geq \lceil \frac{B}{\log M} \rceil} p(x). \tag{16}$$

We thus see that in order to obtain zero loss, the learner must use $B \geq N \log M$ bits; this matches the construction from Theorem 2 up to log factors. As a corollary of Theorem 7, we can obtain scaling law lower bounds with respect to model size.

**Corollary 1.** *Assume that $p$ is a power law, i.e $p(x) \propto x^{-\alpha}$ for $\alpha > 1$. Then $\mathbb{E}_{f^*, \hat{f}}\left[L(\hat{f})\right] \gtrsim B^{1-\alpha}$.*

This lower bound is obtained by the MLP associative memory by storing the most probable $\tilde{O}(B)$ associations. This matches the scaling law with respect to model size considered in Michaud et al. (2023); Cabannes et al. (2024), which also considered storing the $\tilde{O}(B)$ most frequent associations.

**Remark.** The constructions in Section 3 require storing $\tilde{O}(N)$ network parameters, along with input and output embeddings. We view $\boldsymbol{F}$ in Theorem 7 as containing only the network parameters, while the embeddings are "global" quantities, independent of the ground truth $f^*$, used to compute the predictions $\hat{f}$. This matches our interpretation of the embeddings as fixed global quantities which cannot be modified by the associative memory. We remark that the associative memory constructions from Section 3 match the lower bound, since they hold for $\tilde{O}(1)$-bit precision (Corollary 2).

**Factual Recall.** We next prove a lower bound for the factual recall task; a similar bound was proven in Allen-Zhu & Li (2024). Let $\mathcal{S}$ and $\mathcal{R}$ be the fixed set of subjects and relations and $\mathcal{V}$ be the full vocabulary, where $|\mathcal{V}| \gg |\mathcal{S}|, |\mathcal{R}|$. The association function $a^* : \mathcal{S} \times \mathcal{R} \to \mathcal{V}$ is sampled randomly as follows. First, for each relation $r \in \mathcal{R}$, the answer set $\mathcal{A}_r$ is chosen to be a uniformly random size $D$ subset of $\mathcal{V}$, conditional on all subsets $\mathcal{A}_r$ being disjoint. For each $s \in \mathcal{S}$, the answer $a^*(s, r)$ is sampled uniformly at random from $\mathcal{A}_r$. The learner sees the association $a^*$, writes down a $B$ bit model $\boldsymbol{F}$, and from $\boldsymbol{F}$ generates a set of predictions $\hat{f} \in \mathbb{R}^{S \times R \times |\mathcal{V}|}$, where $\hat{f}(s, r) \in \Delta_{\mathcal{V}}$ is the prediction for $a^*(s, r)$. We lower bound $\boldsymbol{L}$, the expected cross entropy loss with respect to a distribution $p(s, r)$ over $\mathcal{S} \times \mathcal{R}$, defined as follows:

$$\boldsymbol{L} := \mathbb{E}_{a^*, \hat{f}}\left[L(\hat{f})\right] = \mathbb{E}_{a^*, \hat{f}}\left[-\sum_{s,r} p(s, r) \log \hat{f}(s, r)_{a^*(s, r)}\right]. \tag{17}$$

**Theorem 8.** *Assume that $|\mathcal{V}| \geq 2RD$ and $S \geq CD \log\left(2D^2 \log |V|\right)$ for sufficiently large constant $C$. There exists a constant $c \in (0, 1)$ such that, if $\boldsymbol{L} = 0$, the number of bits $B$ must satisfy*

$$B \geq SR \log D + (1 - c)RD \log\left(|\mathcal{V}|/D\right) \tag{18}$$

We thus see that $\tilde{O}(SR)$ parameters are needed to achieve a loss of zero. For this lower bound, the learner knows the sets $\mathcal{S}$ and $\mathcal{R}$ and does not have to distinguish them from the noise tokens $\mathcal{N}$, making it a strictly easier problem than the factual recall task in Section 4.

# 7 DISCUSSION

In this work, we showed that shallow transformers can use associative memories to obtain near optimal storage capacity for factual recall tasks. Furthermore, by studying the optimization dynamics of a simplified model, we also showed that transformers undergo an intermediate hallucination stage. One interesting direction of future work is to better understand the role of the embeddings, and whether there exists an optimal choice of (non-random) embeddings leading to more efficient constructions. Another important direction is to understand the implication of our results towards understanding empirical LLM scaling laws (Hoffmann et al., 2022). In particular, does there exist a scaling law lower bound for the factual recall task? Finally, it would be very interesting to understand the extent to which larger models utilize similar associative memory constructions, and if one can probe whether specific "facts" are stored in either the self-attention matrices or the MLP.

## ACKNOWLEDGEMENTS

This work was conducted while EN was an intern at the Flatiron Institute. JDL acknowledges support of NSF CCF 2002272, NSF IIS 2107304, NSF CIF 2212262, ONR Young Investigator Award, and NSF CAREER Award 2144994. This work was conducted in part at the Simons Institute.

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

# A   ADDITIONAL EXPERIMENTS

## A.1   MLP ASSOCIATIVE MEMORY

In Figure 6, we train MLP associative memories to store the association $f^*(x) = x \mod M$. We fix $M = 32$ throughout. In this case, we see that the number of associations $N$ which can be stored by a model with $md$ parameters scales as $N \propto md$. This linear scaling in the absence of logarithmic factors is due to the fact that the number of output tokens $M$ is a constant, and does not scale with the number of input tokens $N$.

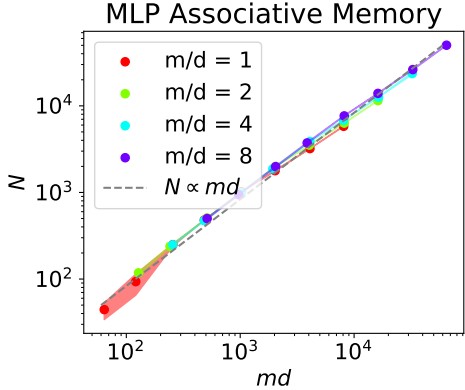

Figure 6: We train an MLP associative memory to store the association $f^*(x) = x \mod 32$. Empirically, $md \propto N$ parameters are required to store $N$ associations.

## A.2   FACTUAL RECALL

In Figures 7 to 9, we repeat the experiment in the left pane of Figure 4, for different choices of $H$ and scalings of $(S, R, D)$. In all plots, we observe the general trend that the number of facts stored scales proportionally to the number of model parameters.

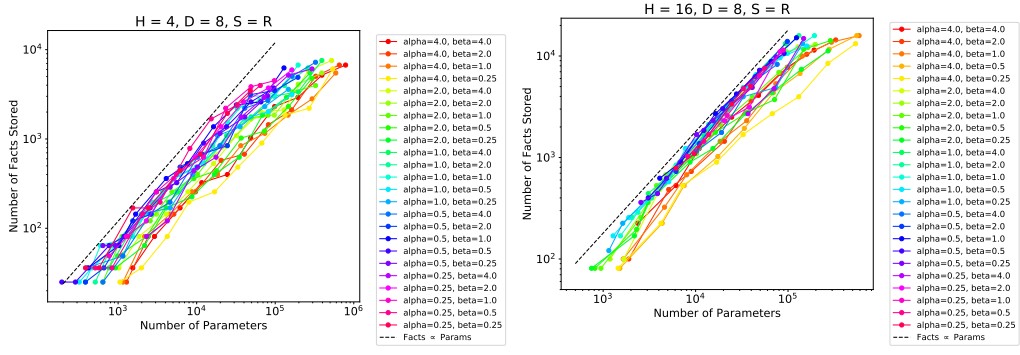

Figure 7: We repeat the experiment in Figure 4, varying the number of head to be 4 (Left) or 16 (Right). In both cases, we observe that the number of facts stored scales linear with the parameter count.

## A.3   EXPERIMENTAL DETAILS

**Figures 1, 6:**   For the MLP associative memory experiments, for each choice of $m, d, N$, we first sample random embeddings $\{e_x\}_{x \in [N]}, \{u_y\}_{y \in [M]}$ i.i.d uniformly over the sphere. We train a two-layer neural network on the cross entropy loss to predict the association $f^*(x) = x$. We use standard parameterization and initialization, and the activation $\sigma = \text{ReLU}$. The network is trained using ADAM with a learning rate of $10^{-2}$ for $2^{14}$ steps. We compute the maximum accuracy the network

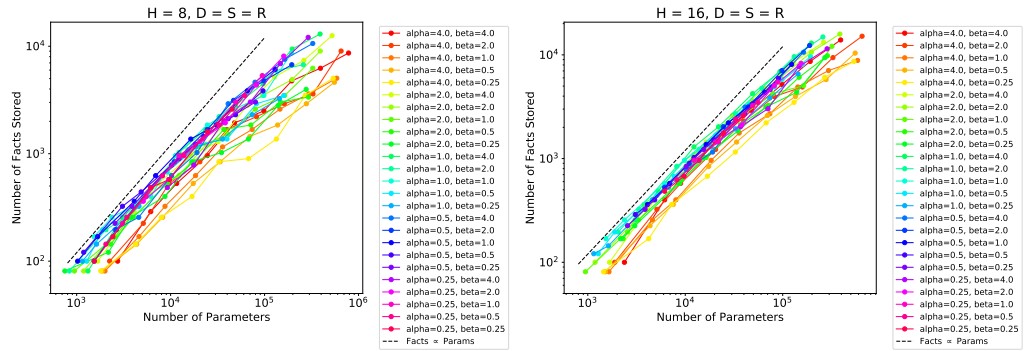

Figure 8: We repeat Figure 4 on factual recall tasks where each subject and relation map to a distinct answer (i.e $D = S$).

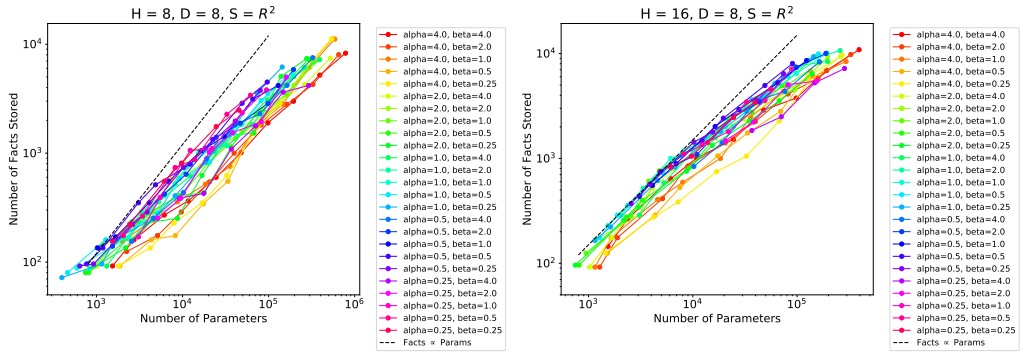

Figure 9: We repeat Figure 4 on factual recall tasks where the number of subjects is much larger than the number of relations; specifically, we take $S = R^2$.

achieves over the training run, and say that the network has "stored" the dataset if the highest accuracy is at least 99%. We repeat this procedure to binary search over $N$, to find the largest value of $N$ such that the network achieves an accuracy of at least 99%. Error bars are shown for 5 random seeds.

**Figures 4, 7, 8, 9:** We consider a fixed prompt length of $T = 32$, and train the models via online batch gradient descent with batch size 1024 on the population loss (i.e we sample an independent batch at each timestep). We use standard parameterization and initialization for both self-attention and the MLP. For a fixed model size, we binary search over the maximum value of $SR$ such that the model achieves an accuracy of at least 99%. All models were trained using ADAM for $2^{14}$ steps, with a sweep over learning rates in $\{.001, .003, .01\}$ (where we consider the best performing model over all learning rates).

**Figure 5:** In the left pane we train a linear attention head with orthogonal embeddings. The weights are all initialized to be equal to $10^{-5}$. In the right plot, we train a softmax attention head with random embeddings, which are fixed throughout training.

Code for all the experiments can be found at `https://github.com/eshnich/factual-recall-iclr`.

## B    PROOFS FOR SECTION 3

*Proof of Theorem 1.* Let us set $\boldsymbol{W} = \sum_{z \in [N]} \boldsymbol{u}_{f^*(z)} \boldsymbol{e}_z^\top$. For $y \neq f^*(x)$, define the quantity $\gamma_{xy}$ by

$$\gamma_{xy} = (\boldsymbol{u}_{f^*(x)} - \boldsymbol{u}_y)^\top \boldsymbol{W} \boldsymbol{e}_x.$$

We first see that (where the expectation is taken over the randomness of the embedding vectors)

$$
\begin{aligned}
\mathbb{E}[\gamma_{xy}] &= \sum_{z \in [N]} \mathbb{E}\big[(\boldsymbol{u}_{f^*(x)} - \boldsymbol{u}_y)^\top \boldsymbol{u}_{f^*(z)}\big]\mathbb{E}\big[\boldsymbol{e}_z^\top \boldsymbol{e}_x\big] \\
&= \mathbb{E}\big[(\boldsymbol{u}_{f^*(x)} - \boldsymbol{u}_y)^\top \boldsymbol{u}_{f^*(x)}\big] \\
&= 1.
\end{aligned}
$$

We can next compute the second moment of $\gamma_{xy}$. Since the $\boldsymbol{e}_z$ are drawn uniformly on the sphere, the $\boldsymbol{e}_z^\top \boldsymbol{e}_x$ terms for $z \neq x$ are independent and mean zero. Therefore

$$
\begin{aligned}
\mathbb{E}[\gamma_{xy}^2] &= \sum_{z \in [N]} \mathbb{E}\Big[\big((\boldsymbol{u}_{f^*(x)} - \boldsymbol{u}_y)^\top \boldsymbol{u}_{f^*(z)}\big)^2\Big]\mathbb{E}\Big[\big(\boldsymbol{e}_z^\top \boldsymbol{e}_x\big)^2\Big] \\
&= \mathbb{E}\Big[\big(1 - \boldsymbol{u}_{f^*(x)}^\top \boldsymbol{u}_y\big)^2\Big] + \frac{1}{d}\sum_{z \neq x}\mathbb{E}\Big[\big((\boldsymbol{u}_{f^*(x)} - \boldsymbol{u}_y)^\top \boldsymbol{u}_{f^*(z)}\big)^2\Big] \\
&= 1 + \frac{1}{d} + \frac{1}{d}\sum_{z \neq x}\left(\frac{2}{d}\cdot\mathbf{1}(f^*(z) \neq y) + \left(1 + \frac{1}{d}\right)\cdot\mathbf{1}(f^*(z) = y)\right) \\
&= \left(1 + \frac{1}{d}\right)^2 + \frac{2(N-2)}{d^2}.
\end{aligned}
$$

Therefore

$$
\mathrm{Var}(\gamma_{xy}) = \frac{2}{d} + \frac{1}{d^2} + \frac{2(N-2)}{d^2} \lesssim \frac{1}{d} + \frac{N}{d^2}.
$$

Let $\delta$ be a fixed failure probability, and let $\delta' = \frac{\delta}{NM}$. Observe that $\gamma_{xy}$ is a degree 4 polynomial. By Lemma 14, by choosing $d \geq C\log^4(1/\delta')$ and $d^2 \geq CN\log^4(1/\delta')$ for a sufficiently large constant $C$, we have that $\frac{16e^{-1}\log^4(1/\delta)\mathrm{Var}(\gamma_{xy})}{(\mathbb{E}\gamma_{xy})^2} \leq 1$, and thus $\mathbb{P}(\gamma_{xy} \leq 0) \leq \delta'$.

Therefore union bounding over all $(x, y)$ pairs with $y \neq f^*(x)$, we have that

$$
\mathbb{P}(\exists\, \gamma_{xy} \leq 0) \leq N(M-1)\delta' \leq \delta.
$$

Thus with probability $1 - \delta$, $\gamma_{xy} > 0$ for all $x$ and $y \neq f^*(x)$, and on this event $\arg\max_{y \in [M]} \boldsymbol{u}_y^\top \boldsymbol{W}\boldsymbol{e}_x = f^*(x)$ for all $x \in [N]$.

$\square$

For Theorem 2, we need the following assumption on the activation $\sigma$.

**Assumption 3.** *$\sigma$ is a polynomial of degree $q$. Furthermore, if $\sigma(z) = \sum_{k=0}^{q} c_k h_k(z)$ is the Hermite decomposition of $\sigma$, then $c_k \neq 0$ for all $0 \leq k \leq q$.*

We prove the following formal version of Theorem 2.

**Theorem 9.** *Let $\epsilon \in (0, 1)$ be a fixed constant. Assume that $d \geq N^\epsilon$ and $N \geq C_1(\epsilon)$, where $C_1(\epsilon)$ is a constant depending only on $\epsilon$. Assume that $q$ in Assumption 3 satisfies $q = \frac{C_2}{\epsilon}$ for some $C_2 > 2$. Then, if $md \gtrsim N(C_3\log(MN/\delta))^{C_4/\epsilon}$, with probability $1 - \delta$ over the draw of the embeddings, there exists $\boldsymbol{V}, \boldsymbol{W}$ such that*

$$
\arg\max_{y \in [M]} \boldsymbol{u}_y^\top \boldsymbol{V}^\top \sigma(\boldsymbol{W}\boldsymbol{e}_x) = f^*(x) \tag{19}
$$

*for all $x \in [N]$.*

*Proof of Theorem 2.* Let us consider the linearization, or Neural Tangent Kernel, of $F$:

$$
F_{\mathrm{NTK}}(\boldsymbol{z}) = \boldsymbol{V}^\top\big(\sigma'(\boldsymbol{W}^0\boldsymbol{z}) \odot (\boldsymbol{W} - \boldsymbol{W}^0)\boldsymbol{z}\big) = \sum_{i \in [m]} \boldsymbol{v}_i\sigma'(\langle \boldsymbol{w}_i^0, \boldsymbol{z}\rangle)\langle \boldsymbol{w}_i - \boldsymbol{w}_i^0, \boldsymbol{z}\rangle.
$$

where $\boldsymbol{W}^0$ is the initialization which we are linearizing with respect to. By rescaling the parameters as $\boldsymbol{W} - \boldsymbol{W}^0 \leftarrow \epsilon(\boldsymbol{W} - \boldsymbol{W}^0)$ and $\boldsymbol{V} \leftarrow \epsilon^{-1}\boldsymbol{V}$, we see that $F \to F_{\mathrm{NTK}}$ as $\epsilon \to 0$. It thus suffices to work with $F_{\mathrm{NTK}}$ instead of $F$. For ease of notation, we redefine $\boldsymbol{W}^0$ as $\boldsymbol{W}$, and $\boldsymbol{W} - \boldsymbol{W}^0$ as $\boldsymbol{Q}$, so that

$$F(\boldsymbol{z}) = \sum_{i \in [m]} \boldsymbol{v}_i \sigma'(\langle \boldsymbol{w}_i, \boldsymbol{z}\rangle)\langle \boldsymbol{q}_i, \boldsymbol{z}\rangle$$

Let $k$ be a even integer, to be chosen later. Assume without loss of generality that $c_{k+1} > 0$ (if it is negative, we can simply negate all the $\boldsymbol{q}_i$ in the construction below). Set

$$\boldsymbol{q}_i = \frac{1}{m} \sum_{z \in [N]} h_k(\langle \boldsymbol{e}_z, \boldsymbol{w}_i\rangle)\langle \boldsymbol{v}_i, \boldsymbol{u}_{f^*(z)}\rangle \boldsymbol{e}_z,$$

where $h_k$ is the $k$th Hermite polynomial. Then

$$F(\boldsymbol{e}_x) = \frac{1}{m} \sum_{i \in [m], z \in [N]} \boldsymbol{v}_i \langle \boldsymbol{v}_i, \boldsymbol{u}_{f^*(z)}\rangle \sigma'(\langle \boldsymbol{w}_i, \boldsymbol{e}_x\rangle)h_k(\langle \boldsymbol{e}_z, \boldsymbol{w}_i\rangle)\langle \boldsymbol{e}_x, \boldsymbol{e}_z\rangle$$

As in the proof of Theorem 1, define the margin between $x$ and some $y \neq f^*(x)$ as

$$\begin{aligned}
\gamma_{xy} &= (\boldsymbol{u}_{f^*(x)} - \boldsymbol{u}_y)^\top F(\boldsymbol{e}_x) \\
&= \frac{1}{m} \sum_{i \in [m], z \in [N]} \langle \boldsymbol{v}_i, \boldsymbol{u}_{f^*(x)} - \boldsymbol{u}_y\rangle\langle \boldsymbol{v}_i, \boldsymbol{u}_{f^*(z)}\rangle \sigma'(\langle \boldsymbol{w}_i, \boldsymbol{e}_x\rangle)h_k(\langle \boldsymbol{e}_z, \boldsymbol{w}_i\rangle)\langle \boldsymbol{e}_x, \boldsymbol{e}_z\rangle.
\end{aligned}$$

We will show that, with high probability over the draw of the embeddings over the sphere, *and* the $\boldsymbol{v}_i, \boldsymbol{w}_i$ independently from the standard Gaussian, that $\gamma_{xy} > 0$ for all $y \neq f^*(x)$.

The expectation of the margin is

$$\begin{aligned}
\mathbb{E}[\gamma_{xy}] &= \sum_z \mathbb{E}\big[\langle \boldsymbol{v}_i, \boldsymbol{u}_{f^*(x)} - \boldsymbol{u}_y\rangle\langle \boldsymbol{v}_i, \boldsymbol{u}_{f^*(z)}\rangle \sigma'(\langle \boldsymbol{w}_i, \boldsymbol{e}_x\rangle)h_k(\langle \boldsymbol{e}_z, \boldsymbol{w}_i\rangle)\langle \boldsymbol{e}_x, \boldsymbol{e}_z\rangle\big] \\
&= c_{k+1} \sum_z \mathbb{E}\big[\langle \boldsymbol{u}_{f^*(x)} - \boldsymbol{u}_y, \boldsymbol{u}_{f^*(z)}\rangle\langle \boldsymbol{e}_z, \boldsymbol{e}_x\rangle^{k+1}\big] \\
&= c_{k+1}.
\end{aligned}$$

We next compute the variance. Define $\omega_{iz}^{xy}$ as

$$\omega_{iz}^{xy} = \langle \boldsymbol{v}_i, \boldsymbol{u}_{f^*(x)} - \boldsymbol{u}_y\rangle\langle \boldsymbol{v}_i, \boldsymbol{u}_{f^*(z)}\rangle \sigma'(\langle \boldsymbol{w}_i, \boldsymbol{e}_x\rangle)h_k(\langle \boldsymbol{e}_z, \boldsymbol{w}_i\rangle)\langle \boldsymbol{e}_x, \boldsymbol{e}_z\rangle,$$

so that $\gamma_{xy} = \frac{1}{m}\sum_{i,z} \omega_{iz}^{xy}$. First, observe that when $z \neq z'$, we have $\mathbb{E}\left[\omega_{iz}^{xy}\omega_{jz'}^{xy}\right] = 0$, since $k$ is even. For $i \neq j$, we have that

$$\mathbb{E}\big[\omega_{iz}^{xy}\omega_{jz}^{xy}\big] = c_{k+1}^2 \mathbb{E}\big[\langle \boldsymbol{u}_{f^*(x)} - \boldsymbol{u}_y, \boldsymbol{u}_{f^*(z)}\rangle^2\big]\mathbb{E}[\langle \boldsymbol{e}_z, \boldsymbol{e}_x\rangle^{2(k+1)}]$$

First, by Lemma 12 we have that

$$\mathbb{E}[\langle \boldsymbol{e}_z, \boldsymbol{e}_x\rangle^{2(k+1)}] \leq \begin{cases} 1 & x = z \\ (2k+2)^{k+1}d^{-(k+1)} & x \neq z \end{cases}.$$

Next, we see that

$$\mathbb{E}\big[\langle \boldsymbol{u}_{f^*(x)} - \boldsymbol{u}_y, \boldsymbol{u}_{f^*(z)}\rangle^2\big] = \begin{cases} 1 + \frac{1}{d} & f^*(x) = f^*(z) \text{ or } y = f^*(z) \\ \frac{2}{d} & \text{otherwise} \end{cases}.$$

Finally, we have that

$$\begin{aligned}
\mathbb{E}[\omega_{iz}^{xy}\omega_{iz}^{xy}] &= \mathbb{E}\big[\langle \boldsymbol{v}_i, \boldsymbol{u}_{f^*(x)} - \boldsymbol{u}_y\rangle^2\langle \boldsymbol{v}_i, \boldsymbol{u}_{f^*(z)}\rangle^2 \sigma'(\langle \boldsymbol{w}_i, \boldsymbol{e}_x\rangle)^2 h_k(\langle \boldsymbol{e}_z, \boldsymbol{w}_i\rangle)^2\langle \boldsymbol{e}_x, \boldsymbol{e}_z\rangle^2\big] \\
&= \mathbb{E}\big[\langle \boldsymbol{v}_i, \boldsymbol{u}_{f^*(x)} - \boldsymbol{u}_y\rangle^2\langle \boldsymbol{v}_i, \boldsymbol{u}_{f^*(z)}\rangle^2\big]\mathbb{E}\big[\sigma'(\langle \boldsymbol{w}_i, \boldsymbol{e}_x\rangle)^2 h_k(\langle \boldsymbol{e}_z, \boldsymbol{w}_i\rangle)^2\langle \boldsymbol{e}_x, \boldsymbol{e}_z\rangle^2\big].
\end{aligned}$$

The first quantity can be bounded as

$$\mathbb{E}\big[\langle \boldsymbol{v}_i, \boldsymbol{u}_{f^*(x)} - \boldsymbol{u}_y\rangle^2 \langle \boldsymbol{v}_i, \boldsymbol{u}_{f^*(z)}\rangle^2\big] \leq \mathbb{E}\big[\langle \boldsymbol{v}_i, \boldsymbol{u}_{f^*(x)} - \boldsymbol{u}_y\rangle^4\big]^{1/2} \mathbb{E}\big[\langle \boldsymbol{v}_i, \boldsymbol{u}_{f^*(z)}\rangle^4\big]^{1/2}$$
$$\leq 2 \cdot \sqrt{3} \cdot \sqrt{3} = 6.$$

The second term is bounded as

$$\mathbb{E}\big[\sigma'(\langle \boldsymbol{w}_i, \boldsymbol{e}_x\rangle)^2 h_k(\langle \boldsymbol{e}_z, \boldsymbol{w}_i\rangle)^2 \langle \boldsymbol{e}_x, \boldsymbol{e}_z\rangle^2\big] \leq \mathbb{E}\big[\sigma'(\langle \boldsymbol{w}_i, \boldsymbol{e}_x\rangle)^8\big]^{1/4} \mathbb{E}\big[h_k(\langle \boldsymbol{e}_z, \boldsymbol{w}_i\rangle)^8\big]^{1/4} \mathbb{E}\big[\langle \boldsymbol{e}_x, \boldsymbol{e}_z\rangle^4\big]^{1/2}$$

By Gaussian hypercontractivity (Lemma 13),

$$\mathbb{E}\big[\sigma'(\langle \boldsymbol{w}_i, \boldsymbol{e}_x\rangle)^8\big]^{1/4} = \|\sigma'\|_{L^8}^2 \leq 8^q \|\sigma'\|_{L^2}^2 \lesssim 8^q,$$

and likewise

$$\mathbb{E}\big[h_k(\langle \boldsymbol{e}_z, \boldsymbol{w}_i\rangle)^8\big]^{1/4} \leq 8^k.$$

Finally, $\mathbb{E}\big[\langle \boldsymbol{e}_x, \boldsymbol{e}_z\rangle^4\big]^{1/2} \leq 4d^{-1}$ if $x \neq z$, and 1 otherwise. Altogether,

$$\mathbb{E}[\omega_{iz}^{xy}\omega_{iz}^{xy}] \lesssim 2^{3q+3k}\big(d^{-1} + \mathbf{1}(x = z)\big).$$

Altogether, we get that

$$\mathbb{E}\big[\gamma_{xy}^2\big] = \frac{m-1}{m}\sum_z \mathbb{E}[\omega_{iz}^{xy}\omega_{jz}^{xy}] + \frac{1}{m}\sum_z \mathbb{E}[\omega_{iz}^{xy}\omega_{iz}^{xy}].$$

The first quantity is

$$\sum_z \mathbb{E}[\omega_{iz}^{xy}\omega_{jz}^{xy}] \leq \left(1 + \frac{1}{d}\right)c_{k+1}^2 + c_{k+1}^2(2k+2)^{k+1}d^{-(k+1)} \cdot 2N.$$

The second quantity is

$$\sum_z \mathbb{E}[\omega_{iz}^{xy}\omega_{iz}^{xy}] \leq 2^{3q+3k}\left(1 + \frac{N}{d}\right).$$

Therefore

$$\mathrm{Var}(\gamma_{xy}) \lesssim \frac{1}{d} + \frac{(2k)^{k+1}N}{d^{k+1}} + \frac{2^{3q+3k}N}{md}.$$

Choose $k = 2\lceil\frac{1}{\epsilon}\rceil$; then

$$\frac{d^k}{(2k)^{k+1}N} \geq \frac{N}{(4/\epsilon)^{1+2/\epsilon}} \geq 1$$

for $N \geq C_1(\epsilon)$, and so

$$\mathrm{Var}(\gamma_{xy}) \lesssim \frac{1}{d} + \frac{2^{3q+3k}N}{md}.$$

Observe that $\gamma_{xy}$ is a degree $2q + 2k + 4 \leq 4q + 4$ polynomial. If $md \gtrsim N \cdot C_3^q \log^{4q+4}(1/\delta')$ for unspecified constant $C_3$, then $\frac{2^{4q+4}e^{-1}\log^{4q+4}(1/\delta')\mathrm{Var}(\gamma_{xy})}{(\mathbb{E}\gamma_{xy})^2} \leq 1$, and thus by Lemma 14, we have that $\mathbb{P}(\gamma_{xy} \leq 0) \leq \delta'$. Choosing $\delta' = \frac{\delta}{MN}$ and union bounding over all $(x, y)$ pairs with $y \neq f^*(x)$ yields the desired result.

$\square$

## B.1 BOUNDED BIT COMPLEXITY

**Corollary 2.** *Under the setting of Theorem 1, if $d^2 \gtrsim N \operatorname{poly} \log N$, then with high probability there exists a quantized weight matrix $\tilde{W}$, where each weight requires $O(\log d)$ bits to store, such that*

$$\arg \max_{y \in [M]} \boldsymbol{u}_y^\top \tilde{W} \boldsymbol{e}_x = f^*(x) \quad \text{for all } x \in [N]. \tag{20}$$

*Proof.* One sees from the proof of Theorem 1 that, with high probability over the embeddings, the weight matrix $W = \sum_{z \in [N]} \boldsymbol{u}_{f^*(z)} \boldsymbol{e}_z^\top$ has a margin $\gamma_{xy}$ satisfies $\gamma_{xy} \geq \frac{1}{2}$ for all $y \neq f^*(x)$. Each entry of $W$ lies in the interval $[-N, N]$. For some $\epsilon > 0$, define $\tilde{W}$ by rounding each entry of $W$ to the nearest multiple of $\epsilon$. By definition, $\left\| W - \tilde{W} \right\|_\infty \leq \epsilon$. We also see that

$$\left| \boldsymbol{u}_y^\top (W - \tilde{W}) \boldsymbol{e}_x \right| \leq \left\| W - \tilde{W} \right\|_\infty \left\| \boldsymbol{u}_y \boldsymbol{e}_x^\top \right\|_1 \leq d\epsilon.$$

Thus choosing $\epsilon < \frac{1}{8d}$, the margin of the quantized network satisfies

$$\begin{aligned}
\tilde{\gamma}_{xy} &:= (\boldsymbol{u}_{f^*(x)} - \boldsymbol{u}_y)^\top \tilde{W} \boldsymbol{e}_x \\
&\geq (\boldsymbol{u}_{f^*(x)} - \boldsymbol{u}_y)^\top W \boldsymbol{e}_x - \left| (\boldsymbol{u}_{f^*(x)} - \boldsymbol{u}_y)^\top (W - \tilde{W}) \boldsymbol{e}_x \right| \\
&\geq \frac{1}{2} - 2d\epsilon \\
&> 0.
\end{aligned}$$

Finally, the number of bits required to store each weight is $\log(2N/\epsilon) = \log(16Nd) = O(\log d)$. $\square$

We remark that a similar quantization argument was proven in Jelassi et al. (2024).

## C PROOFS FOR SECTION 4

**Lemma 1.** *Let $\mathcal{V}^{(h)} \subset \mathcal{S} \cup \mathcal{R}$. Assume that $d \gtrsim \left| \mathcal{V}^{(h)} \right| \log(|\mathcal{V}|/\delta)$. Define $\boldsymbol{v} := \sum_{z \in \mathcal{V}^{(h)}} \boldsymbol{\varphi}(z) + \frac{1}{2} \boldsymbol{\varphi}(\text{EOS})$. Then, with probability $1 - \delta$ over the draw of the embeddings,*

$$\langle \boldsymbol{v}, \boldsymbol{\varphi}(z) \rangle > \langle \boldsymbol{v}, \boldsymbol{\varphi}(\text{EOS}) \rangle + \frac{1}{4} > \langle \boldsymbol{v}, \boldsymbol{\varphi}(z') \rangle + \frac{1}{2}$$

*for any $z \in \mathcal{V}^{(h)}$ and $z' \notin \mathcal{V}^{(h)}$.*

*Proof.* Define $\gamma_z$ as

$$\gamma_z := \begin{cases} \langle \boldsymbol{v}, \boldsymbol{\varphi}(z) \rangle - 1 & z \in \mathcal{V}^{(h)} \\ \langle \boldsymbol{v}, \boldsymbol{\varphi}(\text{EOS}) \rangle - \frac{1}{2} & z = \text{EOS} \\ \langle \boldsymbol{v}, \boldsymbol{\varphi}(z) \rangle & z \notin \mathcal{V}^{(h)} \end{cases}$$

We first see that $\mathbb{E}[\gamma_z] = 0$.

Next, observe that

$$\gamma_z = \begin{cases} \sum_{z' \in \mathcal{V}^{(h)}} \langle \boldsymbol{\varphi}(z), \boldsymbol{\varphi}(z') \rangle & z \notin \mathcal{V}^{(h)} \\ \sum_{z' \in \mathcal{V}^{(h)} \setminus \{z\}} \langle \boldsymbol{\varphi}(z), \boldsymbol{\varphi}(z') \rangle & z \in \mathcal{V}^{(h)} \end{cases}$$

Since each of the $\langle \boldsymbol{\varphi}(z), \boldsymbol{\varphi}(z') \rangle$ are independent subGaussian variables with variance proxy $1/d$, by Hoeffding's inequality we have that, with probability $1 - \delta'$,

$$|\gamma_z| \lesssim \sqrt{\frac{\left| \mathcal{V}^{(h)} \right| \cdot \log(1/\delta')}{d}}.$$

Setting $\delta' = \delta/|\mathcal{V}|$ and union bounding over all $z \in \mathcal{V}$ yields the desired result. $\square$

### C.1 CONSTRUCTION VIA SELF-ATTENTION

**Lemma 2.** *Let $\mathcal{V}^{(h)} \subset \mathcal{V}$, and for each $z \in \mathcal{V}^{(h)}$, let $\mathcal{A}^z \subset \mathcal{V}$. Assume that $d \gtrsim \max_{z \in \mathcal{V}^{(h)}} |\mathcal{A}^z| \log^6(|\mathcal{V}|/\delta)$ and $d_h \gtrsim |\mathcal{V}^{(h)}| \log^6(|\mathcal{V}|/\delta)$ Define*

$$\boldsymbol{W} := \frac{d}{d_h} \sum_{z \in \mathcal{V}^{(h)}} \sum_{a \in \mathcal{A}^z} \sum_{i=1}^{d_h} \boldsymbol{\varphi}(a)\boldsymbol{\varphi}(z)^\top \boldsymbol{w}_i \boldsymbol{w}_i^\top,$$

*where $\boldsymbol{w}_i$ are chosen uniformly on the sphere of radius 1, conditioned on being orthogonal to $\boldsymbol{\varphi}(\mathrm{EOS})$. Then, with probability $1 - \delta$ over the draw of the embeddings and the $\boldsymbol{w}_i$,*

$$\left| \boldsymbol{\varphi}(a)^\top \boldsymbol{W} \boldsymbol{\varphi}(z) - \mathbf{1}(a \in \mathcal{A}^z) \right| \leq \frac{1}{5}$$

*for all $z \in \mathcal{V}^{(h)}$, $a \in \mathcal{V}$.*

*Proof.* Define

$$\gamma_{az} := \boldsymbol{\varphi}(a)^\top \boldsymbol{W} \boldsymbol{\varphi}(z).$$

We first see that

$$\mathbb{E}[\gamma_{az}] = \mathbb{E}\left[ \sum_{z' \in \mathcal{V}^{(h)}} \sum_{a' \in \mathcal{A}^z} \langle \boldsymbol{\varphi}(a), \boldsymbol{\varphi}(a')\rangle \langle \boldsymbol{\varphi}(z), P^\perp_{\boldsymbol{\varphi}(\mathrm{EOS})}\boldsymbol{\varphi}(z')\rangle \right] = \frac{d-1}{d} \cdot \mathbf{1}(a \in \mathcal{A}^z).$$

We next compute the variance. For $a \notin \mathcal{A}^z$,

$$\mathbb{E}\left[\gamma_{az}^2\right] = \frac{d^2}{d_h^2}\mathbb{E}\left[ \left( \sum_{z' \in \mathcal{V}^{(h)}} \sum_{a' \in \mathcal{A}^{z'}} \sum_{i=1}^{d_h} \langle \boldsymbol{\varphi}(a), \boldsymbol{\varphi}(a')\rangle \langle \boldsymbol{\varphi}(z), \boldsymbol{w}_i\rangle\langle \boldsymbol{w}_i, \boldsymbol{\varphi}(z')\rangle \right)^2 \right]$$

Define $\omega_{a'z'i} = \langle \boldsymbol{\varphi}(a), \boldsymbol{\varphi}(a')\rangle \langle \boldsymbol{\varphi}(z), \boldsymbol{w}_i\rangle\langle \boldsymbol{w}_i, \boldsymbol{\varphi}(z')\rangle$. We see that $\mathbb{E}[\omega_{a_1 z_1 i}\omega_{a_2 z_2 j}]$ is nonzero only if $a_1 = a_2$ and $z_1 = z_2$. For $i \neq j$, we have that

$$\mathbb{E}[\omega_{a'z'i}\omega_{a'z'j}] = \mathbb{E}\left[\langle \boldsymbol{\varphi}(a), \boldsymbol{\varphi}(a')\rangle^2 \langle \boldsymbol{\varphi}(z), \boldsymbol{w}_i\rangle\langle \boldsymbol{w}_i, \boldsymbol{\varphi}(z')\rangle\langle \boldsymbol{\varphi}(z), \boldsymbol{w}_j\rangle\langle \boldsymbol{w}_j, \boldsymbol{\varphi}(z')\rangle\right]$$

$$= d^{-2}\mathbb{E}\left[\langle \boldsymbol{\varphi}(a), \boldsymbol{\varphi}(a')\rangle^2\right]\mathbb{E}\left[\langle \boldsymbol{\varphi}(z), P^\perp_{\boldsymbol{\varphi}(\mathrm{EOS})}\boldsymbol{\varphi}(z')\rangle^2\right]$$

$$\leq d^{-2}\rho_{aa'}\rho_{zz'},$$

where $\rho_{ij} = \begin{cases} 1 & i = j \\ d^{-1} & i \neq j \end{cases}$. Also,

$$\mathbb{E}\left[\omega_{a'z'i}^2\right] = \mathbb{E}\left[\langle \boldsymbol{\varphi}(a), \boldsymbol{\varphi}(a')\rangle^2 \langle \boldsymbol{\varphi}(z), \boldsymbol{w}_i\rangle^2\langle \boldsymbol{w}_i, \boldsymbol{\varphi}(z')\rangle^2\right].$$

Since $\mathbb{E}\left[\boldsymbol{w}_i^{\otimes 4}\right] = \frac{3}{(d-1)(d+1)}\mathrm{Sym}\left((P^\perp_{\boldsymbol{\varphi}(\mathrm{EOS})})^{\otimes 2}\right)$,

$$\mathbb{E}\left[\omega_{a'z'i}^2\right] \leq (d^2 - 1)^{-1}\mathbb{E}\left[\langle \boldsymbol{\varphi}(a), \boldsymbol{\varphi}(a')\rangle^2\right] \left( \mathbb{E}\left[\left\|P^\perp_{\boldsymbol{\varphi}(\mathrm{EOS})}\boldsymbol{\varphi}(z)\right\|^2\right]^2 + 2\mathbb{E}\left[\langle \boldsymbol{\varphi}(z), P^\perp_{\boldsymbol{\varphi}(\mathrm{EOS})}\boldsymbol{\varphi}(z')\rangle^2\right] \right)$$

$$\leq d^{-2}\rho_{aa'}(1 + 2\rho_{zz'}).$$

Altogether,

$$\mathbb{E}\left[\gamma_{az}^2\right] = \frac{d^2}{d_h^2} \sum_{z' \in \mathcal{V}^{(h)}} \sum_{a' \in \mathcal{A}^{z'}} \sum_{i,j=1}^{d_h} \mathbb{E}[\omega_{a'z'i}\omega_{a'z'j}]$$

$$\leq \sum_{z' \in \mathcal{V}^{(h)}} \sum_{a' \in \mathcal{A}^{z'}} \left( \frac{d_h - 1}{d_h}\rho_{aa'}\rho_{zz'} + \frac{1}{d_h}\rho_{aa'}(1 + 2\rho_{zz'}) \right)$$

$$= \frac{d_h + 2}{d_h} \sum_{a' \in \mathcal{A}^z} \rho_{aa'} + \frac{d + d_h + 1}{dd_h} \sum_{z' \in \mathcal{V}^{(h)}\setminus\{z\}} \sum_{a' \in \mathcal{A}^{z'}} \rho_{aa'}$$

$$\leq \left( \frac{d_h + 2}{d_h} \right)\left( \mathbf{1}(a \in \mathcal{A}^z) + \frac{|\mathcal{A}^z|}{d} \right) + \frac{d + d_h + 1}{dd_h} \cdot \sum_{z' \in \mathcal{V}^{(h)}\setminus\{z\}} \left( \mathbf{1}(a \in \mathcal{A}^{z'}) + \frac{|\mathcal{A}^{z'}|}{d} \right),$$

and thus

$$\mathrm{Var}(\gamma_{az}) = \mathbb{E}\big[\gamma_{az}^2\big] - \frac{d-1}{d} \cdot \mathbf{1}(a \in \mathcal{A}^z)$$

$$\lesssim \frac{|\mathcal{A}^z|}{d} + \frac{1}{d_h} \sum_{z' \in \mathcal{V}^{(h)}} \left(\mathbf{1}(a \in \mathcal{A}^{z'}) + \frac{\left|\mathcal{A}^{z'}\right|}{d}\right)$$

$$\lesssim \frac{\max_{z \in \mathcal{V}^{(h)}} |\mathcal{A}^z|}{d} + \frac{|\mathcal{V}^{(h)}|}{d_h},$$

since $d \geq |\mathcal{A}^z|, d_h \geq \left|\mathcal{V}^{(h)}\right|$ Next, since $\gamma_{az}$ is a degree 6 polynomial, with probability $1 - \frac{\delta}{|\mathcal{V}||\mathcal{V}^{(h)}|}$ we have that

$$|\gamma_{az} - \mathbf{1}(a \in \mathcal{A}^z)| \lesssim \sqrt{\mathrm{Var}(\gamma_{az}) \log^6(|\mathcal{V}|/\delta)}$$

$$\lesssim \sqrt{\frac{\max_{z \in \mathcal{V}^{(h)}} |\mathcal{A}^z| \log^6(|\mathcal{V}|/\delta)}{d} + \frac{|\mathcal{V}^{(h)}| \log^6(|\mathcal{V}|/\delta)}{d_h}}$$

$$\leq \frac{1}{5}.$$

Union bounding over all $z \in \mathcal{V}^{(h)}$, $a \in \mathcal{V}$ yields the desired result. $\qquad\square$

Let us state the formal version of Theorem 3 which we aim to prove:

**Theorem 10.** *Assume that $d \gtrsim \max(R, D) \cdot \log^6(|\mathcal{V}|SR/\delta)$ and $Hd_h \gtrsim S \log^6(|\mathcal{V}|SR/\delta)$. Then, with probability $1 - \delta$, there exists a single-layer attention-only transformer $F_{\mathrm{TF}}(\ \cdot\ ; \boldsymbol{\theta}_{\mathrm{TF}})$ with embedding dimension $d$, number of heads $H$ and head dimension $d_h$ such that*

$$\mathbb{P}_{z_{1:T+1} \sim \mathcal{D}} \left[\arg\max_{z \in \mathcal{V}} \boldsymbol{\varphi}(z)^\top F_{\mathrm{TF}}(\boldsymbol{X}; \boldsymbol{\theta}_{\mathrm{TF}}) = z_{T+1}\right] = 1.$$

**Remark.** When $R \geq D$, one can obtain an accuracy of 100% whenever the total parameter count is

$$Hdd_h \gtrsim SR \operatorname{poly} \log(|\mathcal{V}|SR/\delta).$$

*Proof of Theorem 10.* Partition $\mathcal{S}$ into the sets $\mathcal{S}^{(1)}, \ldots, \mathcal{S}^{(N_S)}$ and $\mathcal{R}$ into the sets $\mathcal{R}^{(1)}, \ldots, \mathcal{R}^{(N_R)}$, such that $\left|\mathcal{S}^{(i)}\right|, \left|\mathcal{R}^{(j)}\right| \leq M$ and $N_S = \lceil \frac{S}{M} \rceil, N_R = \lceil \frac{R}{M} \rceil$.

Let us choose $M$ so that $d \geq d_h \gtrsim M \log^6(|\mathcal{V}|/\delta)$. The total number of attention heads is then

$$H = N_S + N_R \gtrsim \frac{S \log^6(|\mathcal{V}|/\delta)}{d_h}.$$

For each $i \in [N_S]$, we construct the attention head $i$ as follows. First, let

$$\boldsymbol{W}_K^{(i)^\top} \boldsymbol{W}_Q^{(i)} = \beta \sum_{z \in \mathcal{S}^{(i)}} \boldsymbol{\varphi}(z)\boldsymbol{\varphi}(\mathrm{EOS})^\top + \frac{\beta}{2}\boldsymbol{\varphi}(\mathrm{EOS})\boldsymbol{\varphi}(\mathrm{EOS})^\top$$

for a large constant $\beta$. Next, set

$$\boldsymbol{W}_O^{(i)^\top} \boldsymbol{W}_V^{(i)} = \frac{d}{d_h} \sum_{z \in \mathcal{S}^{(i)}} \sum_{a \in \mathcal{A}_z} \sum_{i=1}^{d_h} \boldsymbol{\varphi}(a)\boldsymbol{\varphi}(z)^\top \boldsymbol{w}_i \boldsymbol{w}_i^\top,$$

for $\boldsymbol{w}_i$ sampled uniformly on the sphere, orthogonal to $\boldsymbol{\varphi}(\mathrm{EOS})$.

Consider an input sequence $(z_1, \ldots, z_T)$, and let $s$ be the subject token in this sequence. On the event that Lemma 1 holds, if $s \notin \mathcal{S}^{(i)}$, then $z_t \notin \mathcal{S}^{(i)}$, and thus

$$\boldsymbol{\varphi}(z_t)^\top \boldsymbol{W}_K^{(i)^\top} \boldsymbol{W}_Q^{(i)} \boldsymbol{\varphi}(\mathrm{EOS}) < \boldsymbol{\varphi}(\mathrm{EOS})^\top \boldsymbol{W}_K^{(i)^\top} \boldsymbol{W}_Q^{(i)} \boldsymbol{\varphi}(\mathrm{EOS}) - \frac{\beta}{2}$$

for all $t < T$. As $\beta \to \infty$, the self-attention module fully attends to the EOS token. On the other hand, if $s \in \mathcal{S}^{(i)}$, then if $z_{t^*} = s$ we have

$$\boldsymbol{\varphi}(z_t)^\top {\boldsymbol{W}_K^{(i)}}^\top \boldsymbol{W}_Q^{(i)} \boldsymbol{\varphi}(\text{EOS}) < \boldsymbol{\varphi}(z_{t^*})^\top {\boldsymbol{W}_K^{(i)}}^\top \boldsymbol{W}_Q^{(i)} \boldsymbol{\varphi}(\text{EOS}) - \frac{\beta}{2}$$

for all $t \neq t^*$. Likewise, as $\beta \to \infty$, the softmax converges to a hardmax on the $z_{t^*}$ token. Altogether, we get that

$$\boldsymbol{X}^\top \mathcal{S}\left( \boldsymbol{X} {\boldsymbol{W}_K^{(i)}}^\top \boldsymbol{W}_Q^{(i)} \boldsymbol{x}_T \right) = \begin{cases} \boldsymbol{\varphi}(\text{EOS}) & s \notin \mathcal{S}^{(i)} \\ \boldsymbol{\varphi}(s) & s \in \mathcal{S}^{(i)} \end{cases}.$$

Next, on the event that Lemma 2 holds, since $d \gtrsim R \log^6(|\mathcal{V}|/\delta) \geq |\mathcal{A}_s| \log^6(|\mathcal{V}|/\delta)$, we have that

$$\left| \boldsymbol{\varphi}(a)^\top {\boldsymbol{W}_O^{(i)}}^\top \boldsymbol{W}_V^{(i)} \boldsymbol{\varphi}(s) - \mathbf{1}(a \in \mathcal{A}_s) \right| \leq \frac{1}{6}$$

for $s \in \mathcal{S}^{(i)}$. Defining $\text{attn}_i := {\boldsymbol{W}_O^{(i)}}^\top \boldsymbol{W}_V^{(i)} \mathcal{S}\left( \boldsymbol{X} {\boldsymbol{W}_K^{(i)}}^\top \boldsymbol{W}_Q^{(i)} \boldsymbol{x}_T \right)$, we have that

$$\boldsymbol{\varphi}(a)^\top \text{attn}_i \in \begin{cases} \{0\} & s \notin \mathcal{S}^{(i)} \\ [-\frac{1}{5}, \frac{1}{5}] & s \in \mathcal{S}^{(i)}, a \notin \mathcal{A}_s \\ [\frac{4}{5}, \frac{6}{5}] & s \in \mathcal{S}^{(i)}, a \in \mathcal{A}_s \end{cases}.$$

By an identical construction, for each $j \in [N_R]$, with probability $1 - 2\delta$ we can construct the attention head $N_S + j$ such that

$$\boldsymbol{\varphi}(a)^\top \text{attn}_{N_S+j} \in \begin{cases} \{0\} & r \notin \mathcal{R}^{(i)} \\ [-\frac{1}{5}, \frac{1}{5}] & r \in \mathcal{R}^{(i)}, a \notin \mathcal{A}_r \\ [\frac{4}{5}, \frac{6}{5}] & r \in \mathcal{R}^{(i)}, a \in \mathcal{A}_r \end{cases},$$

as long as $d \gtrsim D \log^6(|\mathcal{V}|/\delta) \geq |\mathcal{A}_r| \log^6(|\mathcal{V}|/\delta)$. Therefore by a union bound, with probability $1 - 2SR\delta$ we have that (where $s \in \mathcal{S}^{(i)}$ and $r \in \mathcal{S}^{(j)}$)

$$\boldsymbol{\varphi}(a)^\top F_{\text{MHSA}}(\boldsymbol{X}; \boldsymbol{\theta}) = \sum_{h=1}^{N_S+N_R} \boldsymbol{\varphi}(a)^\top \text{attn}_h$$
$$= \boldsymbol{\varphi}(a)^\top \text{attn}_i + \boldsymbol{\varphi}(a)^\top \text{attn}_{N_S+j}$$

If $a = a^*(s, r)$, then $a \in \mathcal{A}_s \cap \mathcal{A}_r$, and thus

$$\boldsymbol{\varphi}(a)^\top F_{\text{MHSA}}(\boldsymbol{X}; \boldsymbol{\theta}) \geq \frac{4}{5} + \frac{4}{5} = \frac{8}{5}.$$

Otherwise, either $\boldsymbol{\varphi}(a)^\top \text{attn}_i$ or $\boldsymbol{\varphi}(a)^\top \text{attn}_{N_S+j}$ is $\leq \frac{1}{5}$ and thus

$$\boldsymbol{\varphi}(a)^\top F_{\text{MHSA}}(\boldsymbol{X}; \boldsymbol{\theta}) \geq \frac{6}{5} + \frac{1}{5} = \frac{7}{5}.$$

Therefore $\arg\max_{a \in \mathcal{V}} \boldsymbol{\varphi}(a)^\top F_{\text{MHSA}}(\boldsymbol{X}; \boldsymbol{\theta}) = a^*(s, r)$. Replacing $2SR\delta$ with $\delta$ yields the desired result. $\qquad\square$

## C.2 CONSTRUCTION VIA MLP

**Lemma 3.** *Let $\epsilon$ be a fixed constant. Assume that $q$ in Assumption 3 satisfies $q = \frac{C_2}{\epsilon}$ for some $C_2 > 2$. Assume that $d \geq S^\epsilon, R^\epsilon$. Define $C(a) = |\{(s, r) : a^*(s, r) = a\}|$.*

*Let $d$ be odd, and let $P, Q$ be orthogonal $\lfloor d/2 \rfloor$ dimensional subspaces of $\mathbb{R}^d$. Define $\tilde{\boldsymbol{\varphi}}(s) = \Pi_P \boldsymbol{\varphi}(s), \tilde{\boldsymbol{\varphi}}(r) = \Pi_Q \boldsymbol{\varphi}(r)$.*

*There exists universal constants $C_3, C_4$ such that if*

$$d \gtrsim (C_3 \log(|\mathcal{V}|/\delta)/\epsilon)^{C_4/\epsilon}$$
$$m \gtrsim (C_3 \log(|\mathcal{V}|/\delta))^{C_4/\epsilon} \cdot \max_a C(a)$$
$$md \gtrsim (C_3 \log(|\mathcal{V}|/\delta))^{C_4/\epsilon} \cdot SR,$$

*then with probability $1 - \delta$ over the draw of the embeddings there exists a two-layer neural network $F(\boldsymbol{z}) = \sum_{i \in [m]} \boldsymbol{v}_i \sigma(\boldsymbol{w}_i^\top \boldsymbol{z})$ of width $m$ satisfying*

$$\arg\max_{a \in \mathcal{V}} \boldsymbol{\varphi}(a)^\top F(\tilde{\boldsymbol{\varphi}}(s) + \tilde{\boldsymbol{\varphi}}(r)) = a^*(s, r)$$

*for all $s \in \mathcal{S}, r \in \mathcal{R}$.*

*Proof.* For odd integers $p, k$ to be determined later, let us set

$$\boldsymbol{v}_i = \frac{1}{m} \sum_{s,r} \langle \mathbf{He}_{p+k}(\boldsymbol{w}_i), \tilde{\boldsymbol{\varphi}}(s)^{\otimes p} \otimes \tilde{\boldsymbol{\varphi}}(r)^{\otimes k} \rangle \cdot \boldsymbol{\varphi}(a^*(s, r)),$$

where $\mathbf{He}_{p+k} : \mathbb{R}^d \to (\mathbb{R}^d)^{\otimes(p+k)}$ is the Hermite tensor of degree $p + k$ (see Appendix F.1). Assume without loss of generality that $c_{p+k} := \mathbb{E}[\sigma^{(p+k)}(z)]$, the $(p + k)$th Hermite coefficient of $\sigma$ is positive (the negative case can be handled by negating all the $\boldsymbol{v}_i$ in the construction)

For some $(s, r)$, the margin for some $a \neq a^*(s, r)$ is

$$
\begin{aligned}
\gamma_{sra} &= \langle \boldsymbol{\varphi}(a^*(s, r)) - \boldsymbol{\varphi}(a), F(\tilde{\boldsymbol{\varphi}}(s) + \tilde{\boldsymbol{\varphi}}(r)) \rangle \\
&= \frac{1}{m} \sum_{i \in [m]} \sum_{s',r'} \sigma(\langle \boldsymbol{w}_i, \tilde{\boldsymbol{\varphi}}(s) + \tilde{\boldsymbol{\varphi}}(r) \rangle) \langle \mathbf{He}_{p+k}(\boldsymbol{w}_i), \boldsymbol{\varphi}(s')^{\otimes p} \otimes \tilde{\boldsymbol{\varphi}}(r')^{\otimes k} \rangle \\
&\quad \cdot \langle \boldsymbol{\varphi}(a^*(s', r')), \boldsymbol{\varphi}(a^*(s, r)) - \boldsymbol{\varphi}(a) \rangle.
\end{aligned}
$$

We first see that

$$
\begin{aligned}
&\mathbb{E}[\gamma_{sra}] \\
&= \sum_{s'r'} \mathbb{E}\big[\sigma(\langle \boldsymbol{w}_i, \tilde{\boldsymbol{\varphi}}(s) + \tilde{\boldsymbol{\varphi}}(r) \rangle) \langle \mathbf{He}_{p+k}(\boldsymbol{w}_i), \boldsymbol{\varphi}(s')^{\otimes p} \otimes \tilde{\boldsymbol{\varphi}}(r')^{\otimes k} \rangle\big] \\
&\quad \cdot (\mathbf{1}(a^*(s, r) = a^*(s', r')) - \mathbf{1}(a = a^*(s', r'))) \\
&= \sum_{s'r'} \mathbb{E}\big[\sigma^{(p+k)}(\langle \boldsymbol{w}_i, \tilde{\boldsymbol{\varphi}}(s) + \tilde{\boldsymbol{\varphi}}(r) \rangle) \langle (\tilde{\boldsymbol{\varphi}}(s) + \tilde{\boldsymbol{\varphi}}(r))^{\otimes(p+k)}, \boldsymbol{\varphi}(s')^{\otimes p} \otimes \tilde{\boldsymbol{\varphi}}(r')^{\otimes k} \rangle\big] \\
&\quad \cdot (\mathbf{1}(a^*(s, r) = a^*(s', r')) - \mathbf{1}(a = a^*(s', r')))
\end{aligned}
$$

If either $s' \neq s$ or $r \neq r'$, we see that conditioned on $\tilde{\boldsymbol{\varphi}}(s), \tilde{\boldsymbol{\varphi}}(r)$, the quantity $\boldsymbol{\varphi}(s')^{\otimes p} \otimes \tilde{\boldsymbol{\varphi}}(r')^{\otimes k}$ is mean zero. Therefore the only nonzero term in the sum is when $(s, r) = (s', r')$, and so

$$\mathbb{E}[\gamma_{sra}] = \mathbb{E}_{Z \sim \mathcal{N}(0,1)}\left[\sigma^{(p+k)}\left(Z\sqrt{\|\tilde{\boldsymbol{\varphi}}(s)\|^2 + \|\tilde{\boldsymbol{\varphi}}(r)\|^2}\right) \cdot \|\tilde{\boldsymbol{\varphi}}(s)\|^{2p} \|\tilde{\boldsymbol{\varphi}}(r)\|^{2k}\right]$$

The quantities $\|\tilde{\boldsymbol{\varphi}}(s)\|^2 - \frac{1}{2}, \|\tilde{\boldsymbol{\varphi}}(r)\|^2 - \frac{1}{2}$ are subexponential random variables with Orlicz norm $1/d$, and therefore we can bound

$$\left|\mathbb{E}[\gamma_{sra}] - c_{p+k} 2^{-p-k}\right| \lesssim \frac{1}{d}$$

We next compute the variance. Define $\omega_{is'r'}$ by

$$\omega_{is'r'} = \sigma(\langle \boldsymbol{w}_i, \tilde{\boldsymbol{\varphi}}(s) + \tilde{\boldsymbol{\varphi}}(r) \rangle) \langle \mathbf{He}_{p+k}(\boldsymbol{w}_i), \boldsymbol{\varphi}(s')^{\otimes p} \otimes \tilde{\boldsymbol{\varphi}}(r')^{\otimes k} \rangle \cdot \langle \boldsymbol{\varphi}(a^*(s', r')), \boldsymbol{\varphi}(a^*(s, r)) - \boldsymbol{\varphi}(a) \rangle$$

We first observe that $\mathbb{E}[\omega_{is_1r_1} \omega_{js_2r_2}]$ is zero, unless $s_1 = s_2$ and $r_1 = r_2$. Next, we compute the expectation of $\omega_{is'r'}$, conditioned on the embeddings (i.e with respect to the randomness $\boldsymbol{w}_i$):

$$
\begin{aligned}
&\mathbb{E}[\omega_{is'r'} \mid \boldsymbol{\varphi}] \\
&= \mathbb{E}_{\boldsymbol{w}_i}\left[\sigma^{(p+k)}(\langle \boldsymbol{w}_i, \tilde{\boldsymbol{\varphi}}(s) + \tilde{\boldsymbol{\varphi}}(r) \rangle)\right] \cdot \langle (\tilde{\boldsymbol{\varphi}}(s) + \tilde{\boldsymbol{\varphi}}(r))^{\otimes(p+k)}, \tilde{\boldsymbol{\varphi}}(s')^{\otimes p} \otimes \tilde{\boldsymbol{\varphi}}(r')^{\otimes k} \rangle \\
&\quad \cdot \langle \boldsymbol{\varphi}(a^*(s', r')), \boldsymbol{\varphi}(a^*(s, r)) - \boldsymbol{\varphi}(a) \rangle \\
&= \mathbb{E}_{Z \sim \mathcal{N}(0,1)}\left[\sigma^{(p+k)}\left(Z\sqrt{\|\tilde{\boldsymbol{\varphi}}(s)\|^2 + \|\tilde{\boldsymbol{\varphi}}(r)\|^2}\right)\right] \cdot \langle \tilde{\boldsymbol{\varphi}}(s), \tilde{\boldsymbol{\varphi}}(s') \rangle^p \cdot \langle \tilde{\boldsymbol{\varphi}}(r), \tilde{\boldsymbol{\varphi}}(r') \rangle^k \cdot \langle \boldsymbol{\varphi}(a^*(s', r')), \boldsymbol{\varphi}(a^*(s, r)) - \boldsymbol{\varphi}(a) \rangle.
\end{aligned}
$$

Therefore for $i \neq j$,

$$\mathbb{E}[\omega_{is'r'}\omega_{js'r'}] = \mathbb{E}\Big[\mathbb{E}[\omega_{is'r'} \mid \boldsymbol{\varphi}]^2\Big]$$

$$\lesssim c_{p+k}^2 \mathbb{E}\big[\langle\tilde{\boldsymbol{\varphi}}(s), \boldsymbol{\varphi}(s')\rangle^{2p}\big]\mathbb{E}\big[\langle\tilde{\boldsymbol{\varphi}}(r), \tilde{\boldsymbol{\varphi}}(r')\rangle^{2k}\big]\mathbb{E}\big[\langle\boldsymbol{\varphi}(a^*(s',r')), \boldsymbol{\varphi}(a^*(s,r)) - \boldsymbol{\varphi}(a)\rangle^2\big].$$

When $(s,r) = (s',r')$, then

$$\mathbb{E}[\omega_{isr}\omega_{jsr}] = \mathbb{E}\left[\mathbb{E}_{Z \sim \mathcal{N}(0,1)}\left[\sigma^{(p+k)}\left(Z\sqrt{\|\tilde{\boldsymbol{\varphi}}(s)\|^2 + \|\tilde{\boldsymbol{\varphi}}(r)\|^2}\right)\right]^2\|\tilde{\boldsymbol{\varphi}}(s)\|^{4p}\|\tilde{\boldsymbol{\varphi}}(r)\|^{4k}\right] \cdot (1 + d^{-1})$$

$$= c_{p+k}^2 2^{-2p-2k} + O(1/d)$$

Next, define the quantities

$$\rho_{ss'} = \mathbb{E}\big[\langle\tilde{\boldsymbol{\varphi}}(s), \tilde{\boldsymbol{\varphi}}(s')\rangle^{2p}\big]$$
$$\rho_{rr'} = \mathbb{E}\big[\langle\tilde{\boldsymbol{\varphi}}(r), \tilde{\boldsymbol{\varphi}}(r')\rangle^{2k}\big]$$
$$\rho_{aa'} = \mathbb{E}\big[\langle\boldsymbol{\varphi}(a), \boldsymbol{\varphi}(a')\rangle^2\big],$$

so that

$$\mathbb{E}[\omega_{is'r'}\omega_{js'r'}] \lesssim c_{p+k}^2 \rho_{ss'}\rho_{rr'}\big(\rho_{aa^*(s',r')} + \rho_{a^*(s,r)a^*(s'r')}\big).$$

We see that for $s \neq s', r \neq r', a \neq a'$,

$$\rho_{ss'} \leq (2p)^p (d/2)^{-p} = (4p)^p d^{-p}$$
$$\rho_{rr'} \leq (2k)^k (d/2)^{-k} = (4k)^k d^{-k}$$
$$\rho_{aa'} \leq d^{-1}$$

Next, see that

$$\mathbb{E}\big[\omega_{is'r'}^2\big]$$
$$= \mathbb{E}\big[\sigma(\langle\boldsymbol{w}_i, \tilde{\boldsymbol{\varphi}}(s) + \tilde{\boldsymbol{\varphi}}(r)\rangle)^2\langle\mathbf{He}_{p+k}(\boldsymbol{w}_i), \tilde{\boldsymbol{\varphi}}(s')^{\otimes p} \otimes \tilde{\boldsymbol{\varphi}}(r')^{\otimes k}\rangle^2\big]\mathbb{E}\big[\langle\boldsymbol{\varphi}(a^*(s',r')), \boldsymbol{\varphi}(a^*(s,r)) - \boldsymbol{\varphi}(a)\rangle^2\big]$$
$$\leq \mathbb{E}\big[\sigma(\langle\boldsymbol{w}_i, \tilde{\boldsymbol{\varphi}}(s) + \tilde{\boldsymbol{\varphi}}(r)\rangle)^4\big]^{1/2}\mathbb{E}\big[\langle\mathbf{He}_{p+k}(\boldsymbol{w}_i), \tilde{\boldsymbol{\varphi}}(s')^{\otimes p} \otimes \tilde{\boldsymbol{\varphi}}(r')^{\otimes k}\rangle^4\big]^{1/2}\big(\rho_{aa^*(s',r')} + \rho_{a^*(s,r)a^*(s'r')}\big)$$
$$\leq 2^{4q}\big(\rho_{aa^*(s',r')} + \rho_{a^*(s,r)a^*(s'r')}\big),$$

where we have applied Lemma 13 to the first two expectations. Altogether, we have that

$$\mathbb{E}\big[\gamma_{sra}^2\big] = \sum_{s',r'}\left(\frac{m-1}{m}\mathbb{E}[\omega_{is'r'}\omega_{js'r'}] + \frac{1}{m}\mathbb{E}\big[\omega_{is'r'}^2\big]\right)$$

$$= \frac{m-1}{m}\mathbb{E}[\omega_{isr}\omega_{jsr}] + \frac{m-1}{m}\sum_{(s',r')\neq(s,r)}\mathbb{E}[\omega_{is'r'}\omega_{js'r'}] + \frac{1}{m}\sum_{s',r'}\mathbb{E}\big[\omega_{is'r'}^2\big],$$

and thus

$$\mathrm{Var}(\gamma_{sra}) = \mathbb{E}\big[\gamma_{sra}^2\big] - c_{p+k}^2$$

$$\lesssim c_{p+k}^2\sum_{(s',r')\neq(s,r)}\rho_{ss'}\rho_{rr'}\big(\rho_{aa^*(s',r')} + \rho_{a^*(s,r)a^*(s'r')}\big) + \frac{2^{4q}}{m}\sum_{s',r'}\big(\rho_{aa^*(s',r')} + \rho_{a^*(s,r)a^*(s'r')}\big)$$

For the first sum, we can bound

$$\sum_{(s',r')\neq(s,r)}\rho_{ss'}\rho_{rr'}\big(\rho_{aa^*(s',r')} + \rho_{a^*(s,r)a^*(s'r')}\big) \leq \sum_{(s',r')\neq(s,r)}\rho_{ss'}\rho_{rr'}$$

$$\leq SR \cdot (4p)^p(4k)^k d^{-p-k} + S(4p)^p d^{-p} + R(4k)^k d^{-k}$$

For the second sum, we get that

$$\sum_{s',r'}\big(\rho_{aa^*(s',r')} + \rho_{a^*(s,r)a^*(s'r')}\big) \leq C(a) + C(a^*(s,r)) + \frac{2SR}{d}$$

Altogether,

$$\frac{\text{Var}(\gamma_{sra})}{\mathbb{E}[\gamma_{sra}]^2} \lesssim \frac{S(4p)^p}{d^p} + \frac{R(4k)^k}{d^k} + \frac{S(4p)^p}{d^p} \cdot \frac{R(4k)^k}{d^k} + \frac{2^{4q}(C(a) + C(a^*(s,r)))}{mc_{p+k}^2} + \frac{2^{4q}SR}{mdc_{p+k}^2}.$$

Let $\delta'$ be a fixed failure probability. We see that, for $p = 2\lceil \frac{1}{\epsilon} \rceil + 1$

$$\frac{2^{4q}\log^{4q+2}(1/\delta')S(4p)^p}{d^p} \lesssim \frac{2^{\frac{4C_2}{\epsilon}}\log^{\frac{4C_2}{\epsilon}+2}(1/\delta')(\frac{8}{\epsilon})^{2/\epsilon}}{d^{\frac{1}{\epsilon}}} \lesssim 1,$$

whenever $d \gtrsim (C_3 \log(1/\delta')/\epsilon)^{C_4/\epsilon}$ for appropriately chosen constants $C_3, C_4$. Likewise, setting $k = 2\lceil \frac{1}{\epsilon} \rceil + 1$, we get that

$$\frac{2^{4q}\log^{4q+2}(1/\delta')R(4k)^k}{d^k} \lesssim 1.$$

Next, setting $m \gtrsim 2^{8q+2}\log^{4q+2}(1/\delta')c_{p+k}^{-2} \cdot \max_a C(a)$ and $md \gtrsim 2^{8q+2}\log^{4q+2}(1/\delta')c_{p+k}^{-2}SR$ yields

$$2^{4q+2}\log^{4q+2}(1/\delta') \cdot \frac{2^{4q}(C(a) + C(a^*(s,r)))}{mc_{p+k}^2} \lesssim 1$$

$$2^{4q+2}\log^{4q+2}(1/\delta') \cdot \frac{2^{4q}SR}{mdc_{p+k}^2} \lesssim 1.$$

Altogether, by choosing constants appropriately, we get that

$$2^{4q+2}\log^{4q+2}(1/\delta')e^{-1} \cdot \frac{\text{Var}(\gamma_{sra})}{\mathbb{E}[\gamma_{sra}]^2} \leq 1.$$

Therefore by Lemma 14, with probability $1 - \delta'$ we have that $\gamma_{sra} > 0$. Union bounding over all $s, r, a$ and setting $\delta' = \frac{\delta}{SR|\mathcal{V}|}$ yields the desired result. $\qquad\square$

We next state the formal version of Theorem 4, which we wish to prove:

**Theorem 11.** *Let $\epsilon$ be a fixed constant. Assume that $\sigma$ is a degree $q$ polynomial, where $q = C_1/\epsilon$ for some $C_1 > 2$. Assume that $d \geq S^\epsilon, R^\epsilon$. Define $C(a) = |\{(s,r) : a^*(s,r) = a\}|$.*

*Let $(d, H, d_h, m)$ satisfy*

$$d \gtrsim (C_2 \log(|\mathcal{V}|/\delta)/\epsilon)^{C_3/\epsilon}$$
$$Hd_h \gtrsim (S + R) \log(|\mathcal{V}|/\delta)$$
$$m \gtrsim (C_2 \log(|\mathcal{V}|/\delta))^{C_3/\epsilon} \cdot \max_a C(a)$$
$$md \gtrsim (C_2 \log(|\mathcal{V}|/\delta))^{C_3/\epsilon} \cdot SR,$$

*Then, with probability $1 - \delta$, there exists a single-layer transformer $F_{\text{TF}}(\,\cdot\,; \boldsymbol{\theta}_{\text{TF}})$ with embedding dimension $d$, number of heads $H$, head dimension $d_h$, and MLP width $m$ such that*

$$\mathbb{P}_{z_{1:T+1} \sim \mathcal{D}}\left[\arg\max_{z \in \mathcal{V}} \boldsymbol{\varphi}(z)^\top F_{\text{TF}}(\boldsymbol{X}; \boldsymbol{\theta}_{\text{TF}}) = z_{T+1}\right] = 1.$$

**Remark.** Ignoring polylog factors, and treating $\epsilon$ as a constant, the constraints on the architecture size become

$$Hd_h \gtrsim S + R \quad \text{and} \quad m \gtrsim C(a) \quad \text{and} \quad md \gtrsim SR.$$

We first note that $C(a) \leq S$, and so $m \gtrsim S$ is sufficient. It is possible for $C(a)$ to be much smaller; on average we expect $C(a) \approx S/D$, and we also note that it is possible for $C(a) = 1$. The main constraint is that $md \gtrsim SR$, i.e that the number of MLP parameters scales linearly with the number of facts that need to be stored.

*Proof of Theorem 11.* Partition $\mathcal{S}$ into the sets $\mathcal{S}^{(1)}, \ldots, \mathcal{S}^{(N_S)}$ and $\mathcal{R}$ into the sets $\mathcal{R}^{(1)}, \ldots, \mathcal{R}^{(N_R)}$, such that $\left|\mathcal{S}^{(i)}\right|, \left|\mathcal{R}^{(j)}\right| \leq M$ and $N_S = \lceil \frac{S}{M} \rceil, N_R = \lceil \frac{R}{M} \rceil$. Assume that $d = \Theta(M \log(|\mathcal{V}|/\delta'))$

Let $H = \lceil d/d_h \rceil$. For each $i \in [N_S]$, we construct the $H'$ attention heads corresponding to $h \in \{(i-1)H' + 1, \ldots, iH'\}$ as follows. First, for all such $h$, let

$$\boldsymbol{W}_K^{(h)\top} \boldsymbol{W}_Q^{(h)} = \beta \sum_{z \in \mathcal{S}^{(i)}} \boldsymbol{\varphi}(z)\boldsymbol{\varphi}(\text{EOS})^\top + \frac{\beta}{2}\boldsymbol{\varphi}(\text{EOS})\boldsymbol{\varphi}(\text{EOS})^\top$$

for a large constant $\beta$. By an identical argument to as in Theorem 3, on the event that Lemma 1 holds we have that

$$\boldsymbol{X}^\top \mathcal{S}\left(\boldsymbol{X}\boldsymbol{W}_K^{(h)\top}\boldsymbol{W}_Q^{(h)}\boldsymbol{x}_T\right) = \begin{cases} \boldsymbol{\varphi}(\text{EOS}) & s \notin \mathcal{S}^{(i)} \\ \boldsymbol{\varphi}(s) & s \in \mathcal{S}^{(i)} \end{cases}.$$

The total contribution from these attention heads is then

$$\sum_{h=(i-1)H'+1}^{iH'} \boldsymbol{W}_O^{(h)\top} \text{attn}(\boldsymbol{X}; \boldsymbol{W}_K^{(h)}, \boldsymbol{W}_Q^{(h)}, \boldsymbol{W}_V^{(h)}) = \left(\sum_{h=(i-1)H'+1}^{iH'} \boldsymbol{W}_O^{(h)\top}\boldsymbol{W}_V^{(h)}\right) \cdot \begin{cases} \boldsymbol{\varphi}(\text{EOS}) & s \notin \mathcal{S}^{(i)} \\ \boldsymbol{\varphi}(s) & s \in \mathcal{S}^{(i)} \end{cases}$$

Since $H'd_h \geq d$, we can let $\sum_{h=(i-1)H'+1}^{iH'} \boldsymbol{W}_O^{(h)\top}\boldsymbol{W}_V^{(h)}$ be a projection onto a $\lceil d/2 \rceil$ dimensional subspace $P$, orthogonal to $\boldsymbol{\varphi}(\text{EOS})$, and thus

$$\sum_{h=(i-1)H'+1}^{iH'} \boldsymbol{W}_O^{(h)\top} \text{attn}(\boldsymbol{X}; \boldsymbol{W}_K^{(h)}, \boldsymbol{W}_Q^{(h)}, \boldsymbol{W}_V^{(h)}) = \begin{cases} 0 & s \notin \mathcal{S}^{(i)} \\ \Pi_P\boldsymbol{\varphi}(s) & s \in \mathcal{S}^{(i)} \end{cases}$$

Altogether, if the sequence $(z_1, \ldots, z_T)$ contains the subject $s$, then

$$\sum_{h=1}^{H'N_S} \boldsymbol{W}_O^{(h)\top} \text{attn}(\boldsymbol{X}; \boldsymbol{W}_K^{(h)}, \boldsymbol{W}_Q^{(h)}, \boldsymbol{W}_V^{(h)}) = \Pi_P\boldsymbol{\varphi}(s)$$

Similarly, if we let $Q$ be a $\lceil d/2 \rceil$ dimensional subspace orthogonal to $P$ and $\boldsymbol{\varphi}(\text{EOS})$, then we can construct the attention heads $h \in \{H'N_S + 1, \ldots, H'N_S + H'N_R\}$ such that

$$\sum_{h=H'N_S+1}^{H'N_S+H'N_R} \boldsymbol{W}_O^{(h)\top} \text{attn}(\boldsymbol{X}; \boldsymbol{W}_K^{(h)}, \boldsymbol{W}_Q^{(h)}, \boldsymbol{W}_V^{(h)}) = \Pi_Q\boldsymbol{\varphi}(r),$$

where $r$ is the relation in the sequence $(z_1, \ldots, z_T)$. Such a construction exists with probability $1 - (N_S + N_R)\delta'$. The total number of heads is

$$H = H'N_S + H'N_R \propto \frac{d(S+R)}{d_h M} \propto \frac{(S+R)\log(|\mathcal{V}|/\delta')}{d_h}.$$

The output of the self-attention component is then

$$F_{\text{MHSA}}(\boldsymbol{X}; \boldsymbol{\theta}) = \Pi_P\boldsymbol{\varphi}(s) + \Pi_Q\boldsymbol{\varphi}(r) = \tilde{\boldsymbol{\varphi}}(s) + \tilde{\boldsymbol{\varphi}}(r).$$

On the event that Lemma 3 holds, we have that there exists a two-layer neural network $F(\boldsymbol{z}) = \sum_{i \in [m]} \boldsymbol{v}_i \sigma(\boldsymbol{w}_i^\top \boldsymbol{z})$ of width $m$ such that

$$\arg\max_a \boldsymbol{\varphi}(a)^\top F(\boldsymbol{\varphi}(s) + \tilde{\boldsymbol{\varphi}}(r)) = a^*(s, r).$$

Scaling $\boldsymbol{V}$ by a large enough constant ensures that

$$\arg\max_{z \in \mathcal{V}} \boldsymbol{\varphi}(z)^\top F_{\text{TF}}(\boldsymbol{X}; \boldsymbol{\theta}_{\text{TF}}) = a^*(s, r).$$

Union bounding over all the high probability events and setting $\delta = \delta'/(N_S + N_R + 1)$ yields the desired result. $\qquad\square$

# D PROOFS FOR SECTION 5

## D.1 PRELIMINARIES

Recall that the parameters are $\boldsymbol{\theta} := \{\boldsymbol{W}_{OV}(a,z)\}_{a\in\mathcal{A},z\in\mathcal{V}} \cup \{\boldsymbol{W}_{KQ}(z)\}_{z\in\mathcal{V}}$, and that the cross entropy loss is

$$L(\boldsymbol{\theta}) := \mathbb{E}_{z_{1:T+1}}\left[-\langle\boldsymbol{\varphi}(z_{T+1}), F_{\text{lin}}(\boldsymbol{X};\boldsymbol{\theta})\rangle + \log\left(\sum_{a\in\mathcal{A}}\exp\left(\langle\boldsymbol{\varphi}(a), F_{\text{lin}}(\boldsymbol{X};\boldsymbol{\theta})\rangle\right)\right)\right]$$

where

$$\boldsymbol{\varphi}(a)^\top F_{\text{lin}}(\boldsymbol{X};\boldsymbol{\theta}) = \sum_{t=1}^T \boldsymbol{W}_{OV}(a,z_t)\boldsymbol{W}_{KQ}(z_t).$$

We consider running gradient flow:

$$\dot{\boldsymbol{\theta}} = -\nabla L(\boldsymbol{\theta})$$

from the initialization $\boldsymbol{W}_{OV}(a,z) = \alpha$, $\boldsymbol{W}_{KQ}(z) = \alpha\sqrt{|\mathcal{A}|+1}$ for some $\alpha > 0$.

We also define $\boldsymbol{\Theta}$ by

$$\boldsymbol{\Theta}(a,z) = \boldsymbol{W}_{KQ}(z)\boldsymbol{W}_{OV}(a,z),$$

and remark that the loss $L$ is convex in $\boldsymbol{\Theta}$.

**Lemma 4** (Balancedness). *Let $C(z_{1:T}, z)$ denote the number of tokens in $z_{1:T}$ equal to $z$. The loss gradients are given by*

$$\partial_{\boldsymbol{W}_{VO}(a,z)}L(\boldsymbol{\theta}) = -\boldsymbol{W}_{KQ}(z)\cdot\mathbb{E}_{z_{1:T}}[C(z_{1:T},z)\cdot(\mathbf{1}(a = a^*(z_{1:T})) - \hat{p}(a\mid z_{1:T}))]$$

$$\partial_{\boldsymbol{W}_{KQ}(z)}L(\boldsymbol{\theta}) = -\sum_a \boldsymbol{W}_{OV}(a,z)\cdot\mathbb{E}_{z_{1:T}}[C(z_{1:T},z)\cdot(\mathbf{1}(a = a^*(z_{1:T})) - \hat{p}(a\mid z_{1:T}))]$$

*As such, the quantity*

$$\boldsymbol{W}_{KQ}(z)^2 - \sum_{a\in\mathcal{A}}\boldsymbol{W}_{VO}(a,z)^2$$

*is constant throughout the gradient flow trajectory.*

*Proof.* We first see that

$$\partial_{\boldsymbol{W}_{VO}(a,z)}\left(\boldsymbol{\varphi}(a')^\top F_{\text{lin}}(\boldsymbol{X};\boldsymbol{\theta})\right) = \mathbf{1}(a = a')\cdot C(z_{1:T},z)\cdot\boldsymbol{W}_{KQ}(z),$$

Similarly,

$$\partial_{\boldsymbol{W}_{KQ}(z)}\left(\boldsymbol{\varphi}(a')^\top F_{\text{lin}}(\boldsymbol{X};\boldsymbol{\theta})\right) = C(z_{1:T},z)\cdot\boldsymbol{W}_{OV}(a',z).$$

Therefore

$$\partial_{\boldsymbol{W}_{VO}(a,z)}L(\boldsymbol{\theta})$$

$$= \boldsymbol{W}_{KQ}(z)\cdot\mathbb{E}\left[-\mathbf{1}(z_{T+1} = a)\cdot C(z_{1:T},z) + \frac{\sum_{a'\in\mathcal{A}}\exp\left(\langle\boldsymbol{\varphi}(a'), F_{\text{lin}}(\boldsymbol{X};\boldsymbol{\theta})\rangle\right)\cdot\mathbf{1}(a = a')\cdot C(z_{1:T},z)}{\sum_{a'\in\mathcal{A}}\exp\left(\langle\boldsymbol{\varphi}(a'), F_{\text{lin}}(\boldsymbol{X};\boldsymbol{\theta})\rangle\right)}\right]$$

$$= -\boldsymbol{W}_{KQ}(z)\cdot\mathbb{E}_{z_{1:T}}[C(z_{1:T},z)\cdot(\mathbf{1}(a = a^*(z_{1:T})) - \hat{p}(a\mid z_{1:T}))].$$

By a similar computation,

$$\partial_{\boldsymbol{W}_{KQ}(z)}L(\boldsymbol{\theta})$$

$$= \mathbb{E}_{z_{1:T}}\left[-\boldsymbol{W}_{OV}(z_{T+1},z)\cdot C(z_{1:T},z) + \sum_a \hat{p}(a\mid z_{1:T})\boldsymbol{W}_{OV}(a,z)\cdot C(z_{1:T},z)\right]$$

$$= \mathbb{E}_{z_{1:T}}\left[C(z_{1:T},z)\cdot\left(-\boldsymbol{W}_{OV}(a^*(z_{1:T}),z) + \sum_a \hat{p}(a\mid z_{1:T})\boldsymbol{W}_{OV}(a,z)\right)\right]$$

$$= -\sum_a \boldsymbol{W}_{OV}(a,z)\cdot\mathbb{E}_{z_{1:T}}[C(z_{1:T},z)\cdot(\mathbf{1}(a = a^*(z_{1:T})) - \hat{p}(a\mid z_{1:T}))].$$

Under gradient flow, we see that

$$\frac{1}{2}\frac{d}{dt}\left(\boldsymbol{W}_{KQ}(z)^2 - \sum_{a\in\mathcal{A}}\boldsymbol{W}_{VO}(a,z)^2\right)$$

$$= \boldsymbol{W}_{KQ}(z)\cdot\frac{d}{dt}\boldsymbol{W}_{KQ}(z) - \sum_{a\in\mathcal{A}}\boldsymbol{W}_{VO}(a,z)\cdot\frac{d}{dt}\boldsymbol{W}_{VO}(a,z)$$

$$= -\boldsymbol{W}_{KQ}(z)\cdot\partial_{\boldsymbol{W}_{KQ}(z)}L(\boldsymbol{\theta}) + \sum_{a\in\mathcal{A}}\boldsymbol{W}_{VO}(a,z)\cdot\partial_{\boldsymbol{W}_{VO}(a,z)}L(\boldsymbol{\theta})$$

$$= \boldsymbol{W}_{KQ}(z)\sum_a \boldsymbol{W}_{OV}(a,z)\cdot\mathbb{E}_{z_{1:T}}[C(z_{1:T},z)\cdot(\boldsymbol{1}(a=a^*(z_{1:T})) - \hat{p}(a\mid z_{1:T}))]$$

$$\qquad - \sum_{a\in\mathcal{A}}\boldsymbol{W}_{OV}(a,z)\boldsymbol{W}_{KQ}(z)\cdot\mathbb{E}_{z_{1:T}}[C(z_{1:T},z)\cdot(\boldsymbol{1}(a=a^*(z_{1:T})) - \hat{p}(a\mid z_{1:T}))]$$

$$= 0.$$

$\square$

**Corollary 3.** *Throughout the gradient flow trajectory, $\boldsymbol{W}_{KQ}(z) \geq \alpha$.*

*Proof.* At initialization, $\boldsymbol{W}_{KQ}(z)^2 - \sum_{a\in\mathcal{A}}\boldsymbol{W}_{VO}(a,z)^2 = \alpha^2$. Since this quantity is an invariant of gradient flow, it is impossible for $\boldsymbol{W}_{KQ}(z) = 0$, and thus $\boldsymbol{W}_{KQ}(z) > 0$ throughout the entire trajectory. Furthermore,

$$\boldsymbol{W}_{KQ}(z)^2 = \sum_{a\in\mathcal{A}}\boldsymbol{W}_{VO}(a,z)^2 + \alpha^2 \geq \alpha^2,$$

and thus $\boldsymbol{W}_{KQ}(z) \geq \alpha$. $\square$

### D.2 PROOF OF THEOREM 5

*Proof of Theorem 5.* Let us select

$$\epsilon \leq \min\left(\frac{1}{2}\alpha p(s,r)|\mathcal{A}|^{-1}T^{-2}|\mathcal{N}|^{-(T-3)}, \frac{1}{2}\alpha|\mathcal{A}|^{-1}S^{-1}R^{-1}\delta\right).$$

There exists a time $T_\epsilon$ such that for all $t \geq T_\epsilon$, $\|\nabla_{\boldsymbol{\theta}}L(\boldsymbol{\theta}(t))\| \leq \epsilon$. Let us set $t_\delta = T_\epsilon$. Now, consider some iterate $\boldsymbol{\theta} := \boldsymbol{\theta}(t)$ for $t \geq t_\delta$.

First, see that for $s \in \mathcal{S}$,

$$\partial_{\boldsymbol{W}_{OV}(a,s)}L(\boldsymbol{\theta}) = -\boldsymbol{W}_{KQ}(s)\cdot\mathbb{E}_{z_{1:T}}[C(z_{1:T},z)\cdot(\boldsymbol{1}(a=a^*(z_{1:T})) - \hat{p}(a\mid z_{1:T}))]$$

$$= -\boldsymbol{W}_{KQ}(s)\cdot p(s)\cdot\mathbb{E}_{z_{1:T}}[\boldsymbol{1}(a=a^*(z_{1:T})) - \hat{p}(a\mid z_{1:T})\mid s\in z_{1:T}].$$

Consider some $a \notin \mathcal{A}_s$. Then $\mathbb{E}_{z_{1:T}}[\boldsymbol{1}(a=a^*(z_{1:T}))\mid s\in z_{1:T}] = 0$, and thus

$$\partial_{\boldsymbol{W}_{OV}(a,s)}L(\boldsymbol{\theta}) = \boldsymbol{W}_{KQ}(s)\cdot p(s)\cdot\mathbb{E}_{z_{1:T}}[\hat{p}(a\mid z_{1:T})\mid s\in z_{1:T}]$$

$$= \boldsymbol{W}_{KQ}(s)\sum_{r\in\mathcal{R}}p(s,r)\cdot\mathbb{E}_{z_{1:T}}[\hat{p}(a\mid z_{1:T})\mid s,r\in z_{1:T}]$$

As such, since $\left|\partial_{\boldsymbol{W}_{OV}(a,s)}L(\boldsymbol{\theta})\right| \leq \epsilon$,

$$\mathbb{E}_{z_{1:T}}[\hat{p}(a\mid z_{1:T})\mid s,r\in z_{1:T}] \leq \epsilon\alpha^{-1}p(s,r)^{-1}.$$

By an identical argument, since $\left|\partial_{\boldsymbol{W}_{OV}(a,r)}L(\boldsymbol{\theta})\right| \leq \epsilon$, then for $a \notin \mathcal{A}_r$

$$\mathbb{E}_{z_{1:T}}[\hat{p}(a\mid z_{1:T})\mid s,r\in z_{1:T}] \leq \epsilon\alpha^{-1}p(s,r)^{-1}.$$

For any $a \neq a^*(s,r)$, either $a \notin \mathcal{A}_s$ or $a \notin \mathcal{A}_r$. Therefore $\mathbb{E}_{z_{1:T}}[\hat{p}(a\mid z_{1:T})\mid s,r\in z_{1:T}] \leq \epsilon\alpha^{-1}p(s,r)^{-1}$ for all $a \neq a^*(s,r)$, and thus

$$\mathbb{E}_{z_{1:T}}[\hat{p}(a^*(s,r)\mid z_{1:T})\mid s,r\in z_{1:T}] \geq 1 - \epsilon\alpha^{-1}p(s,r)^{-1}|\mathcal{A}|.$$

There are at most $T^2|\mathcal{N}|^{T-3}$ sequences $z_{1:T}$ containing $(s,r)$, each of which occurs with equal probability. Therefore

$$\hat{p}(a^*(s,r) \mid z_{1:T}) \geq 1 - T^2|\mathcal{N}|^{T-3} \cdot \epsilon\alpha^{-1}p(s,r)^{-1}|\mathcal{A}|$$

for all such $z_{1:T}$. Then, bounding $-\log(1-z) \leq 2z$ for $z \in [0, \frac{1}{2}]$,

$$\mathbb{E}[-\log\hat{p}(a^*(s,r) \mid z_{1:T}) \mid s, r \in z_{1:T}] \leq 2\mathbb{E}[1 - \hat{p}(a^*(s,r) \mid z_{1:T})) \mid s, r \in z_{1:T}]$$
$$\leq 2\epsilon\alpha^{-1}p(s,r)^{-1}|\mathcal{A}|.$$

Altogether, the loss is

$$\mathbb{E}[-\log\hat{p}(z_{T+1} \mid z_{1:T})] = \sum_{s,r} p(s,r) \cdot \mathbb{E}[-\log\hat{p}(a^*(s,r) \mid z_{1:T}) \mid s, r \in z_{1:T}]$$
$$\leq 2\epsilon\alpha^{-1}|\mathcal{A}|SR$$
$$\leq \delta,$$

as desired. □

## D.3 Sequential Learning

The goal of this section is to show that the model learns *sequentially*; first, the relation components grow, then the subject components grow. This is given formally by Theorem 6

We first prove that weights corresponding to the subject and noise tokens stay bounded during the beginning of the trajectory.

**Lemma 5.** *For $s \in \mathcal{S}$,*

$$\boldsymbol{W}_{KQ}(z) \leq \exp(2p(s)t) \cdot \alpha\sqrt{|\mathcal{A}| + 1}.$$

*Likewise, for $z \in \mathcal{N}$,*

$$\boldsymbol{W}_{KQ}(z) \leq \exp(2Tt/|\mathcal{N}|) \cdot \alpha\sqrt{|\mathcal{A}| + 1}.$$

*Proof.* Recall that the update for $\boldsymbol{W}_{KQ}(s)$ is

$$\dot{\boldsymbol{W}}_{KQ}(s) = p(s)\langle\boldsymbol{W}_{OV}(\cdot, s), p^*(\cdot \mid s) - \mathbb{E}_{z_{1:T}}[\hat{p}(\cdot \mid z_{1:T}) \mid s \in z_{1:T}]\rangle$$
$$\leq p(s)\|\boldsymbol{W}_{OV}(\cdot, s)\|\|p^*(\cdot \mid s) - \mathbb{E}_{z_{1:T}}[\hat{p}(\cdot \mid z_{1:T}) \mid s \in z_{1:T}]\|$$
$$\leq 2p(s)\|\boldsymbol{W}_{OV}(\cdot, s)\|$$
$$\leq 2p(s)\boldsymbol{W}_{KQ}(s)$$

Therefore by Gronwall's inequality,

$$\boldsymbol{W}_{KQ}(s) \leq \exp(2p(s)t) \cdot \alpha\sqrt{|\mathcal{A}| + 1}.$$

Similarly, the update for $\boldsymbol{W}_{KQ}(z)$ for $z \in \mathcal{N}$ is

$$\dot{\boldsymbol{W}}_{KQ}(z) = \langle\boldsymbol{W}_{OV}(\cdot, z), \mathbb{E}_{z_{1:T}}[C(z_{1:T}, z) \cdot (\mathbf{1}(\cdot = a^*(z_{1:T})) - \hat{p}(\cdot \mid z_{1:T}))]\rangle$$
$$\leq \|\boldsymbol{W}_{OV}(\cdot, z)\| \cdot \mathbb{E}[C(z_{1:T}, z)\|\mathbf{1}(\cdot = a^*(z_{1:T})) - \hat{p}(\cdot \mid z_{1:T})\|]$$
$$\leq 2\boldsymbol{W}_{OV}(\cdot, z)\mathbb{E}[C(z_{1:T}, z)]$$
$$\leq \frac{2T}{|\mathcal{N}|}\boldsymbol{W}_{KQ}(z).$$

Again by Gronwall's inequality,

$$\boldsymbol{W}_{KQ}(z) \leq \exp(2Tt/|\mathcal{N}|) \cdot \alpha\sqrt{|\mathcal{A}| + 1}.$$

□

The following lemma is our key result, and shows that, assuming that the subject and noise weights stay bounded, the relation weights grow until the output of the model approximates the best relation-only prediction.

**Lemma 6.** *Let $\alpha_{sm}, \epsilon > 0$ be arbitrary parameters satisfying*

$$\alpha_{sm}^2 T \le \frac{1}{150} \log \left( \frac{\epsilon^2}{\alpha^2(|\mathcal{A}|+1)} \right)^{-1} \cdot \min_r \|p^*(\cdot \mid r) - p_0\|.$$

$$\epsilon^2 \le \frac{1}{50(|\mathcal{A}|+1)} \cdot \min_r \|p^*(\cdot \mid r) - p_0\|$$

*For a target accuracy $\epsilon_{min} > 0$, define $T^*$ by*

$$T^* = \max_r p(r)^{-1} \|p^*(\cdot \mid r) - p_0\|^{-1} \log \left( \frac{\epsilon}{\alpha \sqrt{|\mathcal{A}|+1}} \right) + 100(|\mathcal{A}|+1) \log |\mathcal{A}| \epsilon^{-2} \epsilon_{min}^{-2}$$

*Assume that for $z \in \mathcal{S} \cup \mathcal{N}$ that $\boldsymbol{W}_{KQ}(z) \le \alpha_{sm}$. Then, there exists $t \le T^*$ such that*

$$\sum_r p(r)^2 \|p^*(\cdot \mid r) - \mathbb{E}_{z_{1:T}}[\hat{p}(\cdot \mid z_{1:T}) \mid r \in z_{1:T}]\|^2 \le \epsilon_{min}^2.$$

*Proof.* The proof proceeds in three stages. First, we bound the time required for the relation weights to escape the origin. Next, we prove that the relation weights stay large. Finally, we show convergence.

**Stage 1: Escaping the origin.** The gradient flow update on $\boldsymbol{W}_{OV}(a, r)$ is

$$\dot{\boldsymbol{W}}_{OV}(a, r) = \boldsymbol{W}_{KQ}(r) \cdot p(r)(p^*(a \mid r) - \mathbb{E}_{z_{1:T}}[\hat{p}(a \mid z_{1:T}) \mid r \in z_{1:T}])$$

We thus have

$$\left\| \dot{\boldsymbol{W}}_{OV}(\cdot, r) - \boldsymbol{W}_{KQ}(r) \cdot p(r)(p^*(\cdot \mid r) - p_0(a)) \right\| \le \boldsymbol{W}_{KQ}(r) \cdot p(r) \|\mathbb{E}_{z_{1:T}}[\hat{p}(\cdot \mid z_{1:T}) \mid r \in z_{1:T}] - p_0\|$$

Define $p_0 = \frac{1}{|\mathcal{A}|} \mathbf{1}_\mathcal{A}$. Observe that

$$\|\mathbb{E}_{z_{1:T}}[\hat{p}(\cdot \mid z_{1:T}) \mid r \in z_{1:T}] - p_0\| \le \mathbb{E}_{z_{1:T}}[\|[\hat{p}(a \mid z_{1:T}) - p_0\| \mid r \in z_{1:T}]$$

$$\le \mathbb{E}_{z_{1:T}} \left[ \sum_t \boldsymbol{W}_{KQ}(z_t) \|\boldsymbol{W}_{OV}(\cdot, z_t)\| \mid r \in z_{1:T} \right]$$

$$\le \boldsymbol{W}_{KQ}(r) \|\boldsymbol{W}_{OV}(\cdot, r)\| + T\alpha_{sm}^2$$

$$\le \boldsymbol{W}_{KQ}(r)^2 + T\alpha_{sm}^2.$$

Thus

$$\left\| \dot{\boldsymbol{W}}_{OV}(\cdot, r) - \boldsymbol{W}_{KQ}(r) \cdot p(r)(p^*(\cdot \mid r) - p_0(a)) \right\| \le p(r) \boldsymbol{W}_{KQ}(r) \left( \boldsymbol{W}_{KQ}(r)^2 + T\alpha_{sm}^2 \right)$$

Likewise,

$$\dot{\boldsymbol{W}}_{KQ}(r) = p(r) \langle \boldsymbol{W}_{OV}(\cdot, r), (p^*(\cdot \mid r) - \mathbb{E}_{z_{1:T}}[\hat{p}(a \mid z_{1:T}) \mid r \in z_{1:T}]) \rangle,$$

and thus

$$\left| \dot{\boldsymbol{W}}_{KQ}(r) - p(r) \langle \boldsymbol{W}_{OV}(\cdot, r), (p^*(\cdot \mid r) - p_0) \rangle \right| \le p(r) \|\boldsymbol{W}_{OV}(\cdot, r)\| \|\mathbb{E}_{z_{1:T}}[\hat{p}(\cdot \mid z_{1:T}) \mid r \in z_{1:T}] - p_0\|$$

$$\le p(r) \boldsymbol{W}_{KQ}(r) \left( \boldsymbol{W}_{KQ}(r)^2 + T\alpha_{sm}^2 \right)$$

Define the vector $\boldsymbol{u} \in \mathbb{R}^2$ by

$$\boldsymbol{u} = \begin{bmatrix} \boldsymbol{W}_{KQ}(r) \\ \langle \boldsymbol{W}_{OV}(\cdot, r), \frac{p^*(\cdot \mid r) - p_0}{\|p^*(\cdot \mid r) - p_0\|} \rangle \end{bmatrix}$$

We see that

$$\left\| \dot{\boldsymbol{u}} - p(r) \|p^*(\cdot \mid r) - p_0\| \cdot \begin{bmatrix} 0 & 1 \\ 1 & 0 \end{bmatrix} \boldsymbol{u} \right\| \le 2p(r) \boldsymbol{W}_{KQ}(r) \left( \boldsymbol{W}_{KQ}(r)^2 + T\alpha_{sm}^2 \right)$$

Therefore

$$\frac{d}{dt}(\|\boldsymbol{u}\|^2) \le 2\langle \dot{\boldsymbol{u}}, \boldsymbol{u}\rangle$$

$$\le 2p(r)\|p^*(\cdot \mid r) - p_0\|\|\boldsymbol{u}\|^2 + 4p(r)\|\boldsymbol{u}\|\boldsymbol{W}_{KQ}(r)\big(\boldsymbol{W}_{KQ}(r)^2 + T\alpha_{sm}^2\big)$$

$$\le 2p(r)\|p^*(\cdot \mid r) - p_0\|\|\boldsymbol{u}\|^2 + 4p(r)\|\boldsymbol{u}\|^2\Big(\|\boldsymbol{u}\|^2 + T\alpha_{sm}^2\Big)$$

$$\le 2p(r)\big(\|p^*(\cdot \mid r) - p_0\| + 2T\alpha_{sm}^2\big)\|\boldsymbol{u}\|^2 + 4p(r)\|\boldsymbol{u}\|^4.$$

where the last inequality bounds $\boldsymbol{W}_{KQ}^2(r) \le \|\boldsymbol{u}\|^2$.

Define $\gamma_r := 2p(r)\big(\|p^*(\cdot \mid r) - p_0\| + 2T\alpha_{sm}^2\big)$. By Lemma 7 we have that for

$$t < \gamma_r^{-1} \log\left(\frac{\gamma_r}{4p(r)\|\boldsymbol{u}_0\|^2} + 1\right),$$

$$\|\boldsymbol{u}\|^2 \le \frac{\gamma_r\|\boldsymbol{u}_0\|^2 \exp(\gamma_r t)}{\gamma_r + 4p(r)^2(1 - \exp(\gamma_r t))}$$

Let $T_\epsilon$ be the first time that $\|\boldsymbol{u}\| \ge \epsilon$. If $T_\epsilon < \gamma_r^{-1} \log\left(\frac{\gamma_r}{4p(r)}\|\boldsymbol{u}_0\|^2 + 1\right)$, then

$$\epsilon^2 \le \|\boldsymbol{u}\|^2 \le \frac{\gamma_r\|\boldsymbol{u}_0\|^2 \exp(\gamma_r T_\epsilon)}{\gamma_r + 4p(r)^2(1 - \exp(\gamma_r T_\epsilon))} \le \frac{\gamma_r \alpha^2(|\mathcal{A}| + 1)\exp(\gamma_r T_\epsilon)}{\gamma_r + 4p(r)^2(1 - \exp(\gamma_r T_\epsilon))}.$$

Therefore

$$T_\epsilon \ge \gamma_r^{-1} \log\left(\frac{\epsilon^2\gamma_r + 4p(r)\epsilon^2\alpha^2(|\mathcal{A}| + 1)}{\alpha^2(|\mathcal{A}| + 1)\gamma_r + 4p(r)\epsilon^2\alpha^2(|\mathcal{A}| + 1)}\right) \ge \gamma_r^{-1} \log\left(\frac{\epsilon^2}{2\alpha^2(|\mathcal{A}| + 1)}\right)$$

for $\epsilon^2 \le \frac{\gamma_r}{4p(r)}$ On this assumption, $\frac{\epsilon^2}{2\alpha^2(|\mathcal{A}|+1)} \le \frac{\gamma_r}{4p(r)\|\boldsymbol{u}_0\|^2}$, and thus we always have $T_\epsilon \ge \gamma_r^{-1} \log\left(\frac{\epsilon^2}{2\alpha^2(|\mathcal{A}|+1)}\right)$.

Define $L_r$ by

$$L_r(\boldsymbol{\theta}) := p(r)\mathbb{E}_{z_{1:T+1}}\left[-\langle\boldsymbol{\varphi}(z_{T+1}), F_{\text{lin}}(\boldsymbol{X};\boldsymbol{\theta})\rangle + \log\left(\sum_{a\in\mathcal{A}} \exp\left(\langle\boldsymbol{\varphi}(a), F_{\text{lin}}(\boldsymbol{X};\boldsymbol{\theta})\rangle\right)\right) \mid r \in z_{1:T}\right]$$

Let us define the relation-only model as

$$\boldsymbol{\varphi}(a)^\top F_{\text{rel}}(\boldsymbol{X};\boldsymbol{\theta}) = \boldsymbol{W}_{OV}(a, r)\boldsymbol{W}_{KQ}(r)$$

where $r \in z_{1:T}$. We see that

$$\left|\boldsymbol{\varphi}(a)^\top F_{\text{rel}}(\boldsymbol{X};\boldsymbol{\theta}) - \boldsymbol{\varphi}(a)^\top F_{\text{lin}}(\boldsymbol{X};\boldsymbol{\theta})\right| \le (T - 1)\alpha_{sm}^2.$$

Define $g : \mathbb{R}^{|\mathcal{A}|} \to \mathbb{R}$ by $g(\boldsymbol{z}) = \log\left(\sum_a \exp(\boldsymbol{z}_a)\right)$. We see that $\nabla_{\boldsymbol{z}} g(\boldsymbol{z}) = \mathcal{S}(\boldsymbol{z})$, where $\mathcal{S}$ is the softmax, and thus

$$|g(\boldsymbol{z}_1) - g(\boldsymbol{z}_2)| \le \sup_{\boldsymbol{z}} \|\nabla_{\boldsymbol{z}} g(\boldsymbol{z})\|_1 \cdot \|g(\boldsymbol{z}_1) - g(\boldsymbol{z}_2)\|_\infty \le \|g(\boldsymbol{z}_1) - g(\boldsymbol{z}_2)\|_\infty.$$

Therefore defining the relation-only loss $\bar{L}_r$ as

$$\bar{L}_r(\boldsymbol{\theta}) := p(r)\mathbb{E}_s\left[-\langle\boldsymbol{\varphi}(a^*(s, r)), F_{\text{rel}}(r;\boldsymbol{\theta})\rangle + \log\left(\sum_{a\in\mathcal{A}} \exp\left(\langle\boldsymbol{\varphi}(a), F_{\text{lin}}(r;\boldsymbol{\theta})\rangle\right)\right)\right]$$

$$= -p(r)\sum_a p(a \mid r)\boldsymbol{W}_{OV}(a^*(s, r), r)\boldsymbol{W}_{KQ}(r) + p(r)\log\left(\sum_{a\in\mathcal{A}} \exp\left(\boldsymbol{W}_{OV}(a, r)\boldsymbol{W}_{KQ}(r)\right)\right),$$

we see that

$$\left|L_r(\boldsymbol{\theta}) - \bar{L}_r(\boldsymbol{\theta})\right| \le 2(T-1)\alpha_{sm}^2.$$

Since log-sum-exp is 1-strongly-convex, recalling that $\boldsymbol{\Theta}(a, r) := \boldsymbol{W}_{OV}(a, r)\boldsymbol{W}_{KQ}(r)$,

$$\log\left(\sum_a \exp(\boldsymbol{\Theta}(a,r))\right) \le \log(|\mathcal{A}|) + \sum_a \frac{1}{|\mathcal{A}|}\boldsymbol{\Theta}(a,r) + \frac{1}{2}\|\boldsymbol{\Theta}(\cdot,r)\|^2.$$

Therefore

$$\bar{L}_r \le p(r)\log|\mathcal{A}| - p(r)\langle\boldsymbol{\Theta}(\cdot,r), p^*(\cdot \mid r) - p_0\rangle + \frac{1}{2}p(r)\|\boldsymbol{\Theta}(\cdot,r)\|^2$$

$$= L_{r,0} - p(r)\|p^*(\cdot \mid r) - p_0\| \cdot \boldsymbol{u}_1\boldsymbol{u}_2 + \frac{1}{2}p(r)\|\boldsymbol{\Theta}(\cdot,r)\|^2.$$

We next track the evolution of $\boldsymbol{u}_1\boldsymbol{u}_2$:

$$\frac{d}{dt}(\boldsymbol{u}_1\boldsymbol{u}_2) = \dot{\boldsymbol{u}}_1\boldsymbol{u}_2 + \boldsymbol{u}_1\dot{\boldsymbol{u}}_2$$

$$\ge p(r)\|p^*(\cdot \mid r) - p_0\|\|\boldsymbol{u}\|^2 - 4p(r)\|\boldsymbol{u}\|^2\left(\|\boldsymbol{u}\|^2 + (T-1)\alpha_{sm}^2\right)$$

$$\ge p(r)\left(\|p^*(\cdot \mid r) - p_0\| - 4\|\boldsymbol{u}\|^2 - 4(T-1)\alpha_{sm}^2\right)\|\boldsymbol{u}\|^2$$

$$\ge p(r)\left(\|p^*(\cdot \mid r) - p_0\| - 4T\alpha_{sm}^2\right)\|\boldsymbol{u}\|^2 - 4p(r)\|\boldsymbol{u}\|^4.$$

for $t \le T_\epsilon$. Since $\|\boldsymbol{u}\| \le \epsilon$, this is increasing in $\|\boldsymbol{u}\|$.

We first have the bound $\|\boldsymbol{u}\|^2 \ge \boldsymbol{W}_{KQ}(r)^2 \ge \alpha^2$. Next, we have the bound $\|\boldsymbol{u}\|^2 \ge 2\boldsymbol{u}_1\boldsymbol{u}_2$. Pick some time $\tau \le T_\epsilon$. Define $\gamma_r^- := 2p(r)\left(\|p^*(\cdot \mid r) - p_0\| - 4T\alpha_{sm}^2\right)$. We see that

$$(\boldsymbol{u}_1\boldsymbol{u}_2)(\tau) \ge \left(\frac{1}{2}\gamma_r^-\alpha^2 - 4p(r)\alpha^4\right)\tau \ge \frac{1}{4}\gamma_r^-\alpha^2\tau$$

Next, by Lemma 7, for $t \le T_\epsilon$ we have

$$(\boldsymbol{u}_1\boldsymbol{u}_2)(t) \ge \frac{\gamma_r^-(\boldsymbol{u}_1\boldsymbol{u}_2)(\tau)\exp(\gamma_r^-(t-\tau))}{\gamma_r^- + 8p(r)(\boldsymbol{u}_1\boldsymbol{u}_2)(\tau)\exp\left(\gamma_r^-(t-\tau)\right)}.$$

Plugging in $t = \gamma_r^{-1}\log\left(\frac{\epsilon^2}{2\alpha^2(|\mathcal{A}|+1)}\right)$,

$$(\boldsymbol{u}_1\boldsymbol{u}_2)(\tau)\exp\left(\gamma_r^-(t-\tau)\right) \ge \frac{1}{4}\gamma_r^-\alpha^2\tau\exp\left(-\gamma_r^-\tau\right) \cdot \exp\left(\frac{\gamma_r^-}{\gamma_r}\log\left(\frac{\epsilon^2}{2\alpha^2(|\mathcal{A}|+1)}\right)\right)$$

Selecting $\tau = 1/\gamma_r^-$, we get

$$(\boldsymbol{u}_1\boldsymbol{u}_2)(\tau)\exp\left(\gamma_r^-(t-\tau)\right) \ge \frac{\alpha^2}{4e} \cdot \exp\left(\frac{\|p^*(\cdot \mid r) - p_0\| - 4T\alpha_{sm}^2}{\|p^*(\cdot \mid r) - p_0\| + 2T\alpha_{sm}^2}\log\left(\frac{\epsilon^2}{2\alpha^2(|\mathcal{A}|+1)}\right)\right)$$

$$\ge \frac{\alpha^2}{4e} \cdot \frac{\epsilon^2}{2\alpha^2(|\mathcal{A}|+1)} \cdot \exp\left(\frac{-6T\alpha_{sm}^2}{\|p^*(\cdot \mid r) - p_0\| + 2T\alpha_{sm}^2}\log\left(\frac{\epsilon^2}{2\alpha^2(|\mathcal{A}|+1)}\right)\right)$$

$$\ge \frac{\alpha^2}{4e} \cdot \frac{\epsilon^2}{2\alpha^2(|\mathcal{A}|+1)} \cdot \left(1 - \frac{6T\alpha_{sm}^2}{\|p^*(\cdot \mid r) - p_0\| + 2T\alpha_{sm}^2}\log\left(\frac{\epsilon^2}{2\alpha^2(|\mathcal{A}|+1)}\right)\right)$$

$$\ge \frac{\epsilon^2}{50(|\mathcal{A}|+1)}.$$

whenever $\frac{T\alpha_{sm}^2}{\|p^*(\cdot|r) - p_0\| + 2T\alpha_{sm}^2}\log\left(\frac{\epsilon^2}{\alpha^2(|\mathcal{A}|+1)}\right) \le \frac{1}{150}$.

Therefore

$$(\boldsymbol{u}_1 \boldsymbol{u}_2)(t) \geq \frac{\epsilon^2}{50(|\mathcal{A}|+1)} \cdot \frac{1}{1 + 8\frac{p(r)}{\gamma_r^-}\frac{\epsilon^2}{50(|\mathcal{A}|+1)}}$$

$$\geq \frac{\epsilon^2}{50(|\mathcal{A}|+1)} \cdot \frac{1}{1 + \frac{4\epsilon^2}{\|p^*(\cdot|r)-p_0\|50(|\mathcal{A}|+1)}}$$

$$\geq \frac{\epsilon^2}{100(|\mathcal{A}|+1)},$$

Altogether, we get that the loss is

$$\bar{L}_r \leq L_{r,0} - p(r)\|p^*(\cdot \mid r) - p_0\| \cdot \frac{\epsilon^2}{100(|\mathcal{A}|+1)} + \frac{1}{2}p(r)\epsilon^4$$

$$\leq L_{r,0} - p(r)\|p^*(\cdot \mid r) - p_0\| \cdot \frac{\epsilon^2}{200(|\mathcal{A}|+1)}$$

whenever $\epsilon^2 \leq \frac{\|p^*(\cdot|r)-p_0\|}{100(|\mathcal{A}|+1)}$.

**Stage 2: Norm stays large**   Next, we want to show that $\boldsymbol{W}_{KQ}(r)$ stays large. We first show that the relation-only loss $\bar{L}$ is decreasing. We can compute that

$$\frac{d}{dt}\bar{L}_r(\boldsymbol{\theta}) = \langle \nabla_{\boldsymbol{\theta}}\bar{L}_r, \nabla_{\boldsymbol{\theta}}L_r \rangle$$

Define $\hat{p}(\cdot \mid r)$ by

$$\hat{p}(a \mid r) = \frac{\exp(\boldsymbol{W}_{KQ}(r)\boldsymbol{W}_{OV}(a,r))}{\sum_{a'}\exp(\boldsymbol{W}_{KQ}(r)\boldsymbol{W}_{OV}(a',r))}.$$

We observe that

$$\partial_{\boldsymbol{W}_{OV}(\cdot,r)}\bar{L}_r = \boldsymbol{W}_{KQ}(r)p(r)(p^*(\cdot \mid r) - \hat{p}(\cdot \mid r))$$

$$\partial_{\boldsymbol{W}_{OV}(\cdot,r)}L_r = \boldsymbol{W}_{KQ}(r)p(r)(p^*(\cdot \mid r) - \mathbb{E}_{z_{1:T}}[\hat{p}(\cdot \mid z_{1:T}) \mid r \in z_{1:T}])$$

and thus

$$\left\|\partial_{\boldsymbol{W}_{OV}(\cdot,r)}\bar{L}_r - \partial_{\boldsymbol{W}_{OV}(\cdot,r)}L_r\right\| \leq \boldsymbol{W}_{KQ}(r)p(r)\|\mathbb{E}_{z_{1:T}}[\hat{p}(\cdot \mid z_{1:T}) \mid r \in z_{1:T}] - \hat{p}(\cdot \mid r)\|$$

$$\leq \boldsymbol{W}_{KQ}(r)p(r)T\alpha_{sm}^2.$$

Likewise,

$$\left|\partial_{\boldsymbol{W}_{KQ}(r)}\bar{L}_r - \partial_{\boldsymbol{W}_{KQ}(r)}L_r\right| = p(r)|\langle \boldsymbol{W}_{OV}(\cdot,r), \mathbb{E}_{z_{1:T}}[\hat{p}(\cdot \mid z_{1:T}) \mid r \in z_{1:T}] - \hat{p}(\cdot \mid r)\rangle|$$

$$\leq p(r)\|\boldsymbol{W}_{OV}(\cdot,r)\|\|\mathbb{E}_{z_{1:T}}[\hat{p}(\cdot \mid z_{1:T}) \mid r \in z_{1:T}] - \hat{p}(\cdot \mid r)\|$$

$$\leq \boldsymbol{W}_{KQ}(r)p(r)T\alpha_{sm}^2.$$

Therefore

$$\frac{d}{dt}L_r(\boldsymbol{\theta}) \geq \left\|\nabla_{\boldsymbol{\theta}}\bar{L}_r\right\|^2 - \left\|\nabla_{\boldsymbol{\theta}}\bar{L}_r\right\|\left\|\nabla_{\boldsymbol{\theta}}\bar{L}_r - \nabla_{\boldsymbol{\theta}}L_r\right\|$$

$$\geq \left\|\nabla_{\boldsymbol{\theta}}\bar{L}_r\right\|^2 - \left\|\nabla_{\boldsymbol{\theta}}\bar{L}_r\right\|\sqrt{2}\boldsymbol{W}_{KQ}(r)p(r)T\alpha_{sm}^2.$$

Assume that $\frac{d}{dt}L_r(\boldsymbol{\theta}) < 0$. Then

$$\sqrt{2}\boldsymbol{W}_{KQ}(r)p(r)T\alpha_{sm}^2 \geq \left\|\nabla_{\boldsymbol{\theta}}\bar{L}_r\right\| \geq \boldsymbol{W}_{KQ}(r)p(r)\|p^*(\cdot \mid r) - \hat{p}(\cdot \mid r)\|,$$

i.e

$$\|p^*(\cdot \mid r) - \hat{p}(\cdot \mid r)\| \leq \sqrt{2}T\alpha_{sm}^2.$$

Assuming that $\sqrt{2}T\alpha_{sm}^2 < \frac{1}{2S}$, since $p^*(a \mid r) > \frac{1}{S}$ for $p * (a \mid r) > 0$ we have that

$$\left| p^*(a \mid r) \log \frac{p^*(a \mid r)}{\hat{p}(a \mid r)} \right| = p^*(a \mid r) \left| \log \left( 1 + \frac{\hat{p}(a \mid r) - p^*(a \mid r)}{p^*(a \mid r)} \right) \right| \le 2|\hat{p}(a \mid r) - p^*(a \mid r)|$$

Therefore $\bar{L}_r - p(r)H(p^*(\cdot \mid r)) \le 2p(r)\|\hat{p}(\cdot \mid r) - p^*(\cdot \mid r)\|_1 \le \sqrt{2}T\alpha_{sm}^2 D$. As such, we have that $\bar{L}_r$ stays below $L_{r,0} - p(r)\|p^*(\cdot \mid r) - p_0\| \cdot \frac{\epsilon^2}{200(|\mathcal{A}|+1)}$ for the remainder of the gradient flow trajectory.

By convexity in $\boldsymbol{\Theta}$ space,

$$\begin{aligned} L_{r,0} - \bar{L}_r(\boldsymbol{\Theta}) &\le -\langle \nabla_{\boldsymbol{\Theta}(\cdot,r)}\bar{L}(\boldsymbol{0}), \boldsymbol{\Theta}(\cdot, r)\rangle \\ &\le \left\| \nabla_{\boldsymbol{\Theta}(\cdot,r)}\bar{L}(\boldsymbol{0}) \right\| \|\boldsymbol{\Theta}(\cdot, r)\| \\ &\le p(r) \cdot \|p^*(\cdot \mid r) - p_0\| \cdot \|\boldsymbol{\Theta}(\cdot, r)\|. \end{aligned}$$

Therefore

$$\|\boldsymbol{\Theta}(\cdot, r)\| \ge \frac{\epsilon^2}{100(|\mathcal{A}| + 1)}$$

**Stage 3: Convergence.** Next, we can bound the loss decrease by

$$\begin{aligned} \frac{d}{dt}L(\boldsymbol{\theta}) &= -\|\nabla_{\boldsymbol{\theta}}L\|^2 \\ &\le -\sum_r \left\| \partial_{\boldsymbol{W}_{OV}(\cdot,r)}L \right\|^2 \\ &= -\sum_r \left\| \partial_{\boldsymbol{W}_{OV}(\cdot,r)}L_r \right\|^2 \\ &= -\sum_r \boldsymbol{W}_{KQ}(r)^2 p(r)^2 \|p^*(\cdot \mid r) - \mathbb{E}_{z_{1:T}}[\hat{p}(\cdot \mid z_{1:T}) \mid r \in z_{1:T}]\| \\ &\le -\sum_r \|\boldsymbol{\Theta}(\cdot, r)\| p(r)^2 \|p^*(\cdot \mid r) - \mathbb{E}_{z_{1:T}}[\hat{p}(\cdot \mid z_{1:T}) \mid r \in z_{1:T}]\|^2 \\ &\le -\frac{\epsilon^2}{200(|\mathcal{A}| + 1)} \sum_r p(r)^2 \|p^*(\cdot \mid r) - \mathbb{E}_{z_{1:T}}[\hat{p}(\cdot \mid z_{1:T}) \mid r \in z_{1:T}]\|^2 \end{aligned}$$

Since $L(\boldsymbol{\Theta}) \le L(\boldsymbol{\Theta}_0) = \log|\mathcal{A}|$, and $L(\boldsymbol{\Theta}) \ge 0$, there exists some $t \le \max_r t^* + 100(|\mathcal{A}| + 1)\log|\mathcal{A}|\epsilon^{-2}\epsilon_{min}^{-2}$ such that

$$\sum_r p(r)^2\|p^*(\cdot \mid r) - \mathbb{E}_{z_{1:T}}[\hat{p}(\cdot \mid z_{1:T}) \mid r \in z_{1:T}]\|^2 \le \epsilon_{min}^2,$$

as desired. $\qquad\square$

To conclude, we must set $\alpha, \alpha_{sm}, T^*$ appropriately in terms of $\epsilon$ in order to apply Lemma 6.

*Proof of Theorem 6.* Let $\epsilon = \epsilon_{min} \le \frac{1}{100(|\mathcal{A}|+1)} \cdot \min_r \|p^*(\cdot \mid r) - p_0\|$ be the target accuracy. Let us choose the initialization $\alpha$ so that $\log\left( \frac{\epsilon}{\alpha\sqrt{|\mathcal{A}|+1}} \right) = \iota$, where $\iota$ is chosen so that

$$\iota \ge 100(|\mathcal{A}| + 1)\log|\mathcal{A}|\epsilon^{-4}p(r)\|p^*(\cdot \mid r) - p_0\|.$$

In this case, we see that

$$T^* \le 2\iota \max_r \left( p(r)^{-1}\|p^*(\cdot \mid r) - p_0\|^{-1} \right).$$

Since $p^*(\cdot \mid r)$ is supported on at most $D$ elements, and $|\mathcal{A}| \geq 2D$, we have $\|p^*(\cdot \mid r) - p_0\|^{-1} \leq \sqrt{2D}$. Therefore $T^* \leq 2R\sqrt{D} \cdot \iota$. Let us compute $\alpha_{sm}$. We see that, since $S \geq 8R\sqrt{2D}$,

$$
\begin{aligned}
\boldsymbol{W}_{KQ}(s) &\leq \exp\left(\frac{4R\sqrt{D}}{S}\iota\right) \cdot \alpha\sqrt{|A|+1} \\
&\leq \exp\left(\frac{1}{2}\iota\right)\alpha\sqrt{|A|+1} \\
&= \sqrt{\epsilon\alpha} \cdot (|\mathcal{A}|+1)^{1/4} \\
&\leq \sqrt{\alpha}
\end{aligned}
$$

Similarly, since $\mathcal{N} \geq 4R\sqrt{2D}T$,

$$
\begin{aligned}
\boldsymbol{W}_{KQ}(s) &\leq \exp\left(\frac{4R\sqrt{D}T}{|\mathcal{N}|}\iota\right) \cdot \alpha\sqrt{|A|+1} \\
&\leq \exp\left(\frac{1}{2}\iota\right)\alpha\sqrt{|A|+1} \\
&\leq \sqrt{\alpha}
\end{aligned}
$$

Therefore the assumption holds for $\alpha_{sm} = \sqrt{\alpha}$. To conclude, we must verify that

$$
T\alpha \leq \frac{1}{300}\iota^{-1} \cdot \min_r \|p^*(\cdot \mid r) - p_0\|.
$$

But since $\alpha = \frac{\epsilon}{\sqrt{|\mathcal{A}|+1}}e^{-\iota}$, the RHS scales with $e^{-\iota}$ and the RHS scales with $\iota$, and thus the condition can be obtained for choosing $\iota$ sufficiently large.

Under the setting of paramters the conditions of Lemma 6 are satisfied, and thus the claim holds. $\square$

### D.4 HELPER LEMMA

**Lemma 7.** *Let $z(t) \geq 0$ satisfy*

$$
\dot{z} \leq Az + Bz^2.
$$

*for positive constants $A, B$. Then*

$$
z(t) \leq \frac{Az(0)e^{At}}{A + Bz(0)(1 - e^{At})}.
$$

*Furthermore, if*

$$
\dot{z} \geq Az - Bz^2,
$$

*and $z \in [0, \frac{A}{2B}]$ on the interval $[0, T]$, then*

$$
z(t) \geq \frac{Az(0)e^{At}}{A + Bz(0)(e^{At} - 1)}.
$$

*for all $t \in [0, T]$.*

Both claims follow from the Bihari-LaSalle inequality.

## E PROOFS FROM SECTION 6

### E.1 ASSOCIATIVE MEMORIES

*Proof of Theorem 7.* $f^* \to \boldsymbol{F} \to \hat{f}$ is a Markov chain, so by the data processing inequality,

$$
I(f^*; \hat{f}) \leq I(f^*; \boldsymbol{F}).
$$

Also, by definition of mutual information

$$I(f^*; \boldsymbol{F}) \leq H(\boldsymbol{F}) \leq B,$$

where the last inequality follows since $\boldsymbol{F}$ is an $B$-bit message. Thus $I(f^*; \hat{f}) \leq B$.

Let $q_x(\cdot \mid \hat{f})$ be the conditional distribution of $f^*(x)$ given $\hat{f}$. Consider some fixed $\hat{f}$. $\hat{f}(x)$ is also a probability distribution over $[M]$, and thus by Gibbs' inequality

$$\mathbb{E}_{y \sim q_x(\cdot|\hat{f})}\left[-\log \hat{f}(x)_y\right] \geq \mathbb{E}_{y \sim q_x(\cdot|\hat{f})}\left[-\log q_x(y \mid \hat{f})\right].$$

Therefore, letting $q$ be the marginal distribution over $\hat{f}$ and $q_x$ the marginal over $f^*(x)$,

$$\begin{aligned}
\mathbb{E}_{f^*, \hat{f}}\left[-\log \hat{f}(x)_{f^*(x)}\right] &= \mathbb{E}_{\hat{f}}\left[\mathbb{E}_{y \sim q_x(\cdot|\hat{f})}\left[-\log \hat{f}(x)_y\right]\right] \\
&\geq \mathbb{E}_{\hat{f}}\left[\mathbb{E}_{y \sim q_x(\cdot|\hat{f})}\left[-\log q_x(y \mid \hat{f})\right]\right] \\
&= \mathbb{E}_{f^*, \hat{f}}\left[-\log q_x(f^*(x) \mid \hat{f})\right] \\
&= \mathbb{E}_{f^*, \hat{f}}\left[-\log \frac{q_x(f^*(x), \hat{f})}{q(\hat{f})q_x(f^*(x))} - \log q_x(f^*(x))\right] \\
&= -I(f^*(x); \hat{f}) + \log M.
\end{aligned}$$

where in the last step we use the fact that $q_x$ is uniform over $[M]$, and plug in the definition of mutual information. The total loss is thus

$$\mathbb{E}_{f^*, \hat{f}}\left[L(\hat{f})\right] \geq \sum_{x \in [N]} p(x)\left(-I(f^*(x); \hat{f}) + \log M\right). \tag{21}$$

Since the $y_i$ are independent,

$$B \geq I(f^*; \hat{f}) \geq \sum_{x \in [N]} I(f^*(x); \hat{f}).$$

Also, $0 \leq I(f^*(x); \hat{f}) \leq H(f^*(x)) = \log M$. Therefore equation 21 is minimized when $I(f^*(x); \hat{f}) = \log M$ for the $B/\log M$ most frequent tokens. Altogether,

$$\mathbb{E}_{f^*, \hat{f}}\left[L(\hat{f})\right] \geq \log M \cdot \sum_{x > \lceil \frac{B}{\log M} \rceil} p(x).$$

$\square$

*Proof of Corollary 1.* Let $p(x) = Z_\alpha x^{-\alpha}$, where $Z_\alpha = \sum_{x \in [N]} x^{-\alpha}$. We can bound

$$\sum_{x=k}^{n} p(x) = \frac{\sum_{x=k}^{n} x^{-\alpha}}{\sum_{x=1}^{n} x^{-\alpha}} \asymp k^{1-\alpha}.$$

Therefore

$$\mathbb{E}_{f^*, \hat{f}}\left[L(\hat{f})\right] \geq \log M \cdot \sum_{x > \lceil \frac{B}{\log M} \rceil} p(x) \gtrsim \log M \left(\frac{B}{\log M}\right)^{1-\alpha} \gtrsim B^{1-\alpha}.$$

$\square$

## E.2 FACTUAL RECALL

*Proof.* Define $\ell(s, r) := \mathbb{E}_{\mathcal{D}, \hat{f}}\left[-\log \hat{f}(s, r)_{a^*(s,r)}\right]$ so that

$$\mathbf{L} = p(s, r) \cdot \ell(s, r).$$

Let us define the expanded dataset $\mathcal{D} := \{\mathcal{A}_r\}_{r \in \mathcal{R}} \cup \{a^*(s,r)\}_{s \in \mathcal{S}, r \in \mathcal{R}}$. We observe that $\mathcal{D} \to a^* \to \boldsymbol{F} \to \hat{f}$ is a Markov chain, and thus by the data processing inequality

$$B \geq I(\mathcal{D}; \hat{f}).$$

Next, by the chain rule, we can decompose

$$
\begin{aligned}
I(\mathcal{D}; \hat{f}) &= I(\mathcal{A}_1, \ldots, \mathcal{A}_R; \hat{f}) + I(a^*; \hat{f} \mid \mathcal{A}_1, \ldots, \mathcal{A}_R) \\
&\geq I(\mathcal{A}_1, \ldots, \mathcal{A}_R; \hat{f}) + \sum_{s,r} I(a^*(s,r); \hat{f} \mid \mathcal{A}_1, \ldots, \mathcal{A}_R) \\
&= I(\mathcal{A}_1, \ldots, \mathcal{A}_R; \hat{f}) + \sum_{s,r} I(a^*(s,r); \hat{f} \mid \mathcal{A}_r),
\end{aligned}
$$

where the first inequality uses the fact that the $a^*(s,r)$ are conditionally independent given the $\mathcal{A}_r$, and the second uses that $a^*(s,r)$ is independent of $\mathcal{A}_{r'}$ given $\mathcal{A}_r$, for $r \neq r'$.

We can decompose the first mutual information term, using the fact that the $\mathcal{A}_r$ are nearly independent:

**Lemma 8.** *Assume that $|\mathcal{V}| \geq 2RD$. Then*

$$I(\mathcal{A}_1, \ldots, \mathcal{A}_R; \hat{f}) \geq \sum_r I(\mathcal{A}_r; \hat{f}) - \frac{2R^2 D^2}{|\mathcal{V}|}.$$

We next relate $I(\mathcal{A}_r; \hat{f})$ to the loss. The intuition for this lemma is that for a fixed $r$ the quantity $\sum_s \ell(s,r)$ is small, then the predictor $\hat{f}$ must contain information about the answer set $\mathcal{A}_r$.

**Lemma 9.** *Assume that $|\mathcal{V}| \geq 2D$. Define $\eta := C\sqrt{\frac{D}{S} \log(2D^2 \log |\mathcal{V}|)}$ for a sufficiently large constant $C$, and assume that $\eta \leq 1$. Then*

$$I(\mathcal{A}_r; \hat{f}) \geq -(1+\eta)\frac{D}{S} \cdot \sum_{s \in [S]} \ell(s,r) + D \log \frac{|\mathcal{V}|}{D} \underbrace{- \frac{2D \log |\mathcal{V}|}{|\mathcal{V}|} - \frac{2D^2}{|\mathcal{V}|} - \eta D - 1}_{\text{lower order term}}$$

Finally, we relate $I(a^*(s,r); \hat{f} \mid \mathcal{A}_r)$ to the loss. Similarly, the intuition for this lemma is that if the loss $\ell(s,r)$ is small, then $\hat{f}$ must contain information about the true association $a^*(s,r)$.

**Lemma 10.** *For all $s, r$,*

$$I(a^*(s,r); \hat{f} \mid \mathcal{A}_r) \geq \log D - \ell(s,r).$$

The proofs for Lemmas 8 to 10 are deferred to Appendix E.3.

Combining Lemmas 8 to 10, we get

$$
\begin{aligned}
B &\geq I(\mathcal{A}_1, \ldots, \mathcal{A}_R; \hat{f}) + \sum_{s,r} I(a^*(s,r); \hat{f} \mid \mathcal{A}_r) \\
&= -(1+\eta)\frac{D}{S} \sum_{s,r} \ell(s,r) + RD \log \frac{|\mathcal{V}|}{D} \underbrace{- \frac{2RD \log |\mathcal{V}|}{|\mathcal{V}|} - \frac{2RD^2}{|\mathcal{V}|} - \eta RD - R - \frac{2R^2 D^2}{|\mathcal{V}|}}_{\text{lower order term}} \\
&\quad + SR \log D - \sum_{s,r} \ell(s,r) \\
&= -\left((1+\eta)\frac{D}{S} + 1\right) \sum_{s,r} \ell(s,r) + SR \log D + RD \log \frac{|\mathcal{V}|}{D} - \varepsilon_{lot},
\end{aligned}
$$

where $\varepsilon_{lot} := \frac{2RD \log |\mathcal{V}|}{|\mathcal{V}|} + \frac{2RD^2}{|\mathcal{V}|} + \eta RD + R + \frac{2R^2 D^2}{|\mathcal{V}|} \ll RD \log \frac{|\mathcal{V}|}{D}$ is a lower order term.

Altogether, we see that in order for all the losses $\ell(s, r)$ to equal zero, we require

$$B \geq SR \log D + RD \log \frac{|\mathcal{V}|}{D} - \epsilon_{lot} \geq SR \log D + (1 - c)RD \log \frac{|\mathcal{V}|}{D}$$

Furthermore, when $p(s, r) = \frac{1}{RS}$, then $\mathbf{L} = \frac{1}{SR} \sum_{s,r} \ell(s, r)$, and the bound becomes

$$B \geq -((1 + \eta)RD + RS) \cdot \mathbf{L} + SR \log D + RD \log \frac{|\mathcal{V}|}{D} - \varepsilon_{lot}$$

$$- ((1 + c)RD + RS) \cdot \mathbf{L} + SR \log D + (1 - c)RD \log \frac{|\mathcal{V}|}{D}.$$

$\square$

### E.3    AUXILIARY LEMMAS

**Lemma 11.** *For random variables $X, Y, Z$,*

$$I(X, Y; Z) \geq I(X; Z) + I(Y; Z) - I(X; Y)$$

*Proof.* By standard properties of mutual information:

$$\begin{aligned}
&I(X, Y; Z) - I(X; Z) - I(Y; Z) \\
&= H(X, Y) - H(X, Y \mid Z) - H(X) + H(X \mid Z) - H(Y) + H(Y \mid Z) \\
&= I(X; Y \mid Z) - I(X; Y) \\
&\geq -I(X; Y).
\end{aligned}$$

$\square$

*Proof of Lemma 8.* By Lemma 11,

$$\begin{aligned}
I(\mathcal{A}_1, \ldots, \mathcal{A}_R; \hat{f}) &\geq \sum_r I(\mathcal{A}_r; \hat{f}) - \sum_r I(\mathcal{A}_r; \mathcal{A}_1, \ldots, \mathcal{A}_{r-1}) \\
&= \sum_r I(\mathcal{A}_r; \hat{f}) - \left( \sum_r H(\mathcal{A}_r) + H(\mathcal{A}_1, \ldots, \mathcal{A}_{r-1}) - H(\mathcal{A}_1, \ldots, \mathcal{A}_r) \right) \\
&= \sum_r I(\mathcal{A}_r; \hat{f}) - \sum_r H(\mathcal{A}_r) + H(\mathcal{A}_1, \ldots, \mathcal{A}_R).
\end{aligned}$$

Since each $\mathcal{A}_r$ is a uniformly random subset of $\mathcal{V}$, we have $H(\mathcal{A}_r) = \log \binom{|\mathcal{V}|}{D}$. Also, we can bound

$$H(\mathcal{A}_1, \ldots, \mathcal{A}_R) = \log \left( \binom{|\mathcal{V}|}{D} \binom{|\mathcal{V}| - D}{D} \cdots \binom{|\mathcal{V}| - (R-1)D}{D} \right).$$

Thus

$$\begin{aligned}
\sum_r H(\mathcal{A}_r) - H(\mathcal{A}_1, \ldots, \mathcal{A}_R) &\leq R \log \frac{\binom{|\mathcal{V}|}{D}}{\binom{|\mathcal{V}| - (R-1)D}{D}} \\
&= R \log \frac{|\mathcal{V}|!(|\mathcal{V}| - RD)!}{(|\mathcal{V}| - D)!(|\mathcal{V}| - (R-1)D)!} \\
&\leq RD \log \frac{|\mathcal{V}|}{|\mathcal{V}| - RD} \\
&\leq \frac{2R^2 D^2}{|\mathcal{V}|},
\end{aligned}$$

where we used the bound $\log \frac{1}{1-x} \leq 2x$ on $(0, \frac{1}{2})$. Plugging in yields the desired bound.    $\square$

*Proof of Lemma 9.* Let $(z_1, \ldots, z_D)$ be a random permutation of $\mathcal{A}_r$. We first aim to relate $I(\mathcal{A}; \hat{f})$ to $I(z_i; \hat{f})$. By the data processing inequality,

$$I(\mathcal{A}_r; \hat{f}) \geq I(z_1, \ldots, z_D; \hat{f}).$$

By Lemma 11,

$$I(z_1, \ldots, z_D; \hat{f}) \geq \sum_i I(z_i; \hat{f}) - \sum_i I(z_i; z_1, \ldots, z_{i-1})$$

$$= \sum_i I(z_i; \hat{f}) - \sum_i H(z_i) + H(z_1, \ldots, z_D).$$

The tuple $(z_1, \ldots, z_D)$ is chosen uniformly at random from $\mathcal{V}^D$, conditioned on all the $z_i$ being distinct. Therefore $H(z_i) = \log |\mathcal{V}|$, and $H(z_1, \ldots, z_D) = \log (|\mathcal{V}| \cdots (|\mathcal{V}| - D + 1))$. Thus

$$\sum_i H(z_i) - H(z_1, \ldots, z_D) = \log \left( \frac{|\mathcal{V}|^D}{|\mathcal{V}| \cdots (|\mathcal{V}| - D + 1)} \right)$$

$$\leq D \log \frac{|\mathcal{V}|}{|\mathcal{V}| - D}$$

$$\leq \frac{2D^2}{|\mathcal{V}|}.$$

Altogether,

$$I(\mathcal{A}_r; \hat{f}) \geq \sum_i I(z_i; \hat{f}) - \frac{2D^2}{|\mathcal{V}|}.$$

Next, using the definition of mutual information and Gibbs' inequality,

$$I(z_i; \hat{f}) = \mathbb{E}_{z_i, \hat{f}} \left[ \log \frac{\mathbb{P}(z_i \mid \hat{f})}{\mathbb{P}(z_i)} \right] \geq \mathbb{E}_{z_i, \hat{f}} \left[ \log \frac{q(z_i \mid \hat{f})}{\mathbb{P}(z_i)} \right]$$

for any probability distribution $q$. Let us define $q$ as follows. First, define $\tilde{f}(s, r) := (1 - \epsilon)\hat{f}(s, r) + \frac{\epsilon}{|\mathcal{V}|} \mathbf{1} \in \Delta_{\mathcal{V}}$, for a small constant $\epsilon$ to be chosen later. Next, define

$$q(z \mid \hat{f}) := \frac{1}{S} \sum_s \tilde{f}(s, r)_z.$$

Plugging in, and observing that $\mathbb{P}(z_i) = \frac{1}{|\mathcal{V}|}$, we get that

$$I(z_i; \hat{f}) \geq \mathbb{E}_{z_i, \hat{f}} \left[ \log \left( \frac{1}{S} \sum_s \tilde{f}(s, r)_{z_i} \right) \right] + \log |\mathcal{V}|.$$

Define $\mathcal{N}_z := \{s : a^*(s, r) = z\}$. Let $\mathcal{E}$ be the event that $|\mathcal{N}_z| \geq M$ for all $z \in \mathcal{A}$. On the event $\mathcal{E}$, we can bound

$$\log \left( \frac{1}{S} \sum_s \tilde{f}(s, r)_{z_i} \right) \geq \log \left( \frac{1}{S} \sum_{s \in \mathcal{N}_{z_i}} \tilde{f}(s, r)_{a^*(s, r)} \right)$$

$$= \log \left( \frac{1}{|\mathcal{N}_{z_i}|} \sum_{s \in \mathcal{N}_{z_i}} \tilde{f}(s, r)_{a^*(s, r)} \right) + \log \frac{|\mathcal{N}_{z_i}|}{S}$$

$$\geq \frac{1}{|\mathcal{N}_{z_i}|} \sum_{s \in \mathcal{N}_{z_i}} \log \tilde{f}(s, r)_{a^*(s, r)} + \log \frac{|\mathcal{N}_{z_i}|}{S}$$

$$\geq \frac{1}{M} \sum_{s \in \mathcal{N}_{z_i}} \log \tilde{f}(s, r)_{a^*(s, r)} + \log \frac{M}{S}.$$

Thus

$$\sum_{i\in[D]} \log\left(\frac{1}{S}\sum_s \tilde{f}(s,r)_{z_i}\right) \geq \frac{1}{M}\sum_{i\in[D]}\sum_{s\in\mathcal{N}_{z_i}} \log \tilde{f}(s,r)_{a^*(s,r)} + D\log\frac{M}{S}$$

$$= \frac{1}{M}\sum_{s\in[S]} \log\tilde{f}(s,r)_{a^*(s,r)} + D\log\frac{M}{S}$$

$$\geq \frac{1-\epsilon}{M}\sum_{s\in[S]} \log\hat{f}(s,r)_{a^*(s,r)} - \frac{S\epsilon}{M}\log|\mathcal{V}| + D\log\frac{M}{S}.$$

On $\overline{\mathcal{E}}$, we have the naive bound

$$\sum_{i\in[D]} \log\left(\frac{1}{S}\sum_s \tilde{f}(s,r)_{z_i}\right) \geq D\log\frac{\epsilon}{|\mathcal{V}|}.$$

Altogether, we have

$$\sum_{i\in[D]} I(z_i;\hat{f})$$

$$\geq \mathbb{E}_{z_i,\hat{f}}\left[\sum_{i\in[D]} \log\left(\frac{1}{S}\sum_s \tilde{f}(s,r)_{z_i}\right)\right] + D\log|\mathcal{V}|$$

$$\geq \mathbb{E}_{z_i,\hat{f}}\left[\mathbf{1}(\mathcal{E})\cdot\left(\frac{1-\epsilon}{M}\sum_{s\in[S]} \log\hat{f}(s,r)_{a^*(s,r)} - \frac{S\epsilon}{M}\log|\mathcal{V}| + D\log\frac{M}{S}\right)\right]$$

$$+ \mathbb{P}(\overline{\mathcal{E}})\cdot D\log\frac{\epsilon}{|\mathcal{V}|} + D\log|\mathcal{V}|$$

$$\geq \frac{1}{M}\mathbb{E}_{z_i,\hat{f}}\left[\sum_{s\in[S]} \log\hat{f}(s,r)_{a^*(s,r)}\right] - \frac{S\epsilon}{M}\log|\mathcal{V}| + D\log\frac{M}{S} + \mathbb{P}(\overline{\mathcal{E}})\cdot D\log\frac{\epsilon}{|\mathcal{V}|} + D\log|\mathcal{V}|$$

$$= -\frac{1}{M}\cdot\sum_{s\in[S]} \ell(s,r) - \frac{S\epsilon}{M}\log|\mathcal{V}| + D\log\frac{M}{S} + \mathbb{P}(\overline{\mathcal{E}})\cdot D\log\frac{\epsilon}{|\mathcal{V}|} + D\log|\mathcal{V}|$$

By Bernstein's inequality and a union bound (a similar such concentration argument was used in the lower bound proof in Allen-Zhu & Li (2024)), there exists a constant $C$ such that $\mathbb{P}(\mathcal{E}) \geq 1-\delta$ for

$$M = \frac{S}{D} - C\sqrt{\frac{S}{D}\log(D/\delta)},$$

as long as $S \geq D\log(D/\delta)$. Set $\epsilon = \frac{1}{|\mathcal{V}|}$, $\delta = \frac{1}{2D\log|\mathcal{V}|}$, and define $\eta := 2C\sqrt{\frac{D}{S}\log(2D^2\log|\mathcal{V}|)} \leq 1$. We have that

$$\frac{M}{S} = \frac{1}{D}\left(1 - C\sqrt{\frac{D}{S}\log(D/\delta)}\right) = \frac{1}{D}\left(1 - \frac{\eta}{2}\right),$$

and thus

$$\frac{S}{M} = \frac{D}{1-\eta/2} \leq D(1+\eta).$$

Therefore

$$\sum_{i\in[D]} I(z_i;\hat{f}) \geq -(1+\eta)\frac{D}{S}\cdot\sum_{s\in[S]} \ell(s,r) - (1+\eta)\frac{D\log|\mathcal{V}|}{|\mathcal{V}|} + D\log\frac{|\mathcal{V}|}{D} + D\log(1-\eta/2) - 1$$

$$\geq -(1+\eta)\frac{D}{S}\cdot\sum_{s\in[S]} \ell(s,r) + D\log\frac{|\mathcal{V}|}{D} - \frac{2D\log|\mathcal{V}|}{|\mathcal{V}|} - \eta D - 1.$$

Altogether, we have

$$I(\mathcal{A}_r; \hat{f}) \geq \sum_i I(z_i; \hat{f}) - \frac{2D^2}{|\mathcal{V}|}$$

$$\geq -(1+\eta)\frac{D}{S} \cdot \sum_{s \in [S]} \ell(s, r) + D \log \frac{|\mathcal{V}|}{D} - \frac{2D \log |\mathcal{V}|}{|\mathcal{V}|} - \frac{2D^2}{|\mathcal{V}|} - \eta D - 1,$$

as desired. $\qquad\square$

*Proof of Lemma 10.* By the definition of mutual information and Gibbs' inequality,

$$I(a^*(s, r); \hat{f} \mid \mathcal{A}_r) = \mathbb{E}_{\mathcal{A}_r}\left[\mathbb{E}_{a^*(s,r),\hat{f}|\mathcal{A}_r}\left[\log \frac{\mathbb{P}(a^*(s, r) \mid \hat{f}, \mathcal{A}_r)}{\mathbb{P}(a^*(s, r) \mid \mathcal{A}_r)}\right]\right]$$

$$= \mathbb{E}_{\mathcal{A}_r}\left[\mathbb{E}_{a^*(s,r),\hat{f}|\mathcal{A}_r}\left[\log \mathbb{P}(a^*(s, r) \mid \hat{f}, \mathcal{A}_r)\right]\right] + \log D$$

$$\geq \mathbb{E}_{\mathcal{A}_r}\left[\mathbb{E}_{a^*(s,r),\hat{f}|\mathcal{A}_r}\left[\log q(a^*(s, r) \mid \hat{f}, \mathcal{A}_r)\right]\right] + \log D$$

where $q(\cdot \mid \hat{f}, \mathcal{A}_r)$ is any distribution over $\mathcal{V}$. Let us define $q$ to be

$$q(a \mid \hat{f}, \mathcal{A}_r) \propto \hat{f}(s, r)_a \cdot \mathbf{1}(a \in \mathcal{A}_r)$$

Since $a^*(s, r) \in \mathcal{A}_r$ always, we have that $q(a^*(s, r) \mid \hat{f}, \mathcal{A}_r) \geq \hat{f}(s, r)_{a^*(s,r)}$, and thus

$$I(a^*(s, r); \hat{f} \mid \mathcal{A}_r) \geq \mathbb{E}_{\mathcal{A}_r}\left[\mathbb{E}_{a^*(s,r),\hat{f}|\mathcal{A}_r}\left[\log \hat{f}(s, r)_{a^*(s,r)}\right]\right] + \log D$$

$$= \mathbb{E}_{\mathcal{D}}\left[\log \hat{f}(s, r)_{a^*(s,r)}\right] + \log D$$

$$= \log D - \ell(s, r).$$

$\qquad\square$

## F   TECHNICAL LEMMAS

**Lemma 12.** *Let $u, v$ be drawn uniformly over the $d$-dimensional sphere of radius 1. Then*

$$\mathbb{E}\left[\langle u, v \rangle^{2p}\right] \leq (2p)^p d^{-p}$$

**Lemma 13** (Hypercontractivity for product distributions). *Let $f : (\mathbb{S}^{d-1})^k \times \mathbb{R}^m \to \mathbb{R}$ be a polynomial of total degree at most $p$. Then*

$$\|f\|_{L^q(\nu_d^{\otimes k} \otimes \mu_m)} \leq (q-1)^{p/2}\|f\|_{L^2(\nu^{\otimes k} \otimes \mu_m)},$$

*where $\nu_d$ is the uniform distribution over the sphere $\mathbb{S}^{d-1}$, and $\mu_m$ is the standard Gaussian in $m$ dimensions.*

Hypercontractivity for the Boolean hypercube (which implies hypercontractivity for Gaussian space) and for the sphere are consequences of Beckner (1975; 1992). To show Lemma 13, one can use similar techniques to the proof of Corollary 12 in Montanaro (2012).

**Lemma 14.** *Let $f : (\mathbb{S}^{d-1})^k \times \mathbb{R}^m \to \mathbb{R}$ be a polynomial of total degree at most $p$. Assume that $\mathbb{E}f \geq 0$, where the expectation is taken with respect to $\nu_d^{\otimes k} \otimes \mu_m$. Then if*

$$\frac{2^p e^{-1} \log^p(1/\delta)\mathrm{Var}(f)}{(\mathbb{E}f)^2} \leq 1,$$

*$f \leq 0$ with probability at most $\delta$.*

*Proof.* By Markov's inequality,

$$\mathbb{P}(f \leq 0) \leq \mathbb{P}(|f - \mathbb{E}f| \geq \mathbb{E}f)$$
$$\leq \mathbb{P}(|f - \mathbb{E}f|^q \geq (\mathbb{E}f)^q)$$
$$\leq \frac{\mathbb{E}[|f - \mathbb{E}f|^q]}{(\mathbb{E}f)^q}.$$

Since $f$ is a degree $p$ polynomial, by Lemma 13 we have that

$$\mathbb{E}[|f - \mathbb{E}f|^q]^{1/q} \leq q^{p/2} \mathrm{Var}(f)^{1/2}.$$

Therefore

$$\mathbb{P}(f \leq 0) \leq \left( \frac{q^p \mathrm{Var}(f)}{(\mathbb{E}f)^2} \right)^{q/2}.$$

Setting $q = 2\log(1/\delta)$, we see that whenever

$$\frac{2^p e^{-1} \log^p(1/\delta) \mathrm{Var}(f)}{(\mathbb{E}f)^2} \leq 1,$$

we have

$$\mathbb{P}(f \leq 0) \leq \left( \frac{q^p \mathrm{Var}(f)}{(\mathbb{E}f)^2} \right)^{q/2} \leq \delta,$$

as desired. $\qquad\square$

### F.1 HERMITE POLYNOMIALS

Let $\mu$ be the standard Gaussian in 1 dimension, and let $L^2(\mu)$ be the function space of square-integrable functions with respect to this Gaussian measure. The Hermite polynomials $\{h_k\}_{k \geq 0}$ form an orthonormal basis of $L^2(\mu)$. In particular, $h_k$ is a degree $k$ polynomial, satisfying

$$\langle h_i, h_k \rangle_{L^2(\mu)} = \delta_{ij}.$$

One useful property of Hermite polynomials is the following:

**Lemma 15.** *Let $\boldsymbol{u}, \boldsymbol{w} \in \mathbb{R}^d$ with $\|\boldsymbol{u}\| = \|\boldsymbol{w}\| = 1$, and let $\boldsymbol{x} \sim \mathcal{N}(0, \boldsymbol{I}_d)$. Then*

$$\mathbb{E}_{\boldsymbol{x}}[h_k(\langle \boldsymbol{u}, \boldsymbol{x} \rangle) h_k(\langle \boldsymbol{w}, \boldsymbol{x} \rangle)] = \langle \boldsymbol{u}, \boldsymbol{w} \rangle^k.$$

Next, let $\mu_d$ be the standard Gaussian in $d$ dimensions. The function space $L^2(\mu_d)$ has an orthonormal basis of Hermite *tensors* $\{\mathbf{He}_k\}_{k \geq 0}$, where $\mathbf{He}_k : \mathbb{R}^m \to \left(\mathbb{R}^d\right)^{\otimes k}$:

**Definition 1.** *Let the kth Hermite tensor $\mathbf{He}_k : \mathbb{R}^m \to \left(\mathbb{R}^d\right)^{\otimes k}$ be defined as*

$$\mathbf{He}_k(\boldsymbol{x}) = (-1)^k \frac{\nabla^k \mu_d(\boldsymbol{x})}{\mu_d(\boldsymbol{x})},$$

*where $\mu_m(\boldsymbol{x}) = (2\pi)^{-d/2} \exp\left(-\frac{1}{2}\|\boldsymbol{x}\|^2\right)$ is the Gaussian density. We remark that each entry of $\mathbf{He}_k(\boldsymbol{x})$ is a degree $k$ polynomial in $\boldsymbol{x}$, and*

The Hermite tensors satisfy the following useful properties:

**Lemma 16** (Properties of Hermite Tensors)**.**

- *(Connection to Hermite Polynomials) If $\boldsymbol{w} \in \mathbb{R}^d$, $\|\boldsymbol{w}\| = 1$, then*

$$h_k(\langle \boldsymbol{w}, \boldsymbol{x} \rangle) = \langle \mathbf{He}_k(\boldsymbol{x}), \boldsymbol{w}^{\otimes k} \rangle$$

- *(Stein's Lemma) For $\boldsymbol{x} \sim \mathcal{N}(0, \boldsymbol{I}_d)$, $f \in L^2(\mu_d)$,*

$$\mathbb{E}_{\boldsymbol{x}}[f(\boldsymbol{x}) \mathbf{He}_k(\boldsymbol{x})] = \mathbb{E}_{\boldsymbol{x}}\left[\nabla^k f(\boldsymbol{x})\right].$$

