# OpenReview forum: "Understanding Factual Recall in Transformers via Associative Memories"
_ICLR.cc/2025/Conference — ICLR 2025 Spotlight_

### Official Review · Reviewer_fXTn · 2024-10-28

**Soundness:** 3
**Presentation:** 2
**Contribution:** 3
**Rating:** 6
**Confidence:** 4

**Summary:**

This paper examines factual recall in shallow Transformers, proposing that transformers can leverage associative memories for near-optimal storage of factual information. The authors demonstrate that linear and multi-layer perceptron (MLP) associative memories achieve storage capacity that scales linearly with the parameter count. They design a synthetic factual recall task to show that a single-layer Transformer can reliably store and retrieve facts using self-attention with or without an MLP. Additionally, their gradient dynamics analysis in a simplified linear attention model reveals a "hallucination" stage, where the model’s loss plateaus and predictions rely solely on relation information.

**Strengths:**

- This work offers a theoretical understanding of how shallow Transformers handle factual recall, specifically revealing linear and MLP associative memory capacity in managing storage.
- The gradient flow analysis, particularly the identification of an intermediate "hallucination" phase, adds depth to the understanding of transformer training dynamics.
- The paper demonstrates mathematical rigor in its proofs regarding associative memory storage capacity

**Weaknesses:**

- The findings may be somewhat narrow, given the reliance on synthetic tasks. While the theoretical insights are valuable, it remains uncertain how well these translate to complex, real-world scenarios involving non-random and interdependent factual associations.
- The study is centered on shallow models, raising questions about the applicability of these findings to deeper, large-scale transformers commonly used in research and industry.
- Certain sections, particularly those describing empirical setups for synthetic tasks, could benefit from added detail. Greater specificity on initialization, parameter tuning, and model configuration across trials would improve clarity and reproducibility.
- More discussion on related work exploring neural networks and attention mechanisms as associative memories [1,2] would strengthen the paper and highlight the novelty of the insight.

[1] Radhakrishnan, Adityanarayanan, Mikhail Belkin, and Caroline Uhler. "Overparameterized neural networks implement associative memory." Proceedings of the National Academy of Sciences 117, no. 44 (2020): 27162-27170.

[2] Le, Hung, Truyen Tran, and Svetha Venkatesh. "Self-attentive associative memory." In International conference on machine learning, pp. 5682-5691. PMLR, 2020.

**Questions:**

- The statement on line 117—that the number of associations scales linearly with the number of parameters (up to log factors)—is somewhat unclear. Without injective assumption, it looks like the number of tokens, not associations, scales linearly. Please elaborate on that point.
- Section 4.2 appears to omit LayerNorm, which is part of the original Transformer layer. Would the theoretical results still hold if LayerNorm were considered?

---

> ### Author Response · Authors · 2024-11-22
> **Response to Reviewer fXTn**
>
> Thank you to the reviewer for your thoughtful and detailed review. We respond to your comments below.
>
> > “While the theoretical insights are valuable, it remains uncertain how well these translate to complex, real-world scenarios involving non-random and interdependent factual associations.”
>
> > “The study is centered on shallow models, raising questions about the applicability of these findings to deeper, large-scale transformers commonly used in research and industry.”
>
> We agree with these comments, but would like to also remark that Allen-Zhu & Li (2024) show empirically that larger-scale models can store an amount of information proportional to their parameter count. Our work demonstrates that a shallow transformer can use associative memories to obtain such linear scaling. Please refer to the global comment.
>
> > “Greater specificity on initialization, parameter tuning, and model configuration across trials would improve clarity and reproducibility.”
>
> Thank you for the suggestion. In Appendix A, we have added additional experimental details.
>
> > “The statement on line 117—that the number of associations scales linearly with the number of parameters (up to log factors)—is somewhat unclear. Without injective assumption, it looks like the number of tokens, not associations, scales linearly. Please elaborate on that point.”
>
> We say that the model can “store” the association $(x, f^*(x))$ if $\arg\max_y u_y^TF(e_x) = f^*(x)$. Theorem 1 states that if $N = \tilde O(d^2)$, then with high probability there exists a model that can store all $N$ associations $\{(x, f^*(x)\}_{x \in [N]}$, as long as $f^*$ is injective. We remark that in the MLP associative memory construction in Theorem 2, we no longer need the injectivity assumption, and thus we can always store $N = \tilde O(md)$ associations regardless of the choice of $f^*$.
>
> > “Would the theoretical results still hold if LayerNorm were considered?”
>
> We expect the theoretical results to still hold with layer norm. On a prompt containing subject $s$ and relation $r$, the transformer will output a vector which is close to $\phi(a^*(s, r))$. Since the embeddings are drawn uniformly on the sphere, as $d$ grows the mean will be close to zero and the variance will concentrate, and as such layer norm would have the same effect on each token.
>
> Thank you for the additional references, we have added discussion on these points to the related work.

---

### Official Review · Reviewer_kBfg · 2024-11-02

**Soundness:** 3
**Presentation:** 3
**Contribution:** 3
**Rating:** 8
**Confidence:** 3

**Summary:**

This papers show shallow transformers can use a combination of associative memories to obtain near optimal storage capacity. The verify this on a synthetic factual recall test.

**Strengths:**

1. The theoretical contributions are solid

**Weaknesses:**

1. The theoretical toy model is rather simple (MLP and one-layer transformer). It is unclear how it generalizes to multi-layer transformer.
2. If I understand correctly, the synthetic setting relies on the fact that noise tokens and subject tokens are disjoint. In reality, usually the last token (not eos in practice) should be somewhat relevant for token selection (for example, “in” as the last token would look for locations). It doesn’t make too much sense to let model do next token prediction after some random tokens.

**Questions:**

1. How did you get the plot for Figure 5 (left)? It seems a bit too smooth. Also, why didn’t the first 2000 GD change the loss? It seems counter-intuitive.
2. Could you give an intuitive explanation for why the model, in theorem 6, learns conditional distribution of relations as opposed to conditional distribution of subject?

---

> ### Author Response · Authors · 2024-11-22
> **Response to Reviewer kBfg**
>
> Thank you to the reviewer for your thoughtful and detailed review. We respond to your comments below.
>
> > “The theoretical toy model is rather simple (MLP and one-layer transformer). It is unclear how it generalizes to multi-layer transformer.”
>
> We agree with this comment, but would like to also remark that Allen-Zhu & Li (2024) show empirically that larger-scale models can store an amount of information proportional to their parameter count. Our work demonstrates that a shallow transformer can use associative memories to obtain such linear scaling. Please refer to the global comment.
>
> > “in reality, usually the last token (not eos in practice) should be somewhat relevant for token selection (for example, “in” as the last token would look for locations). It doesn’t make too much sense to let model do next token prediction after some random tokens”
>
> Our task can be thought of modeling the scenario where the prompt is a question, and the final token in the sequence is a question mark. As you suggest, one alternate task that is also reasonable would be to have the final token in the sequence either be the relation or be a relevant token for the relation (i.e in your example, the token “in” signals that the relation is “location”). We hypothesize that a minor modification of our construction in Section 4.3 succeeds on this task with parameter count remaining proportional to number of facts. A brief sketch is as follows: Each head is associated with a subset $S^{(h)}$ of subjects and a subset $R^{(h)}$ of relations. Setting $W^{(h)}\_{KQ} \propto \sum\_{s \in S^{(h)}, r \in R^{(h)}} \phi(s)\phi(r)^T$, this head attends only to tokens in $S^{(h)}$ assuming that a relation in $R^{(h)}$ is present. We can add a formal description of this construction to the next revision.
>
>
> > “How did you get the plot for Figure 5 (left)? It seems a bit too smooth. Also, why didn’t the first 2000 GD change the loss? It seems counter-intuitive.”
>
> Figure 5 (left) is the loss when training the linear attention model with orthogonal embeddings. Since the embeddings are orthogonal, the dynamics for each subject and relation are decoupled, and thus better behaved than the non-orthogonal case (right).
> The loss does not decrease for the first 2000 steps due to the small initialization, and the existence of a saddle around 0. While the exact saddle only exists for the linear attention model, we do see a similar phenomenology in the softmax attention model (Figure 5; right) where the loss decreases slowly for the first few steps of GD.
>
> > “Could you give an intuitive explanation for why the model, in theorem 6, learns conditional distribution of relations as opposed to conditional distribution of subject?”
>
> In theorem 6, we assume that the number of subjects S is much larger than the number of relations R. Since there are fewer relations, predicting using only the relation is more “useful” than predicting using only the subject. We expect that the reverse order would hold if there are more relations than subjects.

---

### Official Review · Reviewer_Wfua · 2024-11-04

**Soundness:** 3
**Presentation:** 3
**Contribution:** 4
**Rating:** 8
**Confidence:** 4

**Summary:**

This paper is a natural extension to Cabannes et al. (2024) in three ways:

1. It extends the theory to a learnable task with trainable weights, not just studying random patterns
2. It considers the MLP and attention as contributions to the Associative Memory properties of transformers
3. It considers the (single-layer) transformer as a gradient flow

The goals are the same: to study the "memorization capacity" a.k.a. "scaling laws" of a transformer from the perspective of Associative Memory. The authors study this memorization capacity on a synthetic recall task consisting of memorizing subject-relationship correlations.

**Strengths:**

(S1) **Possibly high impact**. If the analyses performed in this paper are shown to hold for deeper transformers (with shared or unshared weights) using softmax activations in the attention and bigger datasets, the paper has the possibility to be high impact -- it would solidify the benefit of viewing transformers as formal Associative Memories, uniting the LLM community and the more niche physics- and math- communities studying associative memories.

(S2) **Good, formal definition of Hallucination**. In a passing statement, this paper defines hallucination as the stage where [L455-460]

> ...the model ignores all other tokens in the sequence $z_{1:T}$ – including the useful subject token $s$ – and predicts based only on the relation r... the model is outputting a plausible, yet ultimately incorrect, answer to the prompt.

As the associative memory perspective of transformers continues to gain traction and the theory of this paper extends to practically sized transformers and real NLP datasets, I can see this formal description of hallucination (i.e., the difference in the Cross-entropy loss of relationships and subjects) as game changing for the study of hallucination in LLMs. Related works do not seem to agree on a definition for "hallucination" except that it encompasses "undesirable predictions" from an LLM.

**Weaknesses:**

(W1) **Unclear application to real Transformers and datasets**. (dual to (S1)) The simplified Transformer architecture studied theoretically and empirically in this work is smaller than real Transformers, uses only a single update step for much of the analysis, only uses a synthetic dataset to validate the theory, and drops the softmax from the attention. It is unclear how the theory would generalize to unshared weights and real datasets where the task is to reproduce sequences of data.


(W2) **No comparison to the empirical scaling laws of modern LLMs**. Like (W1), this paper could be improved if the theoretical results are compared to the empirical results of e.g., the [Chinchilla scaling laws](https://arxiv.org/abs/2203.15556).

(W3) **Associative memories are single-step updates**. One limitation of this work is that the associative memories are not energy-based models where the memories are encoded as fixed points of energy descent dynamics. Indeed, the memory capacity of the linear associative memory model would not scale linearly if formulated as an energy-minimization problem. It would be nice to see memory capacity studied in the recurrent setting.

**Summary**

Despite these weaknesses, I believe that the contributions of this paper earn my vote for acceptance at ICLR. The paper is clearly presented, with testable claims, and sufficient empirical results. My concerns can be addressed in follow-up works by the community.

**Note**

The MLP + attention architecture of Transformers has been explicitly formulated as an Associative Memory (specifically, a Hopfield Network with energy descent dynamics) in [Hoover et al. Energy Transformer (2023)](https://arxiv.org/abs/2302.07253). Please add this work to the related work section.

**Questions:**

(Q1) Standard Transformers typically output a probability distribution over all output tokens. The theory of Eq. (7) only considers the most-likely token (equivalent to setting the sampling temperature of LLMs -> close to 0). Indeed, there is a large debate in the community over the relationship between memorization and generalization. How could the memorization study of this work be scaled to study the *generalization* abilities of Transformers? I feel this theoretical framework could be used to tackle this problem.

(Q2) The paper only studies the causal-language modelling (i.e., GPT-style) objective. How would the theory extend to transformers trained on masked-token (i.e., BERT-style) tasks?

(Q3) Is $\sigma$ in Eq. (2) defined, or just an arbitrary non-linearity? The theory of [Dense Associative Memory](https://arxiv.org/abs/1606.01164) says that the choice of this non-linearity is critical to increasing the memory capacity of Associative Memories

(Q4) The experimental details around Fig 1 are incomplete -- I do not think I could replicate that experiment with the information included in the paper. Additionally, Fig 1 could be improved with error bars and notable "failure cases" e.g., the task of storing one-hot vectors [L130] or under different choices of $\sigma$ in the MLP (see (Q4)).

---

> ### Author Response · Authors · 2024-11-22
> **Response to Reviewer Wfua**
>
> Thank you to the reviewer for your thoughtful and detailed review. We respond to your comments below
>
> > “W1: Unclear application to real Transformers and datasets.”
>
> We agree with this comment, but would like to also remark that Allen-Zhu & Li (2024) show empirically that larger-scale models can store an amount of information proportional to their parameter count. Our work demonstrates that a shallow transformer can use associative memories to obtain such linear scaling. Please refer to the global comment.
>
> > “W2: No comparison to the empirical scaling laws of modern LLMs.”
>
> We briefly discuss one connection to scaling laws in Section 6. For the basic associative memory task, we prove that if the token frequencies are distributed as a power law ($p(x) \propto x^{-\alpha}$, then any model with $B$ parameters must incur a loss of at least $B^{1 - \alpha}$. This lower bound is obtained by an MLP associative memory which stores the $\Theta(B)$ most probable associations, and matches the scaling law with respect to model size obtained in prior works. We agree that it would be an interesting direction of future work to understand what scaling laws are achieved if the subjects and relations in our factual recall task are also distributed according to a power law.
>
> > “W3: Associative memories are single-step updates.”
>
> We remark that the weight matrices in both self-attention and an MLP can be interpreted as these “single-step update” associative memories, as demonstrated in the prior works (Bietti et al. (2023), Cabannes et al. (2024)), which is why we adopt this viewpoint in our work.
>
> > “(Q1) Standard Transformers typically output a probability distribution over all output tokens. The theory of Eq. (7) only considers the most-likely token (equivalent to setting the sampling temperature of LLMs -> close to 0)”
>
> We remark that even though we focus on arg-max prediction, this can be achieved for any fixed sampling temperature by scaling up the magnitude of our constructions to infinity, so that the next token distribution concentrates on the most likely token. We do agree that it would be interesting to design a synthetic task where there are multiple possible valid responses for the next token.
>
> > “How could the memorization study of this work be scaled to study the generalization abilities of Transformers?”
>
> Our factual recall task can be viewed as one instance of generalization, where the model must ignore the noise tokens while storing (s, r, a) tuples; in this case, the model may be able to generalize to sequences with different noise tokens or where the subject and relation are in different positions. In this case, it would be very interesting to prove a finite-sample bound on the number of such sequences the transformer must see in order to obtain such generalization.
>
> > “(Q2)...How would the theory extend to transformers trained on masked-token (i.e., BERT-style) tasks?”
>
> It would be interesting to understand how our theory extends to the masked language modeling setting. One hypothesis is that if the token containing the answer is the one that is masked out, then an identical construction to that in Section 4.3, with $\phi(EOS)$ replaced by $\phi(MASK)$, would succeed.
>
> > “Q3: Is $\sigma$ in Eq. (2) defined, or just an arbitrary non-linearity?”
>
> We require $\sigma$ to be a polynomial of sufficiently high degree. This coincides with the theory for Dense Associative Memories, which require the energy function to also be a polynomial of high degree in order to store many more associations. We expect that this polynomial assumption can be relaxed to instead assume that the Hermite coefficients of $\sigma$ are nonzero, which is a common assumption in prior deep learning theory work and includes standard activations such as ReLU with random bias.
>
> > “(Q4) The experimental details around Fig 1 are incomplete…Fig 1 could be improved with error bars and notable "failure cases" e.g., the task of storing one-hot vectors [L130] or under different choices of $\sigma$ in the MLP."
>
> Thank you for this suggestion. We have added additional experimental details to Appendix A. We will regenerate Figure 1 with more random seeds and error bars and add to the next revision. We remark that in our experiments, the scaling was relatively consistent for different choices of $\sigma$.
>
> Thank you for suggesting the additional reference, we have added it to our related work section.

---

### Author Response · Authors · 2024-11-22
**Global Comment**

Thank you to all of the reviewers for their detailed feedback and the overall positive evaluation of our paper.

## Re: Application of our theory to larger models and more realistic datasets.

The reviewers point out that one limitation of our work is that our theory focuses only on a single-layer transformer. We do acknowledge this point. We would like to additionally remark that our work is motivated by the prior work Allen-Zhu & Li (2024), which shows empirically that larger scale (GPT2 size) transformers trained on a dataset consisting of synthetic biographies can store an amount of information proportional to the model’s parameter count. One of our contributions is that a shallow transformer can provably obtain such linear in number of parameters scaling using associative memories to memorize the factual information, providing evidence that associative memories may be used to store information for large-scale models and more complex factual recall tasks. Understanding how our mechanism generalizes to larger models and more realistic datasets is indeed an interesting direction of future work.

We respond to individual comments below.

---

### Meta-Review · Area_Chair_Ydvd · 2024-12-19

**Metareview:**

The paper offers compelling theoretical insights into transformers' factual recall capabilities through associative memories, demonstrating significant contributions to understanding information storage and retrieval. Despite initial concerns about synthetic tasks, reviewers praised the mathematical rigor, with multiple reviewers supporting acceptance and highlighting the novel analysis of gradient flow dynamics. The authors effectively contextualized their work within broader empirical findings, showing how their shallow transformer model provides evidence for associative memory mechanisms in larger-scale models. Comprehensive responses to reviewer questions and the potential for future research extensions make this paper a strong candidate for acceptance.

**Additional Comments On Reviewer Discussion:**

Reviewers raised concerns about the paper's narrow scope and limited applicability to real-world transformers. The authors addressed these by referencing prior empirical work and emphasizing their contribution of proving linear scaling through associative memories. Detailed technical responses, novel insights into transformer memory mechanisms, and strong theoretical foundations ultimately supported the paper's acceptance.

---

### Decision · Program_Chairs · 2025-01-22

Accept (Spotlight)